# Aligning Frozen LLMs by Reinforcement Learning: An Iterative Reweight-then-Optimize Approach

## Abstract

Aligning large language models (LLMs) with human preferences usually requires fine-tuning methods such as RLHF and DPO. These methods directly optimize the model parameters, so they cannot be used in inference time to improve model performance, nor are they applicable when the model weights are not accessible. In contrast, inference-time methods sidestep weight updates by leveraging reward functions to guide and improve output quality. However, they incur high inference costs, and their one-shot guidance is often based on imperfect reward or value functions, leading to suboptimal outputs. In this work, we present a method named Iterative Reweight-then-Optimize (IRO), a reinforcement learning (RL) framework that performs RL-style alignment of the (frozen) base model without touching its parameters. During training, each iteration *(i)* samples candidates from the base model, *(ii)* resamples using current value functions, and *(iii)* trains a new lightweight value function that guides the next decoding pass. At inference time, the value functions are used to guide the base model generation via a search-based optimization process. We prove that under some mild conditions, IRO is a kind of policy iteration and attains the performance of Best-of-N (BoN) search with exponentially fewer tokens at inference time. Experimental results demonstrate that IRO significantly improves length-controlled win rates on challenging instruction-following benchmarks, such as AlpacaEval 2.0, achieving a substantial performance boost (e.g., $30.71\% \rightarrow 43.80\%$ for `Llama-3-8B-Instruct` and $43.11\% \rightarrow 49.77\%$ for `Llama-3-70B-Instruct` compared against GPT-4 responses). Further, IRO consistently outperforms SOTA inference-time alignment baselines such as BoN and weak-to-strong search, even when using much smaller value functions (of size 1B or 7B) to guide a large base model (of size 6.9B or 70B). Moreover, we demonstrate that multiple value functions can be compressed into a single lightweight value function, substantially reducing memory usage and latency while preserving competitive performance.

## 1 Introduction

Large Language Models (LLMs) have demonstrated impressive capabilities across tasks such as AI assistance (Achiam et al., 2023; Guo et al., 2025) and multi-step reasoning (Bai et al., 2023). To align model behavior with human preferences, most existing approaches rely on training-time techniques, notably Reinforcement Learning with Human Feedback (RLHF) (Ouyang et al., 2022) and Direct Preference Optimization (DPO) (Rafailov et al., 2024b). These methods require access to model parameters and perform direct weight updates using human-labeled data or preference-derived rewards. While effective, such approaches suffer from a fundamental limitation, **that they cannot be applied to further improve model performance or adapt to new domains at inference time**. Indeed, once a model is frozen or deployed, these methods offer no means of further fine-tuning based on downstream user feedback or new reward signals. Any significant shift in user preferences or application context typically necessitates costly re-optimization. These limitations are especially problematic in real-world deployment settings where models must quickly adapt to new instructions, tasks, or evaluation objectives without retraining. For instance, a safety-tuned model may require reward-specific customization depending on user goals. In response, there has been growing interest

Figure 1: Illustration of the proposed Iterative Reweight-then-Optimize (IRO). It consists of two steps: (1) Guided generation step: candidate sequences are first sampled from the base model, then reweighted based on scores from value functions, and finally selected using a reward model. (2) Value training step: a reward model scores the generated sample, and a new lightweight value function is trained via regression. At inference time, the trained value functions are used to guide generation.

in **inference-time alignment methods** (Snell et al., 2024; Mudgal et al., 2023; Xu et al., 2024b), which aim to improve generation quality by leveraging external guidance—such as reward or value functions—without modifying the underlying model weights.

Most inference-time methods assume access to an outcome reward model (ORM), which evaluates the quality of a complete output. ORMs are extremely popular in practice partly because they are relatively easy to obtain: they can be trained using standard preference data or derived from existing evaluation pipelines (e.g., through GPT-4 comparisons, rubric scoring, or upvote logs) (Zhu et al., 2023; Jiang et al., 2023). However, ORM-based inference-time alignment methods face a key trade-off between granularity and efficiency: response-level scoring (such as the popular Best-of-$N$ (BoN) (Nakano et al., 2021; Stiennon et al., 2020)) is coarse and costly and requires a large number of generations to match training-time performance (Gao et al., 2023; Dubois et al., 2023). Methods based on token-level scoring are either inaccurate (such as ARGS (Khanov et al., 2024), which constructs a token-level score function by applying the ORM to partial generations) or computationally intensive (requires full-continuation rollouts (Chakraborty et al., 2024; Huang et al., 2024a)).

Alternatively, one can train an external value function, sometimes also called a process reward model (PRM) to guide generation (Mudgal et al., 2023; Han et al., 2024; Kong et al., 2024). These methods provide finer-grained guidance but face two key limitations: (1) high cost or inaccessibility of training supervision (e.g., OpenAI's PRM800K dataset (Lightman et al., 2023) was constructed through extensive manual annotation of intermediate reasoning steps), and (2) inability to correct or refine model behavior over multiple inference steps, i.e., value functions are used to rescore or filter outputs in a single pass, resulting in suboptimal outputs, especially in challenging or long-horizon tasks. For more detailed discussion of ORM and value function/PRM based methods, we defer to Appendix A.

**Research Question, Proposed Work and Contributions.** Based on the above discussion, this work focuses on addressing the following key research question:

> Given access to a frozen LLM and an ORM, how can we best improve or customize the model at inference time, while minimizing inference-time inference cost?

In this paper, we propose Iterative Reweight-then-Optimize (IRO), a Reinforcement Learning (RL) inspired framework that performs RL-style alignment for the (frozen) base model without touching its parameters. Unlike most existing inference-time alignment approaches, which apply a one-shot improvement, our framework enables successive policy improvement by applying *a sequence* of value functions, trained iteratively via the following three key steps: *(i)* samples candidates from the base model, *(ii)* resamples using current value functions, and *(iii)* trains a new lightweight value function that guides the next decoding pass. At inference time, the sequence of learned value functions is then applied to guide the base model generation via a search-based optimization process. We further show that these multiple value functions can be compressed into a single lightweight value function, substantially reducing memory usage and latency while preserving competitive performance. It is worth mentioning that users can use the proposed IRO to finetune a model on their own dataset in the RL style, following a similar procedure to OpenAI's reinforcement fine-tuning (RFT), but without needing to access the model weights. Please see Fig. 1 for an illustration of the proposed scheme. There are a number of major advantages of the proposed approach compared with the existing approaches mentioned above.

**(1).** Compared with ORM-based and approaches such as ARGS (Khanov et al., 2024) and PRM-based methods (Lightman et al., 2023; Cheng et al., 2025), IRO provides an efficient, fine-grained *token-level* guidance without requiring step-level annotations, guiding the *frozen model* generation towards much higher quality.

**(2).** Unlike most existing methods providing one-shot guidance, IRO enables *multi-iteration* policy improvement by employing lightweight, iteratively trained value functions to approximate iterative policy improvement in the frozen-model setting.

**(3).** Compared with the popular baseline BoN, our approach is significantly more computationally efficient at inference time, and attains the performance of BoN with exponentially fewer tokens.

Our specific technical contributions can be summarized as follows:

**(1)** To begin with, we made a key observation that finding the optimal alignment policy is equivalent to finding a sequence of value functions, each solving a constrained policy optimization problem. Based on this, we design a novel IRO approach on reweighting self-generated data to enable *successive* policy improvement, which theoretically converges to the optimal policy. Furthermore, we prove that under some ideal conditions, IRO achieves the same performance as BoN (i.e., the same probability of identifying the optimal generation), but requires *exponentially fewer* tokens and external queries.

**(2)** Empirically, we provide extensive evidence demonstrating that IRO significantly improves over existing inference-time alignment methods. For instance, IRO improves length-controlled win rates on instruction-following benchmarks AlpacaEval 2.0 compared with the frozen base model (e.g., $30.71\% \rightarrow 43.11\%$ for `Llama-3-8B-Instruct` and $43.11\% \rightarrow 49.77\%$ for `Llama-3-70B-Instruct` compared against GPT-4 responses). Further, IRO consistently outperforms SOTA inference-time alignment baselines such as BoN and weak-to-strong-search, even when using much smaller value functions (of size 1B or 7B) to guide large base models (6.9B or 70B). Additionally, extensive ablation studies have been done to analyze the impact of key algorithmic choices and robustness under imperfect reward models.

**(3)** Although IRO involves multiple value functions at inference-time, we further show that they can be compressed into a single value function. This compression significantly reduces 18-45% memory cost and 24-40% latency while only drop 0–1.7% win rates performance compared with IRO Iter3.

Overall, our method proposed a novel framework to achieve RL-style alignment on a frozen base LLM, enabling multi-iteration policy improvement during inference time. Moreover, we also show that, under some standard assumptions, the proposed method is theoretically equivalent to an RL-based policy optimization algorithm, and that it can be exponentially more efficient than the standard inference-time approaches such as BoN.

## 2 PRELIMINARIES AND PROBLEM FORMULATION

### 2.1 NOTATIONS AND PRELIMINARIES

**The Finite-Horizon MDP Model.** A finite Markov decision process (MDP) is defined by the tuple $(\mathcal{S}, \mathcal{A}, \mathcal{P}, \mu, r, H)$, where $\mathcal{S} = \{\mathcal{S}_h\}_{h=1}^H$ is the state space, $\mathcal{A}$ the action space, $\mathcal{P}$ the transition dynamics, $\mu(\cdot)$ the initial state distribution, $r$ the reward function, and $H$ the horizon. A trajectory is denoted as $\tau := (s_1, a_1, s_2, ...s_H, a_H)$. Further define the corresponding state-action visitation measure as $d_h^\pi(s, a) := \mathbb{E}_{s_1 \sim \mu}[\mathcal{P}^\pi(s_h = s, a_h = a|s_1)]$, and state visitation measures as $d_h^\pi(s) := \sum_{a \in \mathcal{A}} d_h^\pi(s, a)$. The value functions and Q functions under policy $\pi$ as

$$V^\pi(s) := \mathbb{E}_{\tau \sim \pi} \left[ \sum_{i=h}^H r(s_i, a_i) \mid s_h = s \right], Q^\pi(s, a) := \mathbb{E}_{\tau \sim \pi} \left[ \sum_{i=h}^H r(s_i, a_i) \mid s_h = s, a_h = a \right]$$

for all $s_h \in \mathcal{S}_h, a \in \mathcal{A}$, and the advantage function is given by $A^\pi(s_h, a_h) := Q^\pi(s_h, a_h) - V^\pi(s_h)$.

**Token-level MDP Model of LLM.** The text generation process of LLM can be modeled as an episodic time-inhomogeneous MDP. Denote the prompt as $x$ and output continuation as $y := (y_1, ..., y_H)$ with maximum length $H$. At each timestep $h$, the state is $s_h := [x, y_{1:h-1}]$, including the prompt and the generated prefix, while the action $a_h = y_h$ is the next token. Since each state $s_h$ encodes a unique prefix, the state spaces $\{\mathcal{S}_h\}_{h=1}^H$ are mutually disjoint. The transition kernel $\mathcal{P}$ is deterministic, i.e. given state $s_h$ and action $a_h$, $s_{h+1} = \mathcal{P}(s_h, a_h) = [s_h, a_h]$. With an outcome reward (ORM), the reward is given only at the end of the generation Ouyang et al. (2022), i.e., $r(s_h, a_h) = 0$ for all $h < H$, and $r(s_H, a_H) = r(\tau)$ where $r(\tau)$ is the reward for the entire sequence $\tau$. For simplicity, we sometimes refer to both $Q^\pi(s_h, a_h)$ and $V^\pi(s_{h+1})$ as the value function.

**Policy Optimization.** Using the above notations, the policy optimization problem is given by

$$\max_\pi J(\pi) := \max_\pi \mathbb{E}_{s_1 \sim \mu, \tau \sim \pi} \, r(\tau), \tag{1}$$

where $\tau := (s_1, a_1, s_2, a_2, \dots)$ denotes one trajectory, corresponding to one data point with prompt(s) and continuation(s). The goal is to find the optimal policy as $\pi^* = \arg\max_\pi J(\pi)$. The optimal value function is defined as $V^*(s_h) = \mathbb{E}_{\tau \sim \pi^*} \left[ \sum_{i=h}^H r(s_i, a_i) | s_h \right]$.

Since this work focuses on fine-tuning a base model, the above policy optimization problem is conducted on top of a base model, denoted as $\pi_{\text{base}}$. Then problem (1) is equivalent to maximizing the following performance gap:

$$\eta^{\pi_{\text{base}}}(\pi) := J(\pi) - J(\pi_{\text{base}}) \overset{(i)}{=} \sum_{h=1}^H \mathbb{E}_{s_h \sim d_h^\pi, a_h \sim \pi(\cdot|s)} \left[ A^{\pi_{\text{base}}}(s_h, a_h) \right], \tag{2}$$

where $(i)$ is due to the performance difference lemma (see Lemma 1 in the appendix). Thus, the policy optimization problem reduces to seeking a policy $\pi$ that induces maximum expected advantage with respect to the base policy $\pi_{\text{base}}$ under its visitation state distribution $d_h^\pi$.

## 2.2 POLICY OPTIMIZATION VIA LEARNING A SEQUENCE OF VALUE FUNCTIONS.

To maximize the performance gap, we can directly optimize (2). However, it is required to sample the data from the visitation measure $d_h^\pi(\cdot)$, which is infeasible in practice. To address this, we follow the trust region policy optimization (TRPO, see Schulman (2015)), which constructs a more conservative but easy surrogate for the performance gap, and successively optimizes it. Assume that we want to improve a *reference* policy $\pi'$ (not necessarily $\pi_{\text{base}}$). Consider the following surrogate objective:

$$\tilde{\eta}^{\pi'}(\pi) = \sum_{h=1}^H \mathbb{E}_{s_h \sim d_h^{\pi'}, a_h \sim \pi(\cdot|s)} \left[ A^{\pi'}(s_h, a_h) \right]. \tag{3}$$

Note that $s_h$ is sampled from the visitation measure of the reference model, which is easier to obtain. $\tilde{\eta}^{\pi'}(\pi)$ serves as a good approximation to $\eta^{\pi'}(\pi)$ (i.e., $\eta^{\pi_{\text{base}}}(\pi)$ in (2) with $\pi_{\text{base}}$ replaced by $\pi'$) when $\pi$ and $\pi'$ are close in terms of the KL-divergence. Thus, one can instead maximize the surrogate objective $\tilde{\eta}^{\pi'}(\pi)$ while penalizing the KL divergence between $\pi$ and $\pi'$. This naturally leads to a KL-constrained optimization problem, which guarantees monotonic performance improvement at each iteration (Shani et al., 2020; Schulman, 2015). The objective is to maximize a surrogate performance difference while limiting deviation from a reference policy $\pi'$:

$$\max_\pi \quad \tilde{\eta}^{\pi'}(\pi) \tag{4a}$$

$$s.t. \quad \sum_{h=1}^H \mathbb{E}_{s_h \sim d_h^{\pi'}} \left[ D_{\text{KL}} \big( \pi(\cdot|s_h) || \pi'(\cdot|s_h) \big) \right] \leq \epsilon, \; \pi(\cdot|s_h) \in \Delta_\mathcal{A}, \; \forall h \in [H], \tag{4b}$$

where $\Delta_\mathcal{A}$ is the probability simplex on the action space. Solving the above KL-constrained problem (4) yields the following closed-form update for each $h \in [H]$ (Schulman, 2015; Zhang et al., 2024) (See Appendix B.1):

$$\pi_{\text{new}}(a_h|s_h) \propto \pi'(a_h|s_h) \exp \left( \frac{1}{\beta} Q^{\pi'}(s_h, a_h) \right) = \pi'(a_h|s_h) \exp \left( \frac{1}{\beta} V^{\pi'}(s_{h+1}) \right), \tag{5}$$

where $\beta$ is the dual variables of the KL constraints, and $Q^{\pi'}$ is the $Q$ function for the reference policy; the last equality is due to the equivalence between $V$ and $Q$ mentioned before. This update softly reweights the reference policy based on the estimated value function. The parameter $\beta$ serves a critical role that regulates exploration and exploitation: large $\beta$ yields conservative steps close to $\pi'$, while small $\beta$ allows more aggressive exploitation of the value estimates. A *key insight* from (5) is that the policy updates depend only on the current policy's value function.

Of course, since (3) is an approximation of the performance gap, optimizing it does not directly yield the globally optimal policy (Lazić et al., 2021). Fortunately, the closed-form solution (5) offers a simple way to iteratively improve the base policy $\pi_{\text{base}}$, namely, by iteratively learning and applying a sequence of value functions, as shown below:

$$\pi_t(a_h|s_h) \propto \pi_{t-1}(a_h|s_h) \exp \left( \frac{1}{\beta_{t-1}} V^{\pi_{t-1}}(s_{h+1}) \right) \propto \pi_{\text{base}}(a|s) \exp \left( \sum_{i=0}^{t-1} \frac{1}{\beta_i} V^{\pi_i}(s_{h+1}) \right), \tag{6}$$

with $s_{h+1} := [s_h, a_h]$. That is, one can improve the base policy by reweighting it via the exponential of a *sum* of a sequence of value functions. Further, each value function $V^{\pi_{t-1}}$ is trained using trajectories sampled from $\pi_{t-1}$, as the value estimates depend on that policy's induced distribution. In the next section, we will leverage this discussion to develop our proposed algorithm.

## 3 THE PROPOSED ALGORITHM

Based on the discussion from the previous section, one can roughly design an algorithm that alternates between two steps: *(i)* fitting a value function for the current policy, and *(ii)* improving the policy via the TRPO-style update in (5). However, there are a few technical challenges. First, the action space is extremely large, making exact policy updates computationally infeasible. Second, rewards are often sparse—typically provided only at the EOS token (Ouyang et al., 2022)—which complicates value learning due to the long-horizon credit assignment problem.

To deal with the first challenge, we avoid the full action space by sampling a small subset of candidates and performing reweighting only over this sampled subset, retaining the top-scoring candidates in a beam-search manner. To deal with the second challenge, we train the value function using Monte Carlo returns obtained from completed trajectories. The proposed algorithm is detailed below.

**Value Function Estimation.** Given the dataset $\mathcal{D}_t := \{(\tau_t, r(\tau_t))\}$ at the $t$-th iteration, where $\tau_t = (x_t, y_t)$, $x_t$ is sampled from the prompt subset $\mathcal{D}_t^p \subset \mathcal{D}^p$, $y_t$ is the continuation generated by previous policy $\hat{\pi}_{t-1}$, and $r(\tau_t)$ is the reward score (note, we use $\hat{\pi}_{t-1}$ to denote the estimated version of the policy $\pi_{t-1}$; for the precise definition please see (8)). The objective of the value training step is to minimize the discrepancy between the predicted values and the actual return values.

We adopt a direct regression approach, where the predicted values are regressed to the observed returns, following Yang & Klein (2021); Mudgal et al. (2023), which is given by:

$$\hat{V}^{\hat{\pi}_{t-1}} := \arg\min_V \; \mathbb{E}_{\tau \sim \mathcal{D}_t}\left[\sum_{h=1}^{H} \left[V(s_h) - r(\tau)\right]^2\right], \quad t = 1, 2, \cdots \tag{7}$$

where $s_h := [x_t, y_{t,1:h-1}]$ denotes the prefix up to step $h$. Here, $\hat{V}$ is used as an estimator of the value function. We will later show that this estimator is theoretically justified.

**Data generation and reweighting.** In this step, we aim to update the policy and construct the dataset $\mathcal{D}_{t+1} := \{\tau_{t+1}, r(\tau_{t+1})\}$. From the previous step, we know that $\hat{V}^{\hat{\pi}_{t-1}}$ is available. Then according to (6), the log probability of selecting token $a_h$ at state $s_h$ under the estimated policy $\hat{\pi}_t$ is

$$\log \hat{\pi}_t(a_h|s_h) \propto \log \pi_{\text{base}}(a_h|s_h) + \sum_{i=0}^{t-1} \frac{1}{\beta_i} \hat{V}^{\hat{\pi}_i}(s_{h+1}), \tag{8}$$

where $\hat{\pi}_0(a_h|s_h) = \pi_{\text{base}}(a_h|s_h)$. Therefore, given a prompt $x_{t+1}$ sampled from $\mathcal{D}_{t+1}^p \subset \mathcal{D}^p$, we can generate the continuation $y_{t+1}$ using only $\pi_{\text{base}}$ and $\{\hat{V}^{\hat{\pi}_i}\}_{i=0}^{t-1}$. However, updating all actions $a \in \mathcal{A}$ in the vocabulary at each step is computationally infeasible, especially in LLMs (e.g., the vocabulary size of GPT4 is 100,256).

To effectively generate the sequence $y_{t+1}$ according to (8), we adopt a value-guided beam search inspired by Zhou et al. (2024). Firstly, initialize $K$ beams given $x_{t+1}$. For each beam, generate $B$ candidate successors (token chunks of size $L$) from $\pi_{\text{base}}$. Rank the successors using scores computed as a weighted summation of the outputs from multiple value functions, and select the top-$K$ beams for the next step. We summarize the proposed algorithm in Algorithm 3 and 4 in Appendix C.

We note that the candidate generation step (line 6 in Algorithm 4) often results in identical successors, which limits the diversity and hinders the exploration. To mitigate this issue, we incorporate a *diversity-first* principle. Specifically, we first cluster identical candidates, and then we prioritize selecting top-scoring successors across different clusters to maintain diversity (see Figure 7). Formally, given a candidate set $\mathcal{C}$, let $g : \mathcal{C} \to \mathcal{K}$ be a grouping function that maps identical candidates to the same group. For any selected subset $\mathcal{S} \in \mathcal{C}$, we define the diversity measure $\text{Div}(\mathcal{S}) = |\{g(y) : y \in \mathcal{S}\}|$, which is exactly the number of different selected candidates. At the selection stage, we choose the $K$ beams by solving

$$\mathcal{S}^* = \arg\max_{\mathcal{S} \in \mathcal{C}, |\mathcal{S}| = K} [\lambda \cdot \text{Div}(\mathcal{S}) + V(y)], \tag{9}$$

where $\lambda$ is sufficiently large to ensure the diversity. We summarize it in Algorithm 5 in Appendix C.

**Value functions compression.** A practical drawback of IRO is that computing the policy $\hat{\pi}_T$ requires loading all $T$ value functions at inference time, which increases both memory cost and latency. To mitigate this overhead, we introduce a simple yet effective procedure that compresses multiple value functions into a single one.

Given the rollout datasets from all iterations $\mathcal{D}_{0:T-1}$, we use each trained value functions to annotate every token in every trajectory. Then we train a single compressed value function to regress toward these aggregated token-level labels, which can be written as

$$\hat{V}^m := \arg\min_V \ \mathbb{E}_{\tau \sim \mathcal{D}_{0:T-1}} \left[ \sum_{h=1}^{H} \Big[ \sum_{t=0}^{T-1} \frac{1}{\beta_i} \hat{V}^{\hat{\pi}_i}(s_h) - V(s_h) \Big]^2 \right]. \tag{10}$$

The compressed value function $\hat{V}^m$ effectively distills the guidance from all iterations into a single lightweight value function. At inference time, we thus replace all $T$ value functions with just one compressed value function, significantly reducing both memory cost and latency.

## 4 THEORETICAL ANALYSIS

In this section, we present two analyses of the proposed method IRO. First, we conduct a performance analysis of IRO, demonstrating that it progressively approaches the optimal policy's performance through multiple iterations. Second, we analyze the sampling and cost efficiency of IRO compared to the baseline BoN. Through these analyses, we show that IRO achieves strong performance in terms of both effectiveness and efficiency.

### 4.1 PERFORMANCE ANALYSIS OF IRO

To begin with, we introduce the following standard assumptions (Agarwal et al., 2019).

**Assumption 1** (Boundedness of Q function class). *Let $\mathcal{F}$ denote a function class to approximate Q functions. Suppose that $Q^{\pi_t} \in \mathcal{F}$ for all iterations $t \in [T]$. In addition, for any $Q \in \mathcal{F}$, we assume the values is bounded as $0 \leq Q(s,a) \leq r_{\max}$ for all $s \in \mathcal{S}, a \in \mathcal{A}$.*

**Assumption 2** (Concentrability). *Suppose that for all $(s,a) \in (\mathcal{S} \times \mathcal{A})$, the base policy $\pi_{\mathrm{base}}$ can cover $\pi^*$ for all $h \in [H]$, i.e., the following holds:*

$$\max_{h \in [H]} \frac{d_h^{\pi^*}(s,a)}{d_h^{\pi_{\mathrm{base}}}(s,a)} = C_{\mathrm{ST}} < \infty. \tag{11}$$

Next, we proceed to analyze the proposed algorithm. First, recall the discussion after (5), that the choice of $\beta_t$ is crucial in controlling the deviation from the base policy. A natural option is to use a constant $\beta_t = \beta$, under which the algorithm reduces to Natural Policy Gradient (Kakade, 2001), with an $\mathcal{O}(1/t)$ convergence rate established in Xiao (2022) toward the optimal policy for (1). However, from an empirical perspective (see Section 5), we observe that increasing $\beta_t$ over time can lead to more stable improvements in performance. While this adaptive scheme may slow down convergence to the optimal policy compared to the constant $\beta$ case, it introduces greater robustness across iterations. Below, we show that when $\beta_t = \mathcal{O}(\sqrt{t})$, our proposed algorithm still converges to the global optimum, achieving $\mathcal{O}(T^{-\frac{1}{2}})$ convergence rate to the global optimum.

**Theorem 1.** *Suppose Assumption 1 and 2 hold. Consider running the IRO algorithm described in Algorithm 3 for $T$ iteration, and at each iteration $m$ samples are generated (i.e., $|\mathcal{D}_t^p| = m$). By choosing $\beta_t = \sqrt{t+1}/\omega$ with $\omega = \sqrt{\frac{2 \log C_{\mathrm{ST}}}{r_{\max}^2 \log T}}$, there exists a $t_0 \in [T]$ with at least probability $1 - \delta$, such that*

$$J(\pi^*) - J(\hat{\pi}_{t_0}) \leq 2\sqrt{r_{\max}^2 H^2 \log T \log C_{\mathrm{ST}}/T} + 2H\sqrt{C_{\mathrm{ST}} \frac{r_{\max}^2}{m} \log \frac{T|\mathcal{F}|}{\delta}}. \tag{12}$$

Theorem 1 indicates that the error scales with $1/T$ and $1/m$ polynomially. Specifically, the error consists of optimization error from the policy update (6) (the first term in (12)), as well as the value function estimation error (the second term in (12)). This result demonstrates that IRO is a kind of policy iteration and can converge to the optimal policy.

## 4.2 Cost and Sample Efficiency Analysis of IRO Compared with BoN

Next, we compare the cost and sample efficiency of the proposed method IRO with BoN. Here, we consider the following comparison setting. The BoN generates multiple sequences and then utilizes the reward model to select the best one. For IRO, it trains value functions based on the reward model and then follows the guided generation step to generate desired sequences. We aim to compare the cost of BoN and IRO in order to obtain the desired optimal sequence.

Specifically, we define $C_{\text{query}}$ as the total number of queries made to the reward model or value functions, and $C_{\text{token}}$ as the total token needed to be generated. For example, the BoN method requires $C_{\text{query}}(\text{BoN}) = N$ reward model evaluations, and IRO needs $C_{\text{query}}(\text{IRO}) = BKHI$, where $I$ is the number of value functions used, $H$ is the horizon, $K$ is the beam width, and $B$ is the number of successors per step. The proof can be found in the Appendix B.3.

**Proposition 1.** *Consider a finite horizon $H$ setting. Let IRO perform guided generation using $I$ value functions, where the weighted combination of $I$ value functions approximate the gold value function, i.e., $\sum_{i=0}^{I-1} \frac{1}{\beta_i} V_i \approx V^*$. Let decoding proceed in chunks of length $L$, with beam width $K$, and $B$ successors per step, as illustrated in Alg. 4. To achieve an equal probability of sampling the optimal sequence, the relative query cost and token number of the two algorithms satisfy:*

$$\frac{C_{query}(\text{BoN})}{C_{query}(\text{IRO})} = \frac{L}{HI}(BK)^{H/L-1}, \quad \frac{C_{token}(\text{BoN})}{C_{token}(\text{IRO})} = (BK)^{H/L-1}. \tag{13}$$

As shown in the above proposition, under ideal situations, where $I$ value functions approximate $V^*$, IRO achieves higher cost-efficiency than BoN. While the query cost of IRO increases with the number of value functions $I$, the token cost ratio remains unaffected. This advantage becomes especially significant as the horizon $H$ increases while the chunk length $L$ remains fixed, as illustrated in Fig. 2. Besides this, we provide a detailed comparison of computational cost between IRO and other inference-time methods in Appendix E.

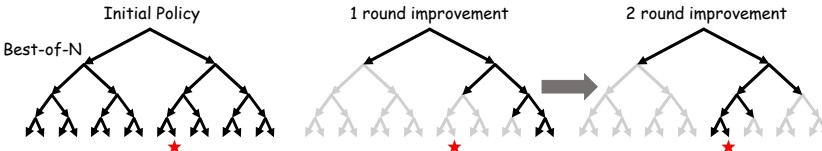

Figure 2: Illustration that IRO is more sample-efficient than BoN. With action space $|\mathcal{A}| = 2$ and horizon $H = 4$, BoN traverses all 30 nodes to find the optimal generation. IRO trains value functions to guide policy to search, prunes the search space, and reaches the optimal solution with only 8 nodes via continual value function updates.

**Remark 4.1.** *In practice, generation quality may fall short of the theoretical guarantee for the following reasons: (1) The IRO is only ran for a few iterations, generating few value functions that may not be able to accurately approximate $V^*$, leading to suboptimal candidate selection; and (2) The reweighting is performed over a sampled subset of the action space rather than the full vocabulary, which introduces additional approximation error during decoding.*

## 5 Experiment

In this section, we provide numerical evaluations of the proposed method IRO. Our experiments demonstrate its effectiveness and flexibility on the TL;DR summarization dataset (Stiennon et al., 2020) and the UltraFeedback dataset (Cui et al., 2023). Specifically, IRO shows: (1) strong weak-to-strong generalization, such as when using 1B value functions to guide a 6.9B policy model, and using 7B value functions to guide a 70B model. (2) IRO scales effectively with the search compute budget (e.g., $U = KB$ in the guided generation step). In addition to tasks with continuous rewards, we also discuss the IRO's generalization to verifiable reward setting in Appendix H.4.

### 5.1 Aligning LLM with TL;DR Dataset

In this experiment, we consider two settings: (1) using 1B value functions to guide a 1B policy, and (2) using 1B value functions to guide a 6.9B policy. We defer more details to Appendix H.1.

**Model and Datasets.** We use the SFT model based on `EleutherAI/pythia` family with size 1B and 6.9B, fine-tuned on TL;DR text-summarization dataset following the methodology outlined in Huang et al. (2024b). We use a `Pythia-1b` based reward model to serve as the reward $r(\cdot)$ needed to run the IRO algorithm.

**Baselines and Evaluation** Our baselines include BoN, three token-level inference-time methods ARGS (Khanov et al., 2024), CARDS (Li et al., 2024), controlled decoding (CD; Mudgal et al. (2023)), and train-based method DPO. Evaluation is based on the win rate of the model-generated summary against the reference summary, evaluated by GPT-4o-mini (see Appendix G for details).

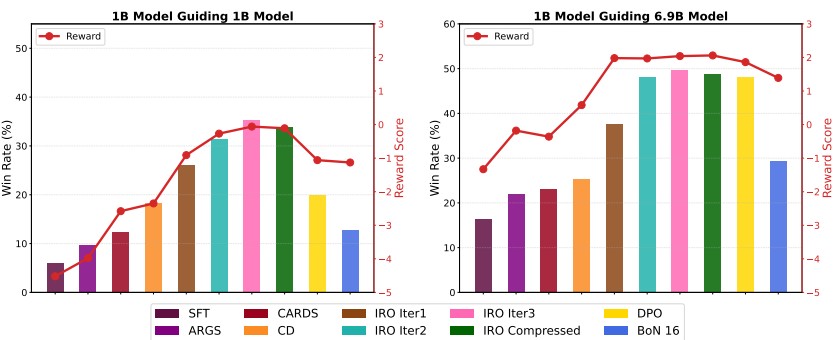

Figure 3: Gold reward score evaluated by a 6.9B reward model (line) and win rates (bar) evaluated by GPT-4o-mini across multiple iterations on 300 samples from the test dataset. The decoding proceeds with $K = 4$ and $B = 4$. For the IRO, we train the algorithm in 3 iterations. During testing, in order to show the benefit of running multiple training iterations, we first evaluate the policy with 1 value function (we call iter 1), and then more value functions (iter 2 and iter 3). We adopt $\beta_i = 1$ for all iterations.

**Results.** As shown in Fig. 3, our method outperforms both the inference-time baseline, including BoN, ARGS, and CARDS, as well as the training-time method DPO. It achieves consistent gains in both reward scores evaluated by a gold reward model and win rate across iterations, which align with our goal of iterative policy improvement (6). For example, with 1B value functions' guidance, IRO improves the win rate over DPO by $14\%$. In the weak-to-strong setting (1B value functions guiding the 6.9B policy), it surpasses the DPO by the third iteration on win rate by $2\%$, demonstrating strong generalization even with a smaller value model. Notably, using only one compressed value function results in only a 1.67% drop in win rate for the 1B policy and 0.9% for the 6.9B policy, while still outperforming IRO Iter2. This shows that the compressed value function retains most information from previous value functions, greatly reducing computational cost.

## 5.2 ALIGNING LLM WITH ULTRAFEEDBACK DATASET

Further, to evaluate the effectiveness of IRO at scale, we conduct a scaled-up experiment. Specifically, 7B value models are used to guide the generation of 8B and 70B models. We defer more details to Appendix H.2.

**Model and Datasets.** We use `Llama-3-8B-Instruct` and `Llama-3-70B-Instruct` as base policies. The reward model is initialized from Mistral-7B-v0.1 and trained with UltraFeedback dataset (Cui et al., 2023).

**Baseline.** We consider the following baselines: (1) BoN with explicit rewards (BoN-E) and implicit rewards (BoN-I), where the implicit reward is calculated as log-probability difference between a base model and a fine-tuned model (Rafailov et al., 2024a); (2) search-based methods Weak-to-Strong Search (Zhou et al., 2024); (3) token-level based method ARGS (Khanov et al., 2024) and controlled decoding (CD; Mudgal et al. (2023)).

**Evaluation.** We evaluate the performance on a standard single-turn instruction-following benchmark, AlpacaEval 2.0 (Li et al., 2023), which consists of 805 prompts from various open-source datasets. Our evaluation reports the length-controlled (LC) win rate (Dubois et al., 2024), which is a metric specifically designed to mitigate biases arising from model verbosity.

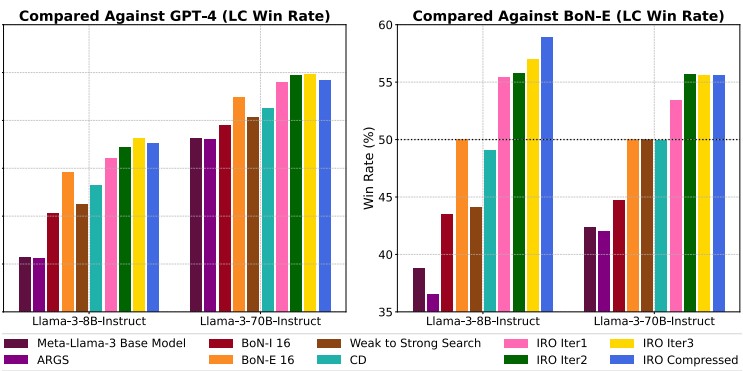

Figure 4: AlpacaEval 2 length-controlled win rate compared against GPT-4 (left) and BoN-E (right) on both 7B and 70B base models, evaluated by GPT-4. We use $\beta_1 = 1, \beta_2 = 2, \beta_3 = 2.5$.

**Results.** As shown in Fig. 4, IRO demonstrates consistent and substantial improvements over the baseline compared against the response generated by GPT-4. Notably, BoN-E is a strong baseline since it significantly outperforms other inference-time algorithms (this result is consistent with findings reported in prior work (Xu et al., 2024b; Liu et al., 2024)). For the IRO, its performance continues to improve as we add value functions to improve the quality of the guidance. This confirms our intuition that rather than requiring additional computational resources to fine-tune the model during training, significant improvements can be achieved successively through value-guided exploration at inference time. Here, we adopt an increasing parameter $\beta_t$ as iterations proceed. This more conservative strategy becomes necessary to ensure stability and prevent excessive deviation from the previous policy. In addition, the compressed value function retains nearly all the benefits of multi-iteration IRO while requiring only a single model at inference time: its win rates drop by only 0.2–1.0% compared with Iter3, while slightly exceeding those of Iter2. This makes IRO more practical, which preserves performance while significantly reducing inference overhead.

## 5.3 EFFICIENCY ANALYSIS

In this section, we empirically evaluate the inference-time efficiency of IRO in terms of decoding latency and peak memory usage compared with BoN and vanilla generation. As shown in Fig. 5, although increasing the number of iterations consistently improves alignment performance, it also leads to higher latency and memory usage. Importantly, our proposed compressing multiple value functions effectively mitigates this overhead.

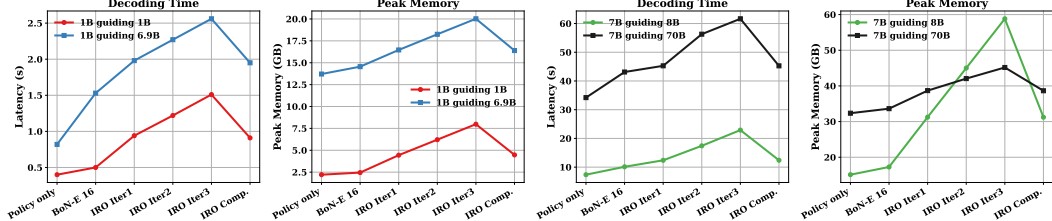

Figure 5: Inference-time latency and peak memory usage. All experiments are conducted over 30 samples on one H100 GPU, except for the 7B guiding 70B configuration, which is measured on 4 H100 GPUs.

## 5.4 TEST-TIME SCALING WITH VALUE FUNCTION

In this section, we explore how the generation quality of IRO, measured by win rate against references and the reward score evaluated by a gold reward model, scales with the search compute budget $U := KB$ across different iterations. Here, both BoN and IRO use the same total number of $UH$ tokens for fair comparison. As shown in Fig. 6 and Fig. 11, increasing the search compute budget

$U$ leads to improvements in both reward scores, which serves as a proxy for generation quality as judged by a gold reward model, and win rate evaluated by GPT-4, indicating the effectiveness of using value functions to guide generation. While the BoN baseline also benefits from a larger search compute budget $U$, its performance saturates early and remains notably below that of IRO. Notably, the value function from the first iteration underperforms in higher budgets (larger $U$), likely due to the limited training data, subsequent iterations effectively correct this and achieve better performance. This iterative improving mechanism can be further illustrated in Fig. 2, where subsequent value functions correct suboptimal paths introduced by earlier guidance.

This empirical observation aligns with our theoretical analysis in Sec. 4: as the number of iterations increases, IRO exhibits convergence behavior, with improved generation quality guided by sequences of value functions. Moreover, the observed trend between win rate and search compute budget $U$ confirms that IRO achieves higher sample efficiency than BoN under comparable compute budgets.

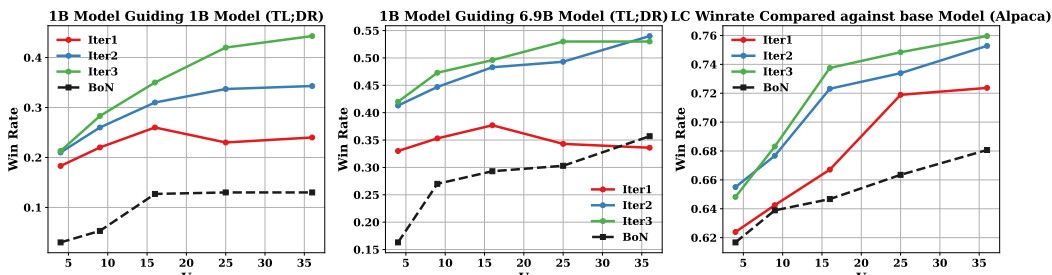

Figure 6: Win rate judged by GPT-4o-mini on the TL;DR task and GPT-4 on the Alpaca subset, showing improvements with increased search compute budget $U$ under three iterations and BoN algorithm. During the search, we set the parameters $K = B = \sqrt{U}$. The first two plots compare against reference summaries, while the third compares against standard generation.

## 5.5 ABLATION STUDY

Due to the page limit, we defer more ablation studies and details to the Appendix H.5.

**Explore different sampling strategies for IRO.** We compare four sampling approaches: beam search (Zhou et al., 2024) with and without the *diversity-first* principle, and stochastic sampling methods with different temperatures. As shown in Fig. 12, incorporating the *diversity-first* principle into beam search significantly improves performance, while stochastic sampling methods are highly sensitive to the temperature parameter $T_{\text{temp}}$.

**Robustness of IRO under imperfect reward model.** To assess robustness, we evaluate IRO using reward models of varying quality on the summarization task (see Table 8 and Table 9) and Instruct-following task (see Table 14). As expected, stronger reward models lead to larger gains in the first iteration. More importantly, even weaker models still enable consistent improvement under the iterative procedure. Thus, while IRO is robust to imperfect reward supervision, the ultimate performance is still bounded by the quality of the reward model, which defines the upper limit of alignment quality.

## 6 CONCLUSION

In this paper, we introduced Iterative Reweight-then-Optimize (IRO), a Reinforcement Learning framework-style alignment of the frozen model at inference time. By learning a sequence of value functions, IRO enables successive policy improvement. We prove that IRO theoretically converges to the optimal policy under certain standard assumptions, and requires exponentially fewer tokens and external function queries than BoN when achieving comparable performance as BoN. In addition, by compressing multiple value functions into a single one, IRO substantially reduces inference-time memory usage and latency while maintaining competitive performance. Empirically, we demonstrate the strong effectiveness of IRO through extensive experiments and ablation studies.

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

# A  RELATED WORK

## A.1  REINFORCEMENT LEARNING WITH HUMAN FEEDBACK

Reinforcement learning from human feedback (RLHF) (Stiennon et al., 2020; Ouyang et al., 2022) is a widely adopted technique for fine-tuning AI systems to align with human preferences and values. Current RLHF approaches typically involve training a reward model using human preference feedback and then fine-tuning the language model via proximal policy optimization (PPO) (Schulman et al., 2017). In addition to PPO, other reinforcement learning solvers such as RLOO (Ahmadian et al., 2024) and GRPO (Shao et al., 2024) have also demonstrated effectiveness in advanced foundation language models. However, optimizing these algorithms for peak performance requires substantial effort and resources, which are often beyond the reach of the open-source community.

## A.2  TRAINING-TIME ALIGNMENT

In an effort to reduce computational overhead in reinforcement learning, alternative alignment strategies such as Direct Preference Optimization (DPO) (Rafailov et al., 2024b) and Inverse Preference Learning (IPL) (Hejna & Sadigh, 2024) remove the need for explicit reward modeling by extracting policies directly from preference data. While this substantially lowers training complexity, these algorithms can be unstable during training (Azar et al., 2023; Xu et al., 2024a), and once the preference dataset is fixed, they offer limited opportunities for further policy improvement.

## A.3  INFERENCE-TIME ALIGNMENT

Although training-time alignment methods are effective, they demand substantial computational and engineering resources. To mitigate these costs, inference-time alignment approaches typically freeze pre-trained models and adjust their outputs during a decoding or test phase, which is handled by smaller, specialized models (Khanov et al., 2024; Mudgal et al., 2023).

The conceptual framework for aligning language models at inference time is rooted in the use of value function (Mudgal et al., 2023) or outcome reward function (Khanov et al., 2024). Based on the type of model used to guide the frozen LLM, we categorize related work into two main approaches: (1) outcome reward guidance, and (2) value function and PRM-based guidance.

### A.3.1  OUTCOME REWARD GUIDANCE

Based on ORM, a widely used approach, Best-of-$N$ (BoN) (Nakano et al., 2021; Stiennon et al., 2020), samples $N$ candidate completions and selects the one with the highest ORM score. While effective, BoN operates at the *response level* and requires a large $N$ (e.g., 1,000–60,000 (Gao et al., 2023; Dubois et al., 2023)) to match training-time performance, resulting in high inference cost and latency. To address this, several methods (Khanov et al., 2024; Liu et al., 2024; Xu et al., 2024b; Li et al., 2024) incorporate *token-level* guidance using only the ORM. ARGS (Khanov et al., 2024) constructs a token-level score function by applying the ORM to partial generations. However, since the ORM is trained only on full outputs, its predictions on incomplete sequences are often unreliable, leading to suboptimal guidance. Furthermore, these methods require the same vocabulary as the frozen LLM, which is often inaccessible in practice, especially for proprietary or closed-source models (Achiam et al., 2023; Bai et al., 2022). Other approaches compute accurate token-level rewards by generating full continuations for each token candidate and scoring them with the ORM (Chakraborty et al., 2024; Huang et al., 2024a). This strategy improves fidelity but introduces prohibitive inference overhead, scaling poorly with sequence length and vocabulary size.

An alternative, but less relevant, line of work leverages *implicit* rewards, such as the log-probability difference between a base model and a fine-tuned model (Rafailov et al., 2024a; Qiu et al., 2024; Zhou et al., 2024; Liu et al., 2024). However, they often misalign with human preferences and can even degrade performance relative to explicit reward models (Liu et al., 2024). This discrepancy arises because log-probability gaps reflect likelihood differences rather than true utility or preference, and thus may reward verbosity, repetition, or high-likelihood but unhelpful completions.

### A.3.2 VALUE FUNCTION AND PRM-BASED GUIDANCE

Another closely related line of work for inference-time alignment is to train an external value function, sometimes also called a process reward model (PRM) to guide generation (Mudgal et al., 2023; Han et al., 2024; Kong et al., 2024; Zhou et al., 2025; Wang et al., 2025). A value function estimates the expected future contribution from a given partial response, enabling more informed token selection during inference. However, applying these methods in practice faces two major challenges. First, training a high-quality value function or PRM —especially one that operates at the step level—typically requires large-scale and fine-grained human-labeled data. For instance, OpenAI's PRM800K dataset (Lightman et al., 2023) was constructed through extensive manual annotation of intermediate reasoning steps, incurring substantial labor and financial costs. This kind of supervision is only feasible in tasks with well-defined reasoning paths, such as math or code generation, and is difficult to generalize to open-ended instruction-following or dialog tasks where step quality is ambiguous. Second, even when a value function or PRM is available, most existing approaches apply it *only once* during decoding. That is, these models are used to resample or filter outputs in a single pass, rather than continuously refining the generation process. Because these models are often imperfect—due to limited training data, model capacity, or domain shift—this single-round guidance can result in suboptimal outputs, especially in challenging or long-horizon tasks.

## B PROOF

### B.1 PROOF OF (5)

*Proof.* In this section, we provide the full derivation of the KL-constrained optimization formulation used in our method. At each policy iteration step, given the current policy $\pi_{\text{old}}$, we aim to maximize the performance gap $\max_\pi \eta^\pi(\pi) := J(\pi) - J(\pi_{\text{old}})$. Following the TRPO (Schulman, 2015) formulation, we consider sampling the data from the visitation measure $d_h^{\pi_{\text{old}}}$ induced by the current policy. This leads to the following KL-constrained optimization problem:

$$\max_\pi \quad \tilde{\eta}^{\pi_{\text{old}}}(\pi) \tag{14a}$$

$$s.t. \quad \sum_{h=1}^{H} \mathbb{E}_{s_h \sim d_h^{\pi_{\text{old}}}} \big[ D_{\text{KL}}\big(\pi(\cdot|s_h)||\pi_{\text{old}}(\cdot|s_h)\big)\big] \leq \epsilon, \tag{14b}$$

$$\sum_{a \in \mathcal{A}} \pi(a|s) = 1, \forall s \in \mathcal{S}_h, \tag{14c}$$

$$\pi(a|s_h) \geq 0, \forall s \in \mathcal{S}_h, a \in \mathcal{A} \quad \text{for all } h \in [H], \tag{14d}$$

where $\tilde{\eta}^{\pi_{\text{old}}}(\pi) = \sum_{h=1}^{H} \mathbb{E}_{s_h \sim d_h^{\pi_{\text{old}}}, a_h \sim \pi(\cdot|s)} \big[ A^{\pi_{\text{old}}}(s_h, a_h)\big]$, $s_h$ is the visitation measure for the state at the $h$th timestep, $d_h^{\pi_{\text{old}}} = \sum_{a \in \mathcal{A}} \mathbb{E}_{s_1 \sim \mu}[\mathcal{P}^\pi(s_h = s, a_h = a|s_1)]$.

To derive the solution, we first write down the *partial* Lagrangian function, which only considers the constraints (14b)-(14c). After solving the partial Lagrangian function, we will show that the constraint (14d) is satisfied.

Let $\beta$ and $\zeta := \{\zeta_{s,h}|s \in \mathcal{S}_h\}$ denote the dual variables of the constraints (14b) and (14c), respectively. Then the partial Lagrangian function can be expressed as below:

$$\mathcal{L}(\pi, \beta, \zeta) := \sum_{h=1}^{H} \mathbb{E}_{s_h \sim d_h^{\pi_{\text{base}}}, a \sim \pi(\cdot|s_h)} \big[ A^{\pi_{\text{base}}}(s_h, a_h)\big] + \beta\Big(\epsilon - \sum_{h=1}^{H} \mathbb{E}_{s_h \sim d_h^{\pi_{\text{base}}}} \big[ D_{\text{KL}}\big(\pi(\cdot|s_h)||\pi_{\text{base}}(\cdot|s_h)\big)\big]\Big)$$

$$+ \sum_{h=1}^{H} \sum_{s \in \mathcal{S}_h} \zeta_{s,h}\big(1 - \sum_{a \in \mathcal{A}} \pi(a|s_h)\big).$$

Through taking partial derivative of $\mathcal{L}(\pi, \beta, \zeta)$ w.r.t. $\pi(a_h|s_h)$, we can obtain the following equation:

$$\frac{\partial \mathcal{L}(\pi, \beta, \zeta)}{\partial \pi(a_h|s_h)} = d_h^{\pi_{\text{base}}}(s_h) A^{\pi_{\text{base}}}(s_h, a_h) - \beta d_h^{\pi_{\text{base}}}(s_h)\Big(-\log \pi_{\text{base}}(a_h|s_h) + \log \pi(a_h|s_h) + 1\Big) - \zeta_{s,h}.$$

Through setting the partial derivative $\frac{\partial \mathcal{L}(\pi, \beta, \zeta)}{\partial \pi(a_h|s_h)}$ to 0, we obtain

$$d^{\pi_{\text{base}}}(s_h) A^{\pi_{\text{base}}}(s_h, a_h) - \beta d^{\pi_{\text{base}}}(s_h)\Big(-\log \pi_{\text{base}}(a_h|s_h) + \log \pi(a_h|s_h) + 1\Big) - \zeta_{s,h} = 0.$$

Then we obtain the closed-form expression of the optimal policy $\pi^*$ as below:

$$\log \pi^*(a_h|s_h) = \frac{A^{\pi_{\text{base}}}(s_h, a_h)}{\beta} + \log \pi_{\text{base}}(a_h|s_h) - 1 - \frac{\zeta_{s,h}}{\beta d^{\pi_{\text{base}}}(s_h)}, \tag{15a}$$

$$\pi^*(a_h|s_h) = \pi_{\text{base}}(a_h|s_h) \exp\Big(\frac{A^{\pi_{\text{base}}}(s_h, a_h)}{\beta}\Big) \exp\Big(-1 - \frac{\zeta_{s,h}}{\beta d^{\pi_{\text{base}}}(s_h)}\Big). \tag{15b}$$

Then according to the expression of $\pi(a_h|s_h)$ in (15b), we obtain the following relation:

$$\pi(a_h|s_h) \propto \pi_{\text{base}}(a_h|s_h) \exp\Big(\frac{1}{\beta} A^{\pi_{\text{base}}}(s_h, a_h)\Big). \tag{16}$$

Based on the constraint (14c), we know that $\pi(\cdot|s_h)$ is a distribution so that $\sum_{a \in \mathcal{A}} \pi(a|s_h) = 1$. Therefore, according to the expressions in (15b) and (16), we can obtain the following expression of the optimal policy $\pi^*$ as below:

$$\pi^*(a_h|s_h) = \frac{\pi_{\text{base}}(a_h|s_h) \exp\Big(\frac{1}{\beta} A^{\pi_{\text{base}}}(s_h, a_h)\Big)}{\sum_{a' \in \mathcal{A}} \pi_{\text{base}}(a'|s_h) \exp\Big(\frac{1}{\beta} A^{\pi_{\text{base}}}(s_h, a')\Big)}.$$

Recall that $A^{\pi_{\text{base}}}(s_h, a_h) := Q^{\pi_{\text{base}}}(s_h, a_h) - V^{\pi_{\text{base}}}(s_h)$, then we can rewrite the expression of $\pi^*(a|s)$:

$$\pi^*(a_h|s_h) = \frac{\pi_{\text{base}}(a_h|s_h) \exp\Big(\frac{1}{\beta}\big(Q^{\pi_{\text{base}}}(s_h, a_h) - V^{\pi_{\text{base}}}(s_h)\big)\Big)}{\sum_{a' \in \mathcal{A}} \pi_{\text{base}}(a'|s_h) \exp\Big(\frac{1}{\beta}\big(Q^{\pi_{\text{base}}}(s_h, a') - V^{\pi_{\text{base}}}(s_h)\big)\Big)}$$

$$= \frac{\pi_{\text{base}}(a_h|s_h) \exp\Big(\frac{1}{\beta} Q^{\pi_{\text{base}}}(s_h, a_h)\Big)}{\sum_{a' \in \mathcal{A}} \pi_{\text{base}}(a'|s_h) \exp\Big(\frac{1}{\beta} Q^{\pi_{\text{base}}}(s_h, a')\Big)}. \tag{17}$$

This completes the proof. $\qquad\square$

### B.2 PROOF OF THEOREM 1

Considering that the exact $Q$ function or value function $Q^{\pi_t}$ is typically unavailable in practice, it is necessary to account for the approximation error. Let $\hat{Q}^{\pi_t}$ denote the estimation of $Q^{\pi_t}$ at the $t$-th iteration via value function training step (7).

More specifically, $\hat{Q}^{\hat{\pi}_t}$ is defined as follows:

$$\hat{Q}^{\hat{\pi}_t} := \arg\min_Q \ \mathbb{E}_{\tau \sim \mathcal{D}_{t+1}}\Big[\sum_{h=1}^H \big[Q(s_h, a_h) - r(\tau)\big]^2\Big], \tag{18}$$

where $\mathcal{D}_{t+1}$ is generated by the policy $\hat{\pi}_t$.

We assume that the estimated function also belongs to the Q function class $\mathcal{F}$, i.e., $\hat{Q}^{\hat{\pi}_t} \in \mathcal{F}$. The policy update then becomes:

$$\hat{\pi}_{k+1}(a|s) \propto \hat{\pi}_t(a|s) \exp(\hat{Q}^{\hat{\pi}_t}(s, a)/\beta_t). \tag{19}$$

We now decompose the performance gap between the optimal policy $\pi^*$ and the current policy $\hat{\pi}_t$ as follows:

$$J(\pi^*) - J(\hat{\pi}_t) \overset{(i)}{=} \sum_{h=1}^{H} \mathbb{E}_{s_h \sim d_h^{\pi^*}, a_h \sim \pi^*(\cdot|s_h)} \left[ A^{\hat{\pi}_t}(s_h, a_h) \right]$$

$$\overset{(ii)}{=} \sum_{h=1}^{H} \mathbb{E}_{s_h \sim d_h^{\pi^*}} \left[ \langle Q^{\hat{\pi}_t}(s_h, \cdot), \pi^*(\cdot|s_h) - \hat{\pi}_t(\cdot|s_h) \rangle \right]$$

$$= \underbrace{\sum_{h=1}^{H} \mathbb{E}_{s_h \sim d_h^{\pi^*}} \left[ \langle \hat{Q}^{\hat{\pi}_t}(s_h, \cdot), \pi^*(\cdot|s_h) - \hat{\pi}_t(\cdot|s_h) \rangle \right]}_{:=\epsilon_1(t)}$$

$$+ \underbrace{\sum_{h=1}^{H} \mathbb{E}_{s_h \sim d_h^{\pi^*}} \left[ \langle Q^{\hat{\pi}_t}(s_h, \cdot) - \hat{Q}^{\hat{\pi}_t}(s_h, \cdot), \pi^*(\cdot|s_h) - \hat{\pi}_t(\cdot|s_h) \rangle \right]}_{:=\epsilon_2(t)}, \tag{20}$$

where step $(i)$ uses the performance difference lemma (Lemma 1), and the step $(ii)$ expands the advantage function as follows:

$$\mathbb{E}_{s_h \sim d_h^{\pi^*}, a_h \sim \pi^*(\cdot|s_h)} \left[ A^{\hat{\pi}_t}(s_h, a_h) \right] = \mathbb{E}_{s_h \sim d_h^{\pi^*}, a_h \sim \pi^*(\cdot|s_h)} \left[ Q^{\hat{\pi}_t}(s_h, a_h) - V^{\hat{\pi}_t}(s_h) \right]$$

$$= \mathbb{E}_{s_h \sim d_h^{\pi^*}, a_h \sim \pi^*(\cdot|s_h)} \left[ Q^{\hat{\pi}_t}(s_h, a_h) \right] - \mathbb{E}_{s_h \sim d_h^{\pi^*}} \left[ V^{\hat{\pi}_t}(s_h) \right]$$

$$= \mathbb{E}_{s_h \sim d_h^{\pi^*}, a_h \sim \pi^*(\cdot|s_h)} \left[ Q^{\hat{\pi}_t}(s_h, a_h) \right] - \mathbb{E}_{s_h \sim d_h^{\pi^*}, a_h \sim \hat{\pi}_t(\cdot|s_h)} \left[ Q^{\hat{\pi}_t}(s_h, a_h) \right]$$

$$= \mathbb{E}_{s_h \sim d_h^{\pi^*}} \left[ \langle Q^{\hat{\pi}_t}(s_h, \cdot), \pi^*(\cdot|s_h) - \hat{\pi}_t(\cdot|s_h) \rangle \right],$$

where the first equality is by the definition of the advantage function, and the third equality follows from the fact that the value function is the expectation of $Q$ under the corresponding policy. The first term $\epsilon_1(t)$ reflects the inherent policy suboptimality due to the iterative policy iteration, while the second term $\epsilon_2(t)$ captures the approximation error resulting from the finite-sample estimation and function approximation for the value function.

Let $t_0$ denote the index that minimizes the performance gap between the optimal policy $\pi^*$ and the iterates $\{\hat{\pi}_t\}_{t=0}^{T-1}$, i.e., $t_0 = \arg\min_{t \in [T]} J(\pi^*) - J(\hat{\pi}_t)$, we can upper-bound the minimal performance gap as the average of performance gap of each iteration:

$$J(\pi^*) - J(\hat{\pi}_{t_0}) \le \frac{1}{\sum_{t=0}^{T-1} 1/\beta_t} \sum_{t=0}^{T-1} \frac{1}{\beta_t} \left( \epsilon_1(t) + \epsilon_2(t) \right), \tag{21}$$

where $\epsilon_1(t)$ and $\epsilon_2(t)$ denote the policy improvement and Q-function approximation errors at iteration $t$, respectively, as defined in (20). The weight $\beta_t$ denotes the dual variable of the KL constraints for the $t$-th iteration and can also be interpreted as the learning rate for iteration $t$.

**Bounding the term $\sum_{t=0}^{T-1} \frac{1}{\beta_t} \epsilon_1(t)$**

Recall the update rule in (19), the policy $\pi_{t+1}$ is updated for each time step $h \in [H]$ according to

$$\hat{\pi}_{t+1}(\cdot|s_h) = \frac{\hat{\pi}_t(\cdot|s_h) \exp(\frac{1}{\beta_t} \hat{Q}^{\hat{\pi}_t}(s_h, \cdot))}{\sum_{a \in \mathcal{A}} \hat{\pi}_t(a|s_h) \exp(\frac{1}{\beta_t} \hat{Q}^{\hat{\pi}_t}(s_h, a))}, \tag{22}$$

where $H$ denotes the episode horizon. Moreover, under Assumption 1, the estimated value function $\hat{Q}^{\hat{\pi}_t}$ is assumed to be uniformly bounded as $\|\hat{Q}^{\hat{\pi}_t}\|_\infty \le r_{\max}$, where $r_{\max}$ is the upper bound on the outcome reward.

Applying the Lemma 2, we obtain

$$\mathbb{E}_{s_h \sim d_h^{\pi^*}} \langle \hat{Q}^{\hat{\pi}_t}(s_h, \cdot), \pi^*(\cdot|s_h) - \hat{\pi}_t(\cdot|s_h) \rangle$$

$$\le \frac{r_{\max}^2}{2\beta_t} + \beta_t D_{\mathrm{KL}}(\pi^*(\cdot|s_h)\|\hat{\pi}_t(\cdot|s_h)) - \beta_t D_{\mathrm{KL}}(\pi^*(\cdot|s_h)\|\hat{\pi}_{t+1}(\cdot|s_h)). \tag{23}$$

Dividing both sides of (23) by $\beta_t$ yields:

$$\frac{1}{\beta_t}\mathbb{E}_{s_h \sim d_h^{\pi^*}}\langle \hat{Q}^{\hat{\pi}_t}(s_h, \cdot), \pi^*(\cdot|s_h) - \hat{\pi}_t(\cdot|s_h)\rangle$$

$$\leq \frac{r_{\max}^2}{2\beta_t^2} + D_{\mathrm{KL}}(\pi^*(\cdot|s_h)||\hat{\pi}_t(\cdot|s_h)) - D_{\mathrm{KL}}(\pi^*(\cdot|s_h)||\hat{\pi}_{t+1}(\cdot|s_h)). \qquad (24)$$

Combine (24), the weighted summation of $\epsilon_1(t)$ can be written as

$$\sum_{t=0}^{T-1}\frac{1}{\beta_t}\epsilon_1(t)$$

$$= \sum_{t=0}^{T-1}\frac{1}{\beta_t}\sum_{h=1}^{H}\mathbb{E}_{s_h \sim d_h^{\pi^*}}\langle \hat{Q}^{\pi_t}(s_h, \cdot), \pi^*(\cdot|s_h) - \hat{\pi}_t(\cdot|s_h)\rangle$$

$$\leq \sum_{h=1}^{H}\mathbb{E}_{s_h \sim d_h^{\pi^*}}D_{\mathrm{KL}}(\pi^*(\cdot|s_h)||\hat{\pi}_0(\cdot|s_h)) + \frac{r_{\max}^2 H}{2}\sum_{t=0}^{T-1}\frac{1}{\beta_t^2}$$

$$\leq H\log(C_{\mathrm{ST}}) + \frac{r_{\max}^2 H}{2}\sum_{t=0}^{T-1}\frac{1}{\beta_t^2}, \qquad (25)$$

where $\hat{\pi}_0 = \pi_{\mathrm{base}}$, the first inequality applies (24), and the second inequality uses the Lemma 3 by bounding the KL divergence using the concentrability constant $C_{\mathrm{ST}} = \max_{h \in [H], s \in \mathcal{S}_h, a \in \mathcal{A}} \frac{d_h^{\pi^*}(s,a)}{d_h^{\pi_{\mathrm{base}}}(s,a)}$.

**Bounding the term** $\sum_{t=0}^{T-1}\frac{1}{\beta_t}\epsilon_2(t)$

The error term $\epsilon_2(t)$ is mainly caused by function approximation error in estimating the value function, which is introduced by using the least-squares objective in (7) using finite samples. As the intermediate rewards are zero and only the terminal reward $r(\tau)$ is assigned, the terminal reward $r(\tau)$ serves as an unbiased Monte Carlo estimate of the Q-value for a given state-action pair $(s_h, a_h)$. That is

$$Q^{\hat{\pi}_t}(s_h, a_h) = \mathbb{E}_{\tau \sim \hat{\pi}_t}\left[\sum_{i=h}^{H}r(s_i, a_i)|s_h, a_h\right], \qquad (26)$$

where $\tau = (s_h, a_h, ..., s_H, a_H)$ is a partial trajectory started from $(s_h, a_h)$. Thus, the relation between the reward and the value function can be built as (note that $\sum_{i=h}^{H}r(s_i, a_i) = r(\tau)$)

$$r(\tau) = Q^{\hat{\pi}_t}(s_h, a_h) + \varepsilon_h, \qquad (27)$$

where the noise term $\varepsilon_h := r(\tau) - \mathbb{E}[\sum_{i=h}^{H}r(s_i, a_i)|s_h, a_h]$ is zero-mean conditioned on $(s_h, a_h)$. As the $\hat{Q}^{\pi_t}$ is the least square solution via

$$\hat{Q}^{\hat{\pi}_t} := \arg\min_{Q} \mathbb{E}_{\tau \sim \mathcal{D}_{t+1}}\left[\sum_{h=1}^{H}\left[Q(s_h, a_h) - r(\tau)\right]^2\right], \qquad (28)$$

where $\mathcal{D}_{t+1}$ is the dataset of $m$ trajectories collected under policy $\hat{\pi}_t$.

To bound the estimation error, we apply the guarantee of least squares (Lemma 8). In particular, we treat each pair $(s_h, a_h)$ as the input $u$, and terminal reward $r(\tau)$ as the noisy label $v$. The conditional distribution $\rho$ in Lemma 8 corresponds to $d_h^{\hat{\pi}_t}$. Since $\hat{Q}^{\hat{\pi}_t}(s, a) \in [-r_{\max}, r_{\max}]$ and $|r(\tau)| \leq r_{\max}$ and belong to the function class $\mathcal{F}$, all assumptions in Lemma 8 are satisfied.

Therefore, for any iteration $t$, with probability at $1 - \delta$, we have

$$\mathbb{E}_{s \sim d_h^{\pi_1}, a \sim \hat{\pi}_0}[|\hat{Q}^{\hat{\pi}_t}(s, a) - Q^{\hat{\pi}_t}(s, a)|] \leq \sqrt{\frac{256 r_{\max}^2}{m}\log\frac{2|\mathcal{F}|}{\delta}}, \qquad (29)$$

where $m$ is the size of the training data, i.e., $m = |\mathcal{D}_{t+1}|$.

To bound the error term $\sum_{t=0}^{T-1} \epsilon_2(t)/\beta_t$, we apply a union bound over $t \in [T]$, we have for all $t \in [T]$ that

$$
\begin{aligned}
\sum_{t=0}^{T-1} \frac{1}{\beta_t} \epsilon_2(k) &= \sum_{h=1}^{H} \sum_{t=0}^{T-1} \frac{1}{\beta_t} \mathbb{E}_{s_h \sim d_h^{\pi^*}} \left[ \langle Q^{\pi_t}(s_h, \cdot) - \hat{Q}^{\pi_t}(s_h, \cdot), \pi^*(\cdot|s_h) - \hat{\pi}_t(\cdot|s_h) \rangle \right] \\
&\leq 2 \sum_{t=0}^{T-1} \frac{1}{\beta_t} \sum_{h=1}^{H} \left| \mathbb{E}_{s_h \sim d_h^{\pi^*}} \left[ Q^{\pi_t}(s_h, \cdot) - \hat{Q}^{\pi_t}(s_h, \cdot) \right] \right| \\
&\leq 2 \sum_{t=0}^{T-1} \frac{1}{\beta_t} \sum_{h=1}^{H} \sqrt{ \mathbb{E}_{s_h \sim d_h^{\pi^*}} \left[ \left| Q^{\pi_t}(s_h, \cdot) - \hat{Q}^{\pi_t}(s_h, \cdot) \right|^2 \right] } \\
&\leq 2 \sum_{t=0}^{T-1} \frac{1}{\beta_t} \sum_{h=1}^{H} \sqrt{ C_{\mathrm{ST}} \mathbb{E}_{s_h \sim d_h^{\pi_1}} \left[ \left| Q^{\pi_t}(s_h, \cdot) - \hat{Q}^{\pi_t}(s_h, \cdot) \right|^2 \right] } \\
&= 2H \sqrt{ C_{\mathrm{ST}} \frac{256 r_{\max}^2}{m} \log \frac{2T|\mathcal{F}|}{\delta} } \sum_{t=0}^{T-1} \frac{1}{\beta_t},
\end{aligned}
\tag{30}
$$

where the first inequality follows the relation $|\pi_1 - \pi_2| < 2$, the second inequality applies the Cauchy-Schwarz inequality, the third follows from Assumption 2, and the final equality uses the value function approximation bound in (29).

In conclusion, by setting the learning rate as $\beta_t = \sqrt{t+1}/\omega$, we obtain the following bound on the performance gap with at least probability $1 - \delta$:

$$
J(\pi^*) - J(\hat{\pi}_{t_0}) \leq \frac{H \log(C_{\mathrm{ST}}) + r_{\max}^2 \omega^2 H \log(T)/2}{\omega \sqrt{T}} + 2H \sqrt{ C_{\mathrm{ST}} \frac{256 r_{\max}^2}{m} \log \frac{2T|\mathcal{F}|}{\delta} }, \tag{31}
$$

where $r_{\max}$ is the upper bound of the Q function.

By choosing $\omega = \sqrt{ \frac{2 \log C_{\mathrm{ST}}}{r_{\max}^2 \log T} }$, with at least probability $1 - \delta$, we obtain

$$
J(\pi^*) - J(\hat{\pi}_{t_0}) \leq 2\sqrt{ r_{\max}^2 H^2 \log T \log C_{\mathrm{ST}}/T } + 2H \sqrt{ C_{\mathrm{ST}} \frac{256 r_{\max}^2}{m} \log \frac{2T|\mathcal{F}|}{\delta} }. \tag{32}
$$

This result shows the trade-off between the number of iterations $T$ and the number of datasets of size $m$ used to estimate the value function.

Theorem 1 differs from Xiao (2022) in the following ways: first, the parameter $\beta_t$ is a constant or decreasing in Xiao (2022), whereas our result utilizes an adaptive increasing $\beta_t$; second, (Xiao, 2022, Theorem 8) considers the sample-based truncation estimation error in the infinite-horizon MDP, while we consider finite-horizon least-square value function estimation error; Third, the rate $\mathcal{O}(T^{-\frac{1}{2}})$ derived in Theorem 1 is slower than the rate of $\mathcal{O}(T^{-1})$ (under constant $\beta_t$) and the superlinear rate (under decreasing $\beta_t$) achieved by Xiao (2022). Note that the last point is reasonable since we are increasing the parameter $\beta_t$, which forces $\theta^t$ to stay closer to $\theta^{t-1}$. Such a choice is practically motivated by the LLM finetuning setting, where it is desirable that the final model is close to the base model (which already has reasonable quality). Further, the $\mathcal{O}(T^{-\frac{1}{2}})$ rate is similar to what can be achieved by Shani et al. (2020), which also uses increasing $\beta_t$'s. The key difference is that the latter considers an infinite-horizon setting, while our work considers the finite-horizon setting.

### B.3 PROOF OF PROPOSITION 1

We begin the analysis in a decision-making setting for the LLM's text generation. Considering a deterministic, episodic, finite-horizon decision process model $(\mathcal{S}, \mathcal{A}, \mathcal{P}, r, H)$ with state space $\mathcal{S}$, action space $\mathcal{A}$, deterministic transition kernel $\mathcal{P}$, and bounded reward function $r$. Starting from an initial state $s_1$ (prompt), the agent chooses an action from the whole action space $a_h \in \mathcal{A}$, receives a

reward $r(s_h, a_h)$, and moves to the next state $s_{h+1}$. Denote the entire input prompt as $x$ and output continuation as $y := (y_1, ..., y_H)$, where $H$ is the maximum generation length. Thus, an complete trajectory is $\tau := (s_1, a_1, s_2, ..., s_H, a_H)$. The optimal trajectory $\tau^* := (s_1^*, a_1^*, s_2^*, ..., s_H^*, a_H^*)$ maximizes the cumulative reward $r(\tau) := \sum_{h=1}^{H} r(s_h, a_h)$. For simplicity, assume $\tau^*$ is unique. At this moment, let us keep the definition of state and actions generic.

Suppose we have access to a gold reward function $r$, and a sequence of value functions $\{V_i\}_{i=0}^{I-1}$ can approximate the gold value function $V^*$. Formally, for a trajectory $\tau = (s_1, a_1, ..., s_H, a_H)$, these functions are given by:

$$r(\tau) = \sum_{h=1}^{H} r(s_h, a_h), \quad V^*(s_h) = \mathbb{E}_\tau \left[ \sum_{i=h}^{H} r(s_i, a_i) | s_h \right]. \tag{33}$$

First, let us clarify the algorithms that we are comparing.

---

**Algorithm 1:** Best-of-N Sampling (BoN)

---

1: **Input:** Initial state $s_1$, base policy $\pi_{\text{base}}$, reward model $r$, number of samples $N$, horizon $H$.
2: **Output:** Trajectory $\tau_{\text{BoN}}$ with the highest reward using BoN.
3: Generate $N$ trajectories $\{\tau^1, \tau^2, ..., \tau^N\}$ started from initial state $s_1$, each with maximum horizon $H$
4: Query the reward model to compute the reward scores $r(\tau)$ for each generated trajectory $\tau \in \{\tau^1, \tau^2, ..., \tau^N\}$
5: Find the trajectory $\tau_{\text{BoN}}$ with the highest reward:

$$\tau_{\text{BoN}} = \arg\max_{\tau \in \{\tau^1, \tau^2, ..., \tau^N\}} r(\tau)$$

6: **return** the trajectory $\tau_{\text{BoN}}$

---

---

**Algorithm 2:** Generation in IRO with $I$ value functions

---

1: **Input:** Initial state $s_1$, beam width $K$, successors per state $B$, chunk length $L$, value function list $\{\hat{V}_i\}_{i=0}^{I-1}$, base policy $\pi_{\text{base}}$, reward model $r$
2: **Output:** Trajectory $\tau_{\text{IRO}}$.
3: Given the initial state $s_1$, generate $U = KB$ partial responses with length $L$ to initialize the candidate set $\mathcal{C} = \{\tau_1, \tau_2, ..., \tau_U\}$
4: **while** $\exists \tau' \in \mathcal{C}$ such that $\tau'$ is incomplete (does not reach [EOS] token) **do**
5:    Query the value function list $\{\hat{V}_i\}_{i=1}^{I}$ to compute the value scores $v_j$ for each candidate $\tau_j$ as $v_j = \sum_{i=0}^{I-1} \frac{1}{\beta_i} \hat{V}_i(\tau_j)$.
6:    Select the top $K$ beams to serve as a parent set.
7:    Generate $B$ successors with length $L$ for each parent node to update the candidate set $\mathcal{C}$.
8: **end while**
9: **return** $\tau_{\text{IRO}} = \arg\max_{\tau' \in \mathcal{C}} r(\tau')$

---

### B.3.1 CHUNK LENGTH $L = 1$

To begin with, let us consider the case where each state and action is a single token. In this case, at each time step $h$, the state is defined as $s_h = [x, y_{1:h-1}]$, including the prompt $x$ and the sequences of tokens $y_{1:h-1}$ generated up to that point. Similarly, each action $a_h = y_h$. Since each state $s_h$ encodes a unique prefix of the output sequence up to position $h - 1$, the state spaces $\{\mathcal{S}_h\}_{h=1}^{H}$ are mutually disjoint. The transition kernel $\mathcal{P}$ is deterministic, i.e. given tokens $s_h = [x, y_{1:h-1}]$ and $a_h$, we have $s_{h+1} = \mathcal{P}(s_h, a_h) = [s_h, a_h]$. In this case, the action space $\mathcal{A}$ is the entire vocabulary.

Given the algorithms above, we assess the computational cost of algorithms (BoN algorithm and IRO algorithm) designed to find the optimal trajectory $\tau^*$ using two key measures:

1. $C_{\text{token}}$: The total number of token costs, representing the number of tokens needed by the algorithm.

2. $C_{\text{query}}$: The query cost, representing the number of calls to an external reward or value function oracle.

Specifically, BoN sampling requires $N$ reward model evaluations to select the optimal trajectory with the highest reward score, while IRO requires querying all $I$ value functions at every decision step $h$ to select candidates.

**Best-of-$N$ algorithm.** Since actions across different time steps $h$ are independent, the probability of sampling the optimal trajectory $\tau^*$ is $\Pr(\tau = \tau^*) = \frac{1}{|\mathcal{A}|^H}$. Sampling $N$ full trajectories uniformly, the probability that at least one of them matches $\tau^*$ is:

$$\Pr[\tau_{\text{BoN}} = \tau^* | N] = 1 - \left(1 - \frac{1}{|\mathcal{A}|^H}\right)^N. \tag{34}$$

When $|\mathcal{A}|^H \gg N$, that is, the number of sampled trajectories is much smaller than the size of full trajectory space, we can use $(1-x)^n \approx 1 - nx$ for small $x$, and this expression can be approximated by:

$$\Pr[\tau_{\text{BoN}} = \tau^* | N] \approx \frac{N}{|\mathcal{A}|^H}. \tag{35}$$

The corresponding costs can then be calculated as follows:

$$C_{\text{token}}(\text{BoN}) = NH,$$
$$C_{\text{query}}(\text{BoN}) = N. \tag{36}$$

**IRO.** Now, consider the inference of IRO with $I$ value functions. At each step $h$, we uniformly sample a subset $\mathcal{A}_h^{\text{sub}} \subset \mathcal{A}$ of a fixed size $|\mathcal{A}_h^{\text{sub}}| = U := KB$. By using value functions $\{\hat{V}_i\}_{i=1}^I$, which can approximate the gold value function $V^*$ (i.e., $\sum_{i=0}^{I-1} \hat{V}_i / \beta_i \approx V^*$), the algorithm selects the action that maximizes the estimated return from this subset:

$$a_h = \arg \max_{a \in \mathcal{A}_h^{\text{sub}}} V^*(s_{h+1}), \tag{37}$$

where $s_{h+1} = [s_h, a]$. For the optimal action $a_h^*$ at $h$-th timestep, if $a_h^* \in \mathcal{A}_h^{\text{sub}}$, then the value functions can be used to correctly select it via (37). Since the subset $\mathcal{A}_h^{\text{sub}}$ is sampled uniformly and independently with size $U$, the probability that it contains the optimal action is

$$\Pr[a_h^* \in \mathcal{A}_h^{\text{sub}}] = 1 - \left(1 - \frac{1}{|\mathcal{A}|}\right)^U.$$

Since each step $h$ is independent, the probability of sampling the optimal trajectory is:

$$\Pr[\tau_{\text{IRO}} = \tau^*] = \left(1 - (1 - \frac{1}{|\mathcal{A}|})^U\right)^H \approx \left(\frac{U}{|\mathcal{A}|}\right)^H, \tag{38}$$

where the approximation holds under the fact that $U$ is much smaller than the size of vocabulary $|\mathcal{A}|$, i.e., $|\mathcal{A}| \gg U$. Therefore, the costs of IRO are given by

$$C_{\text{query}}(\text{IRO}) = UHI,$$
$$C_{\text{token}}(\text{IRO}) = UH. \tag{39}$$

To match the success probability of the Best-of-N method, we require that the probability of (35) and (38) are equal, i.e.,

$$\left(\frac{U}{|\mathcal{A}|}\right)^H = \frac{N}{|\mathcal{A}|^H}. \tag{40}$$

It follows that

$$U^H = N \quad \Leftrightarrow \quad U = \sqrt[H]{N}. \tag{41}$$

Now, we can compare the costs of two methods when achieving the same performance by taking the ratio between terms (36) and (39)

$$\frac{C_{\text{token}}(\text{BoN})}{C_{\text{token}}(\text{IRO})} = \frac{NH}{UH} \overset{(41)}{=} \frac{U^H H}{UH} = U^{H-1},$$

$$\frac{C_{\text{query}}(\text{BoN})}{C_{\text{query}}(\text{IRO})} = \frac{N}{UHI} \overset{(41)}{=} \frac{U^H}{UHI} = \frac{1}{HI}U^{H-1}, \tag{42}$$

### B.3.2 CHUNK LENGTH $L > 1$

In the previous case, each state $s_h$ is the generated prefix of tokens $(x_1, \ldots, x_{h-1})$, and each action $x_h$ is a token from the *entire* vocabulary $\mathcal{A}$. That is, we consider the case where $L = 1$. In practice, choosing $L > 1$ is more efficient because fewer value function generation is needed. In this case, each $a_t$ becomes a chunk of a token, and each $s_t$ is still the accumulation of all the tokens generated up to time $t - 1$. The effective horizon becomes $H_c = H/L$, and the size of the action space will increase to $|\mathcal{A}|^L$.

Similarly as the analysis from the previous subsection, the probability of sampling the optimal trajectory for IRO is

$$\Pr[\tau_{\text{IRO}} = \tau^*] = \left(\frac{U}{|\mathcal{A}|^L}\right)^{H_c} = \frac{U^{H_c}}{|\mathcal{A}|^H}.$$

To ensure the sample success probability with the BoN algorithm, as shown in (35), we require that

$$\frac{U^{H_c}}{|\mathcal{A}|^H} = \frac{N}{|\mathcal{A}|^H} \quad \Leftrightarrow \quad U = \sqrt[H_c]{N}. \tag{43}$$

Therefore, by employing the chunk-level guided generation, the costs of the IRO algorithm will become

$$C_{\text{token}}(\text{IRO}) = U \times L \times H_c = UH,$$

$$C_{\text{query}}(\text{IRO}) = U \times I \times H_c = UHI/L. \tag{44}$$

At this time, the ratio of the costs of two methods when achieving the same performance is

$$\frac{C_{\text{token}}(\text{BoN})}{C_{\text{token}}(\text{IRO})} = \frac{NH}{UH} \overset{(43)}{=} \frac{U^{H_c}H}{UH} = U^{H/L-1} = (BK)^{H/L-1},$$

$$\frac{C_{\text{query}}(\text{BoN})}{C_{\text{query}}(\text{IRO})} = \frac{N}{UHI/L} \overset{(43)}{=} \frac{U^{H_c}}{UHI/L} = \frac{L}{HI}U^{H/L-1} = \frac{L}{HI}(BK)^{H/L-1}, \tag{45}$$

where the last equalities for both the above relations come from the fact that, for IRO, the size of the actions being sampled at each time is given by $U = KB$. The proof is completed.

## C  IRO ALGORITHM

In this section, we give the algorithm flow of our proposed method IRO, which can be separated into two steps: value function training and value function guided generation.

**Algorithm 3:** The Iterative Reweight-and-Optimize (IRO) algorithm

1: **Input:** Prompt dataset $\mathcal{D}^p := \{(x)\}$, a base policy $\pi_{\text{base}}$, reward model $r$.
2: **Output:** Updated policy.
3: Initialize $\hat{\pi}_0 = \pi_{\text{base}}$; Generate $\mathcal{D}_1 = \{\tau_1, r(\tau_1)\}$, where for each $\tau$, $x \sim \mathcal{D}^p$ and $y \sim \hat{\pi}_0$.
4: **for** $t = 1$ to $T - 1$ **do**
5:   Train a value function $\hat{V}^{\hat{\pi}_{t-1}}$ by minimizing the objective (7) using $\mathcal{D}_t$.
6:   Sample a subset $\mathcal{D}_{t+1}^p$ of size $m$ from $\mathcal{D}^p$.
7:   Generate $y_{t+1}$ with $\hat{\pi}_t$ based on $x_{t+1} \in \mathcal{D}_{t+1}^p$ using value-guided generation (Algorithm 4) with $\{\hat{V}^{\hat{\pi}_i}\}_{i=0}^{t-1}$.
8:   Evaluate responses with the reward model $r(\cdot)$ to obtain $\mathcal{D}_{t+1} := \{(\tau_{t+1}, r(\tau_{t+1}))\}$.
9: **end for**
10: **return** $\hat{\pi}_T$ constructed according to (8).

**Algorithm 4:** Value function guided generation at the $t$-th iteration

1: **Input:** Prompt $x$, beam width $K$, successors numbers $B$, chunk length $L$, value functions $\{\hat{V}_i\}_{i=0}^{t-1}$, base policy $\pi_{\text{base}}$, reward model $r(\cdot)$
2: **Output:** Response $y^*$ with the highest reward.
3: Generate $U = KB$ partial responses with length $L$ to form an initial candidate set $\mathcal{C}$.
4: **while** $\exists y' \in \mathcal{C}$ such that $y'$ is incomplete **do**
5:   Compute scores for all $\{y_j\}_{j=1}^N$ as $V(y_j) = \sum_{i=0}^{t-1} \frac{1}{\beta_i} \hat{V}_i(x, y_j)$.
6:   Select top $K$ beams with the highest value scores as parents.
7:   For each parent, expand $B$ successors of length $L$, update new successors to $\mathcal{C}$ while keep size as $U$.
8: **end while**
9: **return** $y^* = \arg\max_{y' \in \mathcal{C}} r(x, y')$

**Algorithm 5:** Diversity-first principle for selection of $K$ beams

1: **Input:** Candidate set $\mathcal{C}$, beam width $K$, combined value function $V(\cdot)$
2: **Output:** Selected beam set $\mathcal{S}^*$
3: Cluster $\mathcal{C}$ into groups via a mapping $g : \mathcal{C} \to \mathcal{K}$.
4: Initialize $\mathcal{S} \leftarrow \emptyset$, and mark all groups in $\mathcal{K}$ as unused.
5: **for** $y \in \mathcal{C}$ sorted by descending $V(y)$ **do**
6:   **if** $|\mathcal{S}| < K$ **and** group $g(y)$ is unused **then**
7:     Add $y$ to $\mathcal{S}$ and mark group $g(y)$ as used.
8:   **end if**
9: **end for**
10: **for** $y \in \mathcal{C}$ sorted by descending $V(y)$ **do**
11:   **if** $|\mathcal{S}| < K$ **and** $y \notin \mathcal{S}$ **then**
12:     Add $y$ to $\mathcal{S}$.
13:   **end if**
14: **end for**
15: **return** $\mathcal{S}^*$ as the final selected beam set

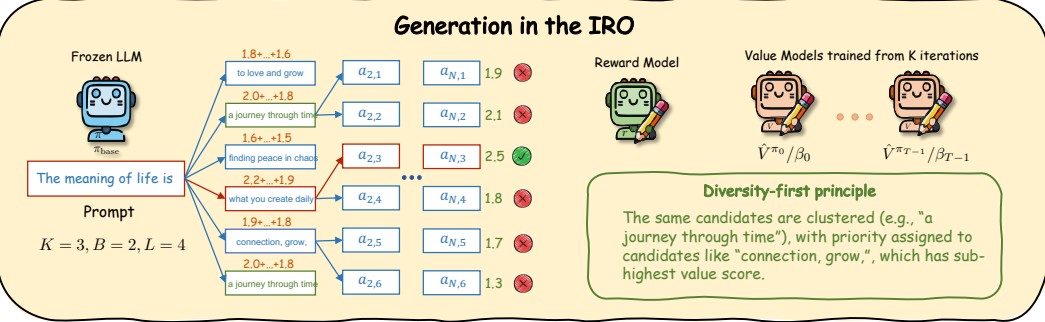

Figure 7: Illustration of decoding process with diversity-first principle.

# D AUXILIARY LEMMAS

**Lemma 1** (Lemma C.4 in Chang et al. (2024)). *For any policy $\pi$ and $\pi'$, the performance difference can be expressed as below:*

$$\eta^{\pi'}(\pi) = \sum_{h=1}^{H} \mathbb{E}_{s_h \sim d_h^\pi, a_h \sim \pi(\cdot|s_h)}\left[A^{\pi'}(s_h, a_h)\right] \tag{46}$$

*where $d_h^\pi(s) := \mathbb{E}_{s_1 \sim \mu}[\sum_{h=1}^{H} \mathcal{P}^\pi(s_h = s|s_1)]$ denotes the state visitation measure.*

**Lemma 2.** *Let $h \in [H]$ be any timestep. Assume the policy $\pi_{t+1}$ is updated from $\pi_t$ via:*

$$\pi_{t+1}(\cdot|s_h) = \frac{\pi_t(\cdot|s_h)\exp(\frac{1}{\beta_t}f(s_h, \cdot))}{\sum_{a \in \mathcal{A}} \pi_t(a|s_h)\exp(\frac{1}{\beta_t}f(s_h, a))},$$

*and assume that the function $f$ is bounded as $\|f\|_\infty \le Q_{\max}$. Then, the following inequality holds:*

$$\langle f(s_h, \cdot), \pi^*(\cdot|s_h) - \pi_t(\cdot|s_h)\rangle$$
$$\le \frac{Q_{\max}^2}{2\beta_t} + \beta_t D_{\mathrm{KL}}(\pi^*(\cdot|s_h)\|\pi_t(\cdot|s_h)) - \beta_t D_{\mathrm{KL}}(\pi^*(\cdot|s_h)\|\pi_{t+1}(\cdot|s_h)) \tag{47}$$

*Proof.* For convenience, we omit the state dependence $s_h$ and use simplified notation: we write $\pi(\cdot|s_h)$ as $\pi$ and $Q(s_h, \cdot)$ as $Q$. For any policy $p$, we have

$$\langle f, p - \pi_{t+1}\rangle = \left\langle \beta_t \log \frac{\pi_{t+1}}{\pi_t} + \log Z_t,\ p - \pi_{t+1}\right\rangle$$
$$= \left\langle \beta_t \log \frac{\pi_{t+1}}{\pi_t},\ p\right\rangle + \langle \log Z_t,\ p\rangle - \left\langle \beta_t \log \frac{\pi_{t+1}}{\pi_t},\ \pi_{t+1}\right\rangle - \langle \log Z_t,\ \pi_{t+1}\rangle$$
$$= \beta_t \left\langle \log \frac{\pi_{t+1}}{p},\ p\right\rangle + \beta_t \left\langle \log \frac{p}{\pi_t},\ p\right\rangle - \beta D_{\mathrm{KL}}(\pi_t\|\pi_{t+1})$$
$$= -\beta_t D_{\mathrm{KL}}(p\|\pi_{t+1}) + \beta_t D_{\mathrm{KL}}(p\|\pi_t) - \beta_t D_{\mathrm{KL}}(\pi_t\|\pi_{t+1}).$$

where $Z_t = Z_t(s_h)$ is the normalization term.

Thus, we have

$$\langle f, p - \pi_t\rangle = \langle f, p - \pi_{t+1}\rangle + \langle f, \pi_{t+1} - \pi_t\rangle$$
$$\le -\beta_t D_{\mathrm{KL}}(p\|\pi_{t+1}) + \beta_t D_{\mathrm{KL}}(p\|\pi_t) + \langle f, \pi_{t+1} - \pi_t\rangle$$
$$- \frac{\beta_t}{2}\|\pi_{t+1} - \pi_t\|_1^2 + \|Q^{\pi_t}\|_\infty \|\pi_{t+1} - \pi_t\|_1$$
$$\le -\beta_t D_{\mathrm{KL}}(p\|\pi_{t+1}) + \beta_t D_{\mathrm{KL}}(p\|\pi_t) + \frac{\|f\|_\infty^2}{2\beta_t}$$

where the first inequality uses Pinsker's inequality (Lemma 5), and Hölder's inequality (Lemma 6), and the second inequality uses the Cauchy–Schwarz inequality (Lemma 4). $\qquad\square$

**Lemma 3** (Proposition B.5 in Chang et al. (2024)). *Let $\pi^*$ denote the optimal policy and $\pi_{\mathrm{base}}$ be a initial policy. Then the following inequality holds:*

$$\sum_{h=1}^{H} \mathbb{E}_{s_h \sim d_h^*} D_{\mathrm{KL}}(\pi^*(\cdot|s_h)\|\pi_{\mathrm{base}}(\cdot|s_h)) \le H \log\left(\max_{h \in [H], s \in \mathcal{S}_h, a \in \mathcal{A}} \frac{d_h^{\pi^*}(s, a)}{d_h^{\pi_{\mathrm{base}}}(s, a)}\right), \tag{48}$$

*where $d_h^{\pi^*}(s, a)$ and $d_h^{\pi_{\mathrm{base}}}(s, a)$ denote the state-action vision measures under $\pi^*$ and $\pi_{\mathrm{base}}$, respectively.*

*Proof.* The proof is the same as in Chang et al. (2024).

$$\sum_{h=1}^{H} \mathbb{E}_{s_h \sim d_h^{\pi^*}} \left[ D_{\mathrm{KL}}(\pi^*(\cdot|s_h) \| \pi_{\mathrm{base}}(\cdot|s_h)) \right] = \sum_{h=1}^{H} \sum_{s \in \mathcal{S}_h} d_h^{\pi^*}(s) \sum_{a \in \mathcal{A}} \pi^*(a|s) \log \frac{\pi^*(a|s)}{\pi_{\mathrm{base}}(a|s)}$$

$$= \sum_{h=1}^{H} \sum_{s \in \mathcal{S}_h, a \in \mathcal{A}} d_h^{\pi^*}(s,a) \log \frac{\pi^*(a|s)}{\pi_{\mathrm{base}}(a|s)}$$

$$\leq \sum_{h=1}^{H} \sum_{s \in \mathcal{S}_h, a \in \mathcal{A}} d_h^{\pi^*}(s,a) \log \frac{\pi^*(a|s)}{\pi_{\mathrm{base}}(a|s)} + \sum_{h=1}^{H} \sum_{s \in \mathcal{S}_h} d_h^{\pi^*}(s) \log \frac{d_h^{\pi^*}(s)}{d_h^{\pi_{\mathrm{base}}}(s)}$$

$$= \sum_{h=1}^{H} \sum_{s \in \mathcal{S}_h, a \in \mathcal{A}} d_h^{\pi^*}(s,a) \log \frac{\pi^*(a|s)}{\pi_{\mathrm{base}}(a|s)} + \sum_{h=1}^{H} \sum_{s \in \mathcal{S}_h, a \in \mathcal{A}} d_h^{\pi^*}(s,a) \log \frac{d_h^{\pi^*}(s)}{d_h^{\pi_{\mathrm{base}}}(s)}$$

$$= \sum_{h=1}^{H} \sum_{s \in \mathcal{S}_h, a \in \mathcal{A}} d_h^{\pi^*}(s,a) \log \frac{d_h^{\pi^*}(s,a)}{d_h^{\pi_{\mathrm{base}}}(s,a)}$$

$$\leq H \log \left( \max_{h \in [H], s \in \mathcal{S}_h, a \in \mathcal{A}} \frac{d_h^{\pi^*}(s,a)}{d_h^{\pi_{\mathrm{base}}}(s,a)} \right).$$

The proof is completed. □

# E   COMPUTATIONAL COMPLEXITY ANALYSIS

In this section, we provide a comprehensive computational complexity analysis of IRO, covering both inference time cost and the offline cost of value-function training. We first compare the inference-time complexity of IRO with other inference-time alignment methods—such as chain-of-thought (CoT) (Wei et al., 2022), BoN, and ARGS (Khanov et al., 2024). Secondly, we present the offline computational cost of IRO required to train value functions and generate rollouts across iterations.

**Inference-time computational complexity analysis**   Assume that the total horizon is $H$ (number of tokens to be generated), and a beam-search-like algorithm selects $K$ candidates and generates $B$ successors for each candidate, which maintains $KB$ search width. For IRO at inference time, it uses $I$ value functions to guide chunk-level search with chunk length $L$. For a fair comparison, here we consider $N = KB$ (where N is the # of generations used in the BoN method).

In Table 1, we present a comprehensive comparison of FLOPs and latency for various methods.

1. FLOPs: Total number of floating-point operations (e.g., additions, multiplications) executed by the algorithm.

2. Latency: The theoretical duration it takes an algorithm to produce a desired output, incorporating both compute and memory overhead. Here, it includes per-token generation and periodic reward model calls.

Since the policy model and reward model may differ in size, we separate each metric into policy and reward. We define $T_{\mathrm{dec}}$ as the time to decode one token by base LLM, and $T_{\mathrm{rm}}$ as the time to score by reward model or value function.

Consider that the policy and reward model are both decoder-only and use KV cache, given prefix length $l$, the FLOPs complexity is $O(l)$ to generate one token. Thus, CoT generates $H$ tokens requires $\sum_{i=1}^{H} O(i) = O(H^2)$ complexity. For BoN, it generates $N$ trajectories and needs the reward model to query $N$ trajectories, so the FLOPs complexity is $O(NH^2)$.

For the method ARGS, it requires querying the reward model to decide every token generation, introducing high reward latency $O(HKBT_{\mathrm{rm}})$. Our algorithm IRO uses chunk-level beam search with chunk length $L$, and employs $I$ value functions, which leads to $O\left(\frac{HIKB}{L}T_{\mathrm{rm}}\right)$ reward model latency.

| Method | FLOPs (Policy) | FLOPs (Reward) | Latency (Policy) | Latency (Reward) |
|--------|----------------|----------------|------------------|------------------|
| CoT | $O(H^2)$ | $-$ | $O(HT_{\text{dec}})$ | $-$ |
| BoN | $O(NH^2)$ | $O(NH^2)$ | $O(HT_{\text{dec}})$ | $O(NT_{\text{rm}})$ |
| ARGS | $O(H^2KB)$ | $O(H^2KB)$ | $O(HT_{\text{dec}})$ | $O(HKBT_{\text{rm}})$ |
| IRO | $O(H^2KB)$ | $O(H^2KBI)$ | $O(HT_{\text{dec}})$ | $O\left(\frac{HIKB}{L}T_{\text{rm}}\right)$ |

Table 1: FLOPs and latency comparison for policy and reward model across different methods. Here $-$ indicates that the method does not involve this complexity.

**Offline computational complexity for IRO** The offline cost of IRO consists of (1) rollout genera­tion and (2) lightweight value-function training.

At the i-th iteration, the frozen policy is guided by $(i-1)$ previously trained value functions. Let the rollout dataset $|\mathcal{D}| = D$, sequence length $H$, beam width $K$, and select $B$ successors in each selection. Since the policy model and value function may differ in size, we use $C_p$ and $C_v$ to denote the per-token cost of the frozen policy and value function, respectively. Thus, the total rollout compute cost at iteration $i$ is

$$O(H^2KBDC_p) + O(DH^2KB(i-1)C_v).$$

After obtaining the data, IRO trains a lightweight value function via supervised regression, which is comparable to SFT and substantially cheaper than training the full policy model.

Aggregating over $I$ iterations, the total offline training cost is

$$O(H^2KBDIC_p) + O(DH^2KBI^2C_v),$$

plus $I - 1$ value function training cost.

# F TECHNICAL LEMMAS

**Lemma 4** (Cauchy-Schwarz Inequality). *For $u, v \in \mathbb{R}^d$, we have*

$$\langle u, v \rangle \leq \|u\|\|v\| \leq \frac{1}{2}\|u\|_2^2 + \frac{1}{2}\|v\|_2^2$$

**Lemma 5** (Pinsker's inequality). *For any two distributions $p$ and $q$, there is always*

$$D_{\text{KL}}(p\|q) \geq \frac{1}{2}\|p - q\|_1^2$$

**Lemma 6** (Hölder's inequality). *Let $p, q > 1$ such that $\frac{1}{p} + \frac{1}{q} = 1$. If $f \in L^p$ and $g \in L^q$, then $fg \in L^1$, and*

$$\|fg\|_1 \leq \|f\|_p \cdot \|g\|_q.$$

**Lemma 7** (Jensen's Inequality). *Suppose that $\phi(w)$ is a convex function on $\Omega$. Consider $w_1, \cdots, w_m \in \Omega$, and non-negative numbers $\alpha_1, \cdots, \alpha_m \in \mathbb{R}$ so that $\sum_{i=1}^m \alpha_i = 1$. Then,*

$$\phi\left(\sum_{i=1}^m \alpha_i w_i\right) \leq \sum_{i=1}^m \alpha_i \phi(w_i).$$

**Lemma 8** (Least Squares Guarantee; Lemma 15 in Song et al. (2022)). *Fix any $R > 0$, $\delta \in (0,1)$ and assume we have a class of real valued functions $\mathcal{H} : \mathcal{U} \mapsto [-R, R]$. Suppose we have $K$ i.i.d samples $\{(u_k, v_k)\}_{k=1}^K$ where $u_k \sim \rho$ that may depend on past observations and $v_k$ is sampled via the conditional probability $p(\cdot \mid u_k)$:*

$$v_k \sim p(\cdot \mid u_k) := h^*(u_k) + \epsilon_k,$$

*where $h^* \in \mathcal{H}$ and $\{\epsilon_k\}_{k=1}^K$ are independent random variables such that $\mathbb{E}[v_k \mid u_k] = h^*(u_k)$. Additionally, suppose that $\max_k |v_k| \leq R$ and $\max_u |h^*(u)| \leq R$. Then the least square solution $\hat{h} \leftarrow \arg\min_{h \in \mathcal{H}} \sum_{k=1}^K (h(u_k) - v_k)^2$ satisfies with probability at least $1 - \delta$,*

$$\mathbb{E}_{x \sim \rho}\left[\left(\hat{h}(u) - h^*(u)\right)^2\right] \leq \frac{256R^2 \log(2|\mathcal{H}|/\delta)}{K}.$$

*The proof is the same as in Song et al. (2022) and thus is omitted here.*

# G GPT AS A JUDGE IN `TL;DR` TASK

> **System Prompt in TL;DR**
>
> Which of the following summaries does a better job of summarizing the most important points in the given forum post, without including unimportant or irrelevant details? Judge based on accuracy, coverage, and coherence.
> **Post:**
> <post>
> **Summary A:**
> <Summary A>
> **Summary B:**
> <Summary B>
> FIRST provide a one-sentence comparison of the two summaries, explaining which you prefer and why. SECOND, on a new line, state only "A"or "B"to indicate your choice. Your response should use the format:
> **Comparison:** <one-sentence comparison and explanation>
> **Preferred:** <"A"or "B">

# H EXTENDED EXPERIMENTAL DETAILS AND RESULTS

## H.1 TL;DR TASK

### H.1.1 MODEL SPECIFICATION

The following table lists the models and their corresponding links.

| Models | Links |
|---|---|
| EleutherAI/pythia-1b SFT | vwxyzjn/EleutherAI_pythia-1b-deduped__sft__tldr |
| EleutherAI/pythia-6.9b SFT | vwxyzjn/EleutherAI_pythia-6.9b-deduped__sft__tldr |
| EleutherAI/pythia-1b Reward | vwxyzjn/EleutherAI_pythia-1b-deduped__reward__tldr |
| EleutherAI/pythia-6.9b Reward | vwxyzjn/EleutherAI_pythia-6.9b-deduped__reward__tldr |

### H.1.2 COMPUTE RESOURCES SPECIFICATION

All 1B value models are trained on an A100 40G GPU. For the inference, 1B model inference takes place on one single A100 40G GPU, while 6.9b model on 2 A100 40G GPU.

### H.1.3 GENERATION PARAMETERS

We used fixed hyperparameters across all tested models. We use temperature $T = 0.7$, top-k $= 50$, and top-p $= 1.0$ with a maximum sequence length of 53 when sampling from the language model. For the beam search, we use $B = K = 4$ and $L = 16$ ($B$ is the beam width, $K$ is the child width, and $L$ is the chunk length). For the BoN, we use $N = 16$ for a fair comparison.

### H.1.4 IMPLEMENTATION

We utilize a public reward model trained from `Pythia-6.9b` with `TL;DR` dataset as the gold evaluator to measure the quality of generated summaries.

For the inference-time alignment baseline, such as BoN, ARGS, and CARDS, we use the 1b reward model[1] to provide the sequential-level or token-level guidance.

---
[1] vwxyzjn/EleutherAI_pythia-1b-deduped__reward__tldr

For the training-time baseline DPO, we use the public available checkpoints: the 1b model[2] and the 6.9b model[3].

For the IRO algorithm, we use $\beta_t = 1$ for all iterations. To obtain the value function, we initialize it from the 1B reward model. In each subsequent iteration, we initialize from the value function trained in the previous iteration. We train each value function with a learning rate $3 \times 10^{-6}$, batch size of 256, and for 10 epochs.

### H.1.5 ABLATION STUDY ON THE CHOICE OF $\beta_t$

In this subsection, we explore the choice of parameter $\beta_t$ in the TL;DR task, which controls the extent to which the updated policy $\hat{\pi}_{t+1}$ incorporates the estimated value function $\hat{V}^{\pi_t}$. Specifically, the update rule is given by

$$\log \hat{\pi}_{t+1}(a_h|s_h) \propto \log \hat{\pi}_t(a_h|s_h) + \hat{V}^{\hat{\pi}_t}(s_{h+1})/\beta_t,$$

where $s_{h+1} = [s_h, a_h]$, and smaller values of $\beta_t$ lead to more aggressive updates toward high-value actions under $\hat{V}^{\hat{\pi}_t}$.

In our implementation, we omit the term $\log \pi_t(a_h|s_h)$ when scoring the candidates, as it is already revealed during candidate generation. In the algorithm iteration, we set $\beta_1 = 1$ for the first iteration and perform ablations by varying $\beta_2$ for the second iteration to study its impact on performance.

As shown in Fig. 8, we find that setting $\beta_2 = 1$ yields the highest performance among the values we considered. When $\beta_2$ is increased, it places more weight on the prior policy by reducing the influence of $Q^{\pi_t}$.

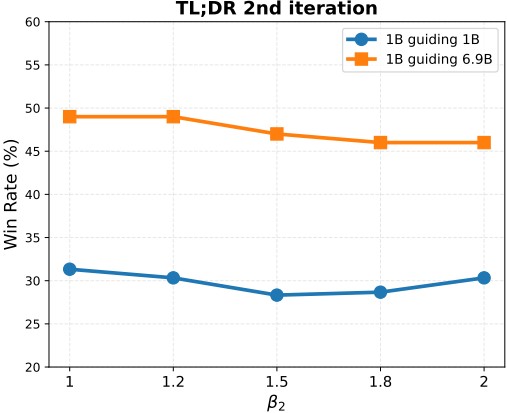

Figure 8: Ablation study on the choice of $\beta_2$ with fixed $\beta_1 = 1$. The results indicate that setting $\beta_2 = 1$ yields the highest win rate.

### H.2 ULTRAFEEDBACK TASK

### H.2.1 MODEL SPECIFICATION

The following table lists the models and their corresponding links.

---

[2]vwxyzjn/EleutherAI_pythia-1b-deduped__dpo__tldr
[3]vwxyzjn/EleutherAI_pythia-6.9b-deduped__dpo__tldr

| Models | Links |
|---|---|
| `Meta-Llama-3-8B-Instruct` | meta-llama/Meta-Llama-3-8B-Instruct |
| `Meta-Llama-3-70B-Instruct` | meta-llama/Meta-Llama-3-70B-Instruct |
| `UltraFeedback dataset` (Cui et al., 2023) | HuggingFaceH4/ultrafeedback_binarized |
| `sfairXC/FsfairX-LLaMA3-RM-v0.1` | sfairXC/FsfairX-LLaMA3-RM-v0.1 |
| `zephyr-7b-beta` | HuggingFaceH4/zephyr-7b-beta |
| `mistral-7b-sft-beta` | HuggingFaceH4/mistral-7b-sft-beta |

### H.2.2 COMPUTE RESOURCES SPECIFICATION

7B value models are trained on 8 A100 40G GPU. For the inference, the 7B model inference takes place on one 4 A100 40G GPU, while the 70b model takes place on one 4 H100 GPU.

### H.2.3 HYPERPARAMETERS SPECIFICATION

We used fixed hyperparameters across all tested models. We use temperature $T = 0.6$, top-k $= 50$, and top-p $= 0.9$ with a maximum sequence length of 2048 when sampling from the language model. For the beam search, we use $B = K = 4$ and $L = 16$ ($B$ is the beam width, $K$ is the child width, and $L$ is the chunk length). For the BoN, we use $N = 16$ for a fair comparison.

### H.2.4 IMPLEMENTATION

For the BoN with explicit reward, we trained the reward model initialized from `mistral-7b-sft-beta` using RLHFlow recipe (Dong et al., 2024) with a learning rate $5 \times 10^{-6}$ and for 3 epochs.

For the implicit reward, we use the relative log probability between `zephyr-7b-beta` and `mistral-7b-sft-beta` to calculate the implicit reward, where `zephyr-7b-beta` is fine-tuned based on the `mistral-7b-sft-beta` using DPO loss on the Ultrafeedback dataset.

For the token-level based method ARGS and CARDS, due to the long time generation for large maximum sequence length, we only consider ARGS due to the time limit. We use the 7b reward model trained by ourselves to provide token-level guidance.

We don't include methods such as GenARM (Xu et al., 2024b) and IVG (Liu et al., 2024) in our comparison, which also perform the inference time alignment, for the following reasons.

GenARM (Xu et al., 2024b) requires that the autoregressive reward model belong to the same model family as the base model. However, their released checkpoints are based on Llama-2, while our base model is Llama-3 family, which uses a different vocabulary size and tokenizer. In addition, in their experiment, GenARM underperforms compared to the BoN baseline when evaluated with GPT-4 under the Alpaca Evaluation benchmark.

For IVG (Liu et al., 2024), the provided value function is trained using the prompt-continuation evaluated by `fairXC/FsfairX-LLaMA3-RM-v0.1`, which leverages extensive pairwise data, making a direct comparison with our setup unfair. In addition, IVG's token-level implicit reward also tied to the same model family as the base model, limiting its capability.

To train the value function in the IRO algorithm, we initialize the first iteration from the 7B reward model. In each subsequent iteration, we initialize from the value function trained in the previous iteration. We train each value function with a learning rate $3 \times 10^{-6}$, batch size of 512, and for 10 epochs.

### H.2.5 ADDITIONAL RESULT

In this section, we present the numerical results on the Ultrafeedback task. According to both the length control win rate and the raw win rate metrics, IRO demonstrates consistent performance improvements across iterations, surpassing other baselines.

| Method | Compared Against GPT-4 | | Compared Against BoN-E | |
|---|---|---|---|---|
| | LC Win Rate | Win Rate | LC Win Rate | Win Rate |
| Meta-Llama-3-8B-Instruct | 30.71 | 29.63 | 38.77 | 38.32 |
| ARGS | 30.62 | 30.35 | 36.51 | 37.38 |
| BoN-I 16 | 35.25 | 32.67 | 43.52 | 40.68 |
| BoN-E 16 | 39.61 | 38.14 | 50.00 | 50.00 |
| weak to strong search | 36.20 | 35.90 | 44.13 | 42.92 |
| Controlled Decoding | 38.24 | 37.56 | 49.05 | 47.82 |
| IRO Iter1 | 41.06 | 37.32 | 55.42 | 53.60 |
| IRO Iter2 | 42.20 | 40.00 | 55.75 | 55.65 |
| IRO Iter3 | 43.11 | 41.00 | 57.00 | 56.46 |
| IRO Compressed | 42.59 | 39.63 | 58.92 | 56.92 |

Table 2: AlpacaEval 2 length-controlled win rate and raw win rate compared against GPT-4. We use $\beta_1 = 1, \beta_2 = 2, \beta_3 = 2.5$, which places more emphasis on the learned value function during generation. The decoding process maintains a beam width of 4 with 4 candidates preserved per state, and $l = 16$ tokens as a state. For fairness, we use $N = 4 * 4$ for BoN.

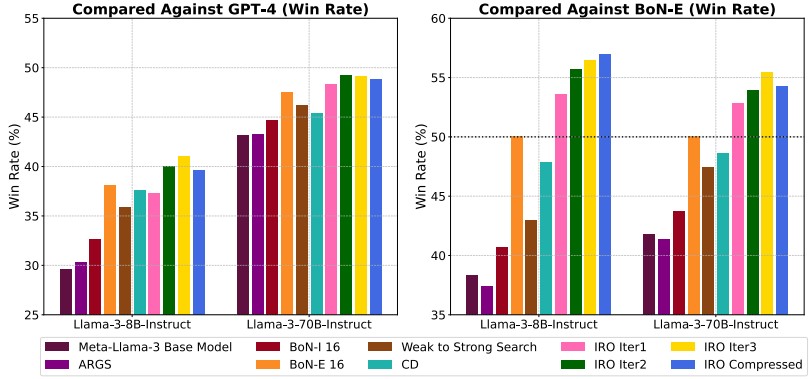

Figure 9: AlpacaEval 2 win rate compared against GPT-4 (left) and BoN-E (right) on both 7B and 70B base models, evaluated by GPT-4. We use $\beta_1 = 1, \beta_2 = 2, \beta_3 = 2.5$ while neglecting the $\log \pi(a|s)$, which places more emphasis on the learned value function during generation. The decoding process maintains a beam width of 4 with 4 candidates preserved per state, and $L = 16$ tokens as a state. For fairness, we use $N = 16$ for BoN.

| Method | Compared Against GPT-4 | | Compared Against BoN-E | |
|---|---|---|---|---|
| | LC Win Rate | Win Rate | LC Win Rate | Win Rate |
| Meta-Llama-3-70B-Instruct | 43.11 | 43.11 | 42.39 | 41.80 |
| ARGS | 43.02 | 43.21 | 42.01 | 41.37 |
| BoN-I 16 | 44.45 | 44.67 | 44.68 | 43.68 |
| BoN-E 16 | 47.45 | 47.58 | 50.00 | 50.00 |
| weak to strong search | 45.37 | 46.21 | 50.00 | 47.45 |
| Controlled Decoding | 46.25 | 45.33 | 49.90 | 48.60 |
| IRO Iter1 | 49.00 | 48.32 | 53.39 | 52.86 |
| IRO Iter2 | 49.75 | 49.19 | 55.69 | 53.91 |
| IRO Iter3 | 49.77 | 49.07 | 55.62 | 55.45 |
| IRO Compressed | 49.24 | 48.83 | 55.56 | 54.29 |

Table 3: AlpacaEval 2 length-controlled win rate and raw win rate compared against GPT-4 for 70B model. We use $\beta_1 = 1, \beta_2 = 2, \beta_3 = 2.5$. The decoding process maintains a beam width of 4 with 4 candidates preserved per state, and $l = 16$ tokens as a state. For fairness, we use $N = 4 * 4$ for BoN.

### H.2.6 ABLATION STUDY ON THE CHOICE OF $\beta_t$

In this subsection, we explore the choice of parameter $\beta_t$ in the Ultrafeedback task. Following the same experimental setup as in the TL;DR ablation, we fix $\beta_1 = 1$ for the first iteration and vary $\beta_2$ in the second iteration to analyze its effect on win rate performance. We observe that setting $\beta_2$ in the range of 1.5 to 2.0 yields the highest win rate, suggesting that this range achieves a favorable trade-off between incorporating guidance from the second-round value function and preserving information from the previous policy. These results indicate that neither overly aggressive ($\beta_2$ too small) nor overly conservative ($\beta_2$ too large) updates are optimal for effective iterative alignment.

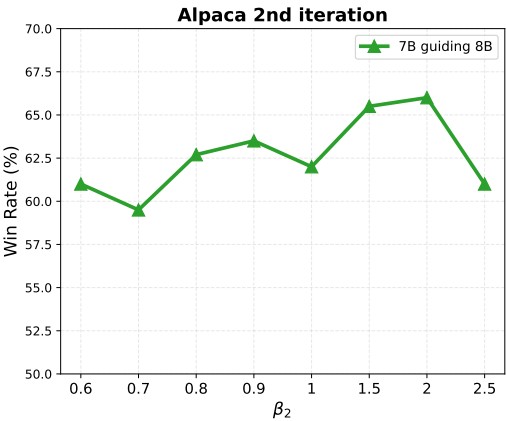

Figure 10: Ablation study on the choice of $\beta_2$ with fixed $\beta_1 = 1$. The win rate is judged by GPT-4 compared against the BoN16 on the Alpaca subset. The results indicate that setting $\beta_2 = 2$ yields the highest win rate, demonstrating the sensitivity of performance to the value of $\beta_2$.

### H.3 TEST-TIME SCALING WITH VALUE FUNCTION

Here we explore how the performance of IRO scales with the search budget $N$ across different iterations. As shown in Figure 11, the reward score evaluated using the gold reward model consistently improves with larger search budgets $N$, indicating that the quality of outputs generated by IRO increases accordingly.

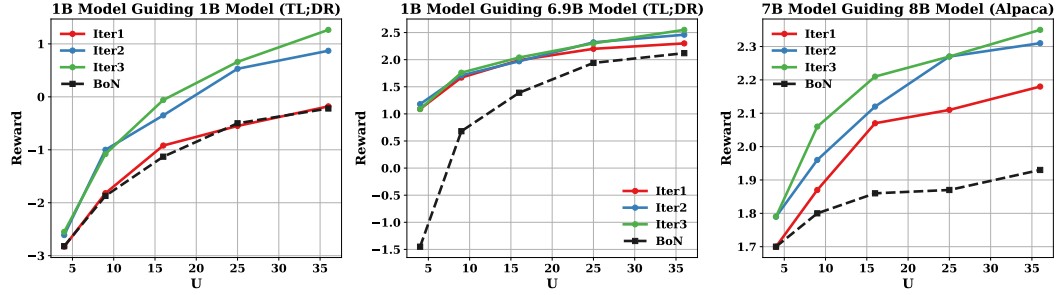

Figure 11: The reward scored by 6.9b gold reward model for `TL;DR` task and Alpaca subset improves with the compute budget under different iterations and BoN. During the search, we set the parameters $K = B = \sqrt{N}$.

### H.4 EXTENSION TO VERIFIABLE REWARD SETTING

In this section, we extended our algorithm to the verifiable reward setting, math reasoning tasks under binary reward. This demonstrates that IRO generalizes beyond continuous reward setting, such as summarization and instruction-following task.

More specifically, for verifiable rewards, we train the value function using the binary cross-entropy (BCE) loss function instead of the regression loss mentioned in (7):

$$\hat{V}^{\hat{\pi}_{t-1}} := \arg\min_V -\mathbb{E}_{\tau \sim \mathcal{D}_t}\left[\sum_{h=1}^{H} r(\tau)\log V(s_h) + (1 - r(\tau))\log(1 - V(s_h))\right], \qquad (49)$$

where $r(\tau) \in \{0, 1\}$ denotes the verifiable reward, which is computed as the exact match with the ground truth label, and $V(s_h)$ denotes the prediction for state $s_h$.

During generation, we use the system prompt "You are a helpful assistant, and put your final answer within boxed{}." to extract the final answer. A candidate is considered correct if the boxed answer matches the ground truth.

In our experiment, we use the `Qwen-3B` as our frozen policy model, and 1.5B model as our value function. We evaluate the performance on a random 500 samples from the GSM8k test set (total 1318 samples) and Math500, which is a subset sampled from the dataset MATH (total 5000 samples).

We consider the following baselines: (1) majority voting, (2) train-based RL algorithm, Group Relative Policy Optimization (GRPO, (Shao et al., 2024)), and (3) Q# (Zhou et al., 2025), which trains a token-level value function to guide generation. To ensure a fair comparison, we fix the frozen base policy as `Qwen-3B`, and we only use the GSM8K train set for all baselines.

To implement the IRO, we generate 16 candidate completions with a temperature of 0.7 and a maximum number of new tokens of 512 for each problem. Then, we filter out all correct and incorrect questions to construct the dataset to train value functions.

| Method | GSM8K (%) | Math500 (%) |
|---|---|---|
| Qwen-3B | 44.80 | 21.00 |
| Majority Voting@16 | 46.80 | 26.80 |
| Q# | 73.60 | 30.60 |
| GRPO | 79.80 | 31.80 |
| IRO Iter1 | 79.61 | 31.50 |
| IRO Iter2 | 81.27 | 33.40 |

Table 4: Comparison between baseline and IRO on math reasoning task. Here, the base model is Qwen-3B. IRO operates with $K = 4, B = 4, L = 8$, it selects the final answer as the most frequently one in the boxed answers.

From Table 4, we observe that IRO demonstrates substantial improvements on math reasoning under binary reward settings, outperforming naive majority voting by over 30% absolute on GSM8K. In addition, IRO's performance is very close to that of the RL-based method, GRPO.

These results confirm that IRO is effective not only in continuous reward setting (e.g., summarization and instruction-following), but also in binary reward settings such as math reasoning task.

## H.5 ABLATION STUDY

In this subsection, we present ablation studies on the effect of chunk length $L$ used during guided generation, and the size of the dataset used to train the value function, on the performance of IRO in its first iteration.

**Explore different sampling strategies for IRO.** We compare (1) beam search (Zhou et al., 2024) with and without the *diversity-first* principle, and (2) stochastic sampling methods with different temperatures. For stochastic sampling, text chunks are sampled from a softmax distribution over value scores, i.e., $a \sim \frac{\exp(V_i/T_{\text{temp}})}{\sum_j \exp V_j/T_{\text{temp}}}$. The results are shown in Fig. 12

**Explore the influence of value function size in IRO** To study the effect of value function size, we fix the policy and reward model while varying the value function. More specifically, on TL;DR dataset, we fix 6.9B policy and 1B reward model, and compare 1B vs. 2.8B value function sizes. On

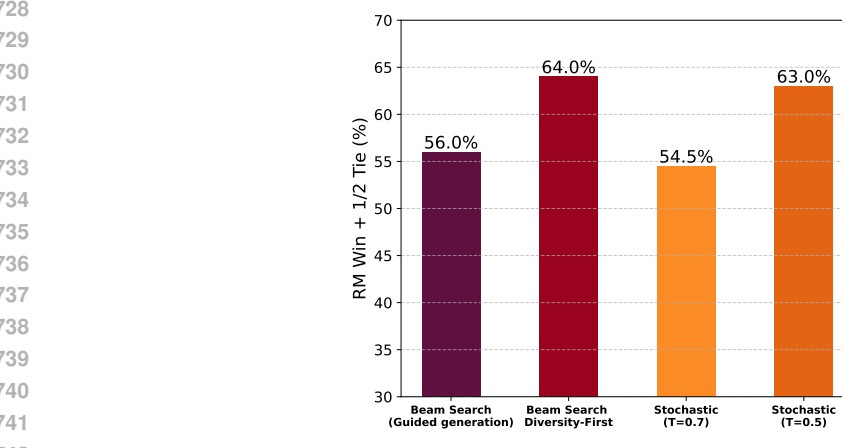

Figure 12: Beam search with the *diversity-first* principle achieves the highest win rate against the BoN baseline under various sampling strategies during the guided generation step on the 200-example subset.

Ultrafeedback dataset, we fix 8B policy and 7B reward model, and compare 2B vs. 7B value function sizes.

From Table 5 and Table 6, it is observed that larger value models (e.g., 2.8B) generally learn better and offer better guidance. Smaller ones (e.g., 2B on Ultrafeedback) may even hurt performance in some cases, even worse than naive Best-of-N.

| Value Size | IRO Iter1 | IRO Iter2 | IRO Iter3 |
|---|---|---|---|
| 2.8B | 36.55% | 50.11% | 51.78% |
| 1B | 37.66% | 48.00% | 49.67% |

Table 5: Win rate on TL;DR task — Effect of value function size on IRO for a 6.9B policy with different value function sizes (2.8B vs 1B).

| Value Size | GPT-4 LC Win Rate | BoN LC Win Rate |
|---|---|---|
| BoN 16 | 38.41 | 50.00 |
| 2B | 37.37 | 38.07 |
| 7B | 43.71 | 55.63 |

Table 6: Instruct-Following task - The effect of the value function size on the 1st iteration of IRO for 8B fixed policy compared against GPT-4 and BoN with $N = 16$ on a 200-sample sub-dataset.

**Explore IRO's robustness under imperfect reward models**  Since IRO involves using a reward model to evaluate the guided sampled data in each iteration, here we consider IRO's robustness under an imperfect reward model on both the summarization task and the instruction following task by using different reward models of varying quality.

For the summarization task, we used three different reward models with different qualities. Here, we assume the quality of the reward model is reflected by the accuracy on two held-out test datasets, which consist of 83.8k and 2.8k pair-wise samples, respectively, shown in Table 7.

We ran IRO using 1B value functions to guide 1B and 6.9B policy models under each reward model, which is used to label and score each prompt-continuation pair during the value function training phase and select the final answer during the guided decoding phase.

From Table 8 and Table 9, despite RM 1 being weaker, IRO consistently shows performance gains over iterations. In addition, a better reward model (RM 2 or RM 3) leads to stronger improvements.

| Reward Model (RM) | Test Set 1 | Test Set 2 |
|---|---|---|
| RM 1 | 65.81% | 62.88% |
| RM 2 | 68.56% | 66.93% |
| RM 3 | 68.74% | 68.82% |

Table 7: Classification accuracies on two held-out test sets for three different reward models.

| IRO Iteration | RM1 Win Rate | RM2 Win Rate | RM3 Win Rate |
|---|---|---|---|
| IRO Iter1 | 26.00% | 28.11% | 28.11% |
| IRO Iter2 | 31.33% | 37.55% | 36.00% |
| IRO Iter3 | 35.33% | 30.33% | 37.22% |

Table 8: IRO with 1B value functions guiding a 1B policy on the TL;DR dataset with different reward models. Reward scores are evaluated by a 6.9B reward model, and win rates are judged by GPT-4o against reference summaries.

| IRO Iteration | RM1 Win Rate | RM2 Win Rate | RM3 Win Rate |
|---|---|---|---|
| IRO Iter1 | 37.66% | 44.55% | 40.56% |
| IRO Iter2 | 48.00% | 56.33% | 55.00% |
| IRO Iter3 | 49.67% | 50.44% | 51.11% |

Table 9: IRO with 1B value functions guiding a 6.9B policy on the TL;DR dataset with different reward models. Reward scores are evaluated by a 6.9B reward model, and win rates are judged by GPT-4o against reference summaries.

On the Ultrafeedback dataset, we consider using a 7B value function to guide the 8B frozen policy model `LLama-3-8B-Instruct`. We tested IRO using two reward models, one trained only on the Ultrafeedback dataset and another one is a public reward model `sfairXC/FsfairX-LLaMA3-RM-v0.1`. The quality of the two reward models is measured on RewardBench, which is shown in Table 10 below.

| Model | Chat | ChatHard | Safety | Reasoning | Avg |
|---|---|---|---|---|---|
| RewardModel | 95.25% | 58.11% | 67.70% | 68.26% | 72.33% |
| sfairXC/FsfairX-LLaMA3-RM-v0.1 | 99.16% | 64.91% | 86.62% | 87.16% | 80.44% |

Table 10: RewardBench evaluations on two reward models.

Here we only show IRO-Iter 1. We measure the performance on a 200-sample Alpaca sub-dataset, which is compared against BoN-16 and GPT-4, all of which are judged by GPT-4.

| Reward Model | GPT-4 LC Win Rate | GPT-4 Raw Win Rate | BoN LC Win Rate | BoN Raw Win Rate |
|---|---|---|---|---|
| Reward Model | 43.71 | 40.75 | 55.63 | 53.92 |
| sfairXC/FsfairX-LLaMA3-RM-v0.1 | 47.35 | 45.25 | 54.17 | 55.17 |

Table 11: IRO Iter1 performance on a 200-sample Alpaca sub-dataset.

Overall, it is observed that by using an imperfect reward model, IRO can still achieve robust behavior and improvements. Moreover, a more accurate reward model will lead to better performance for IRO.

**Explore different chunk lengths for IRO.**  We explore the effect of the chunk length $L$ on the performance of IRO compared against the BoN method with $N = 16$ in the first iteration. As shown in Figure 13, performance first improves with longer chunks but then declines towards BoN as $L$ approaches the full sequence length. Intuitively, when the chunk is too short, the value function is insufficient to distinguish the quality of different candidates. As chunk length increases, the value function can make more accurate judgments between different candidates. However, when the chunk

length approaches the full sequence, the algorithm gradually converges to the BoN method, causing the win rate relative to BoN to approach $50\%$. This confirms that intermediate chunk lengths best balance value-based control and sequence-level coherence.

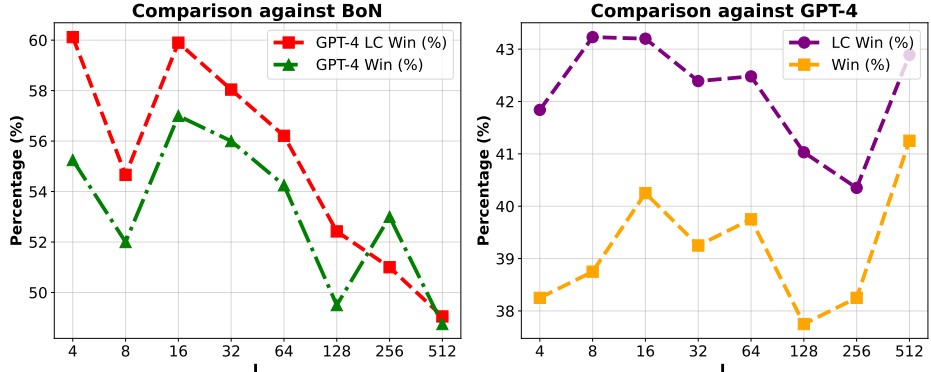

Figure 13: The effect of the chunk length on the 1st iteration of IRO compared against BoN and GPT-4 with $N = 16$ on a 200-sample sub-dataset, judged by GPT-4. Left:The comparison against BoN. Right: The comparison against GPT-4.

**Explore different data sizes on training a value function for IRO.** Table 12 shows the impact of data size for training a value function on the performance under different chunk length settings. When the data size is small (e.g., 1024 or 2048), larger chunk lengths (L = 32, 64) yield better results. This is likely because the value function trained on limited data lacks the ability to provide accurate estimates for smaller token sequences. In contrast, as data size increases (e.g., 4096 or 8192), smaller chunks (L = 16) perform better, suggesting that finer-grained search leads to a more accurate policy update at inference time.

| Chunk Length | Data Size | GPT-4 Evaluation (%) | |
| --- | --- | --- | --- |
| | | LC Win | Win |
| 16 | 1024 | 48.61% | 46.66% |
| | 2048 | 51.26% | 49.68% |
| | 4096 | 56.45% | 53.25% |
| | 8192 | 59.9% | 57% |
| 32 | 1024 | 48.52% | 45.51% |
| | 2048 | 50.92% | 51.85% |
| | 4096 | 48.96% | 48.75% |
| | 8192 | 58.04% | 56% |
| 64 | 1024 | 53.64% | 50.24% |
| | 2048 | 54.63% | 54.5% |
| | 4096 | 53.52% | 51.73% |
| | 8192 | 56.21% | 54.25% |

Table 12: GPT-4 evaluation results on IRO-Iter1 vs. BoN-16, across different chunk lengths and value function training data sizes.

**Explore different value function training.** In this section, we explore several approaches to value function training, including temporal-difference (TD) learning, Generalized Advantage Estimation (GAE; Schulman et al. (2015)), and the FUDGE loss (Yang & Klein, 2021).

Let $V_\phi$ denote the value function parameterized by $\phi$, and $\tau$ be trajectory. We summarize the loss functions for the three methods below.

**TD Learning** TD learning provides a low-variance method for the value function learning. The TD objective minimizes the squared temporal-difference error across a trajectory:

$$\hat{V} := \arg\min_V \ \mathbb{E}_{\tau \sim \mathcal{D}_t}\left[\sum_{h=1}^{H}\left[r(s_h, a_h) + \gamma V_\phi(s_{h+1}) - V_\phi(s_h)\right]^2\right]. \tag{50}$$

**GAE** GAE computes a smoothed advantage estimate that interpolates between Monte Carlo returns and TD errors, trading off bias and variance via the parameter, which is widely used in PPO algorithm.

For each timestep $t$, the one-step TD error is:

$$\delta_t = r_t + \gamma V_\phi(s_{t+1}) - V_\phi(s_t).$$

The GAE advantage is computed recursively backward through the trajectory:

$$\hat{A}_t = \delta_t + \gamma\lambda\hat{A}_{t+1}, \qquad t = T - 1, \ldots, 0.$$

GAE then constructs a target value for each state:

$$\hat{V}_t^{\text{target}} = \hat{A}_t + V_\phi(s_t).$$

which is used as a regression target for value function training. Below, we empirically show that FUDGE loss exhibits better performance for training the token-level value function.

| Method | Reward (1B guiding 1B) | Win Rate (1B guiding 1B) (%) | Reward (1B guiding 6.9B) | Win Rate (1B guiding 6.9B) (%) |
|---|---|---|---|---|
| SFT | -4.52 | 6.00 | -1.33 | 16.33 |
| TD $\lambda = 0.99$ | -2.01 | 17.67 | 1.73 | 23.67 |
| GAE $\gamma = 1, \lambda = 1$ | -1.05 | 23.33 | 1.93 | 36.67 |
| GAE $\gamma = 0.99, \lambda = 1$ | -1.01 | 21.00 | 1.95 | 37.67 |
| GAE $\gamma = 0.98, \lambda = 1$ | -1.00 | 20.33 | 1.92 | 37.00 |
| GAE $\gamma = 0.97, \lambda = 1$ | -1.01 | 21.33 | 1.93 | 33.67 |
| GAE $\gamma = 0.95, \lambda = 1$ | -0.76 | 16.67 | 1.93 | 33.67 |
| IRO Iter1 | -0.91 | 26.05 | 1.98 | 37.67 |

Table 13: Comparison between using FUDGE, TD error, and GAE for value function training on the first iteration of IRO. Reward scores are evaluated by a 6.9B reward model, and win rates against the reference summary are evaluated by GPT-4o-mini on 300 samples from the test dataset.

| Reward Model | GPT-4 LC Win Rate | GPT-4 Raw Win Rate | BoN LC Win Rate | BoN Raw Win Rate |
|---|---|---|---|---|
| IRO Iter1 | 43.71 | 40.75 | 55.63 | 53.92 |
| GAE $\gamma = 0.99, \lambda = 1$ | 47.35 | 45.25 | 54.17 | 55.17 |

Table 14: IRO Iter1 performance on a 200-sample Alpaca sub-dataset.

# I    CASE STUDY

Here, we display several instances from the TL;DR dataset and the Alpaca evaluation dataset. In these cases, the response generated by our algorithm achieves successive improvement.

---

### Example 1 in the TL;DR Taks

**Question** SUBREDDIT: r/loseit\n\nTITLE: Was doing great (lost 20 pounds), now worse than ever and feel like I can't stop eating (30/f)\n\nPOST: Hello! As the title describes, I had been losing weight (or at least maintaining) up until about two months ago. I'm now afraid to look at the scale, but I have NOTICEABLY gained weight. Some of my clothes don't fit me anymore. Formerly: 5'7", 160 lbs.\n\nI've been a subscriber of this site for a long time and totally get that CICO is really the only way to lose weight, that you shouldn't weigh yourself all that often, etc. That was never a problem before. My diet was not incredibly strict to begin with (1 lb/week goal on MFP). I now feel like I'm straight up addicted to food. I am eating in volumes I've NEVER eaten before. Nothing about my life has changed beyond my eating habits.\n\nI have OCD (worried I'm compulsively

eating, if that's even a thing) and bipolar 1 (yes, I am medicated; no, not with the stuff that causes weight gain – quite the opposite). I should note that I don't really feel depressed. My life is pretty decent right now! Also, I've never been a "depression eater" anyway. Oh, also, 110% sure I'm not pregnant.\n\nHow does one suppress this kind of appetite? Lately it seems that if I'm not eating, I am obsessively thinking about eating and believing I'm hungry. I can't possibly be hungry! What can I do to get back to eating a normal, reasonable amount that would result in gradual weight loss (last I checked, my TDEE was something like 2,000 but I've gained weight since then...)? I have NEVER been an overeater before; any weight gain in the past has been due to eating a moderate amount of complete shit, not eating LOTS of complete shit. I don't even know how to go about losing weight when my issue is the volume I'm eating and not the nutrition (or lack thereof).\n\nTL;DR:

**Reference**
Developed an issue with overeating and constantly feeling hungry. This has never been the case with me before, so I have no clue as to how to attack weight loss from a volume standpoint and not just a nutrition standpoint.
**Reward Score** 3.08

**1B Base**
Eating 2-4 meals a day, having gained weight and feeling like I cant stop eating without reaching the point of insanity, and feeling depressed and OCD.\n\n*How to get my weight back where it needs to be?\n\n*How to
**Reward Score** -4.91

**1B BoN 16**
I'm stuck with an abnormally high volume of food even though I'm in control of my eating behavior. Help? #f5\n\n\n\nTL;DR: My eating behavior, though low volume, high variability, and obsessive (
**Reward Score** -0.36

**1B ARGS**
I'm a "depression eater" (depressed/bipolar 1/OCD/depressed/eating compulsively/depressed/eating compulsively/depressed/eating compulsively/depressed/eating compulsively/b
**Reward Score** -1.89

**1B CARDS**
I'm obsessively thinking about eating, I have no appetite, I'm getting fatter. How can I stop?\n\nTL;DR: Obsessively thinking about eating, I have no appetite, I'm getting fatter. How can I stop?
**Reward Score** 0.42

**1B Guiding 1B Iter1**
Lost 20 pounds, gained 20 pounds, feel like I can't stop eating and my life has become miserable, don't know how to get back to eating a normal, reasonable amount of food.\n\nTL;DR: Lost 20 pounds, gained 20
**Reward Score** -3.28

**1B Guiding 1B Iter2**
Lost 20 pounds, gained 20 pounds, gained 10 pounds the past two months and feel like I'm addicted to food and can't control it... \n\n\nGot the title, thanks for your input! \n\nTL
**Reward Score** -1.10

**1B Guiding 1B Iter3**

I've gained weight and am worried I'm addicted to food. I've never been an overeater and want to lose weight, and need advice on how to get back to eating like I used to, not obsessively thinking about eating.\n\n

**Reward Score** 1.64

---

**6.9B Base**

Eating 2,000 calories a day without any exercise, and feeling like I am eating more than I ever have before. Want to lose weight, but I'm not sure how and I'm not sure how to cope. \nI also have a reddit

**Reward Score** -4.88

---

**6.9B BoN 16**

Was losing weight. Loseit helped, but now, after years of being an underweight girl on a steady diet (even when I was working out/eating well!), can't stop eating. I may be an addict...

**Reward Score** 2.30

---

**6.9B ARGS**

Gained 30 lbs, now I'm 30 lbs heavier and feel like I can't stop eating (lots of complete shit). I'm not depressed, just a compulsive eater..

**Reward Score** 0.17

---

**6.9B CARDS**

I'm 30/f and ate a lot, and now gained weight. Now I want to lose weight, but am afraid I'll never be able to eat again...

**Reward Score** -0.89

---

**1B Guiding 6.9B Iter1**

Was doing great. Gained weight, worse than I've ever been, and I can't stop eating. How do I go about eating less to drop the weight?.....

**Reward Score** 2.83

---

**1B Guiding 6.9B Iter2**

Was doing great. Gained weight, feel like I can't stop eating. How do I regain control of my eating?...........................

**Reward Score** 3.20

---

**1B Guiding 6.9B Iter3**

Was doing well losing weight, now in worse shape than ever and feel the need to eat even more. How can I get back to eating a moderate amount of healthy food and losing weight?\n

**Reward Score** 4.44

---

Example 2 in the TL;DR Taks

**Question** SUBREDDIT: r/dating_advice\n\nTITLE: Not even sure if this is the right place for this... but I think I might be confused about my sexuality? [24F]\n\nPOST: I was raised Catholic, and even though I'm not religious anymore, I definitely still deal

with the Catholic guilt. I also think I have body dysmorphic disorder, as I feel extremely self-conscious about my body (even though my ex assured me that I have nothing to be ashamed of, that I'm very sexy, etc.)\n\nOk so here's the part I'm confused about: guys that I think are pretty cute and interesting to talk to, whenever it gets to the point where I should probably kiss them or make some other kind of physical move, I shut down and do nothing. In that moment I find myself doubting whether or not I am actually attracted to them or something? It's like the anxiety of taking it to the next level overrides all the other emotions. \n\nAlso it's this weird thing where I've never really learned to "flirt" and I think that a lot of my interactions with guys tend to skew towards friendliness rather than... flirting or whatever? So then that's like an added layer of awkwardness where there's this platonic exchange going on but we're supposed to suddenly get "sexy" or whatever. \n\nI didn't have a huge sex drive with my ex. He pretty much wanted it every day, whereas I was closer to once every week or two. At the beginning of our relationship I wanted it more often. We had pretty great sex overall, at least when I was sincerely in the mood. \n\nDoes any of this seem to indicate that I'm not heterosexual?\n\nTL;DR:

---

**Reference**

I find myself having sort of platonic relationships with men even when I find them attractive. Hard to initiate the sexual stuff, not sure if it's insecurity/Catholic guilt or cuz I'm not straight.

**Reward Score** | 2.45 |

---

**1B Base**

This whole "sexuality" discussion makes me so very uncomfortable! What do? It's really driving me nuts if so...?\n\n*Sorry if something doesn't make sense*\n\n*Why would I think about sex, or how would that be

**Reward Score** | -5.53 |

---

**1B BoN 16**

Not sure if I'm just asexual or if I'm in denial about being attracted to other people. When I flirt with guys I kind of shut down because I have anxiety that they are somehow into me or it's like they aren't my real friends and

**Reward Score** | -3.56 |

---

**1B ARGS**

I'm not sure if I'm a lesbian or bisexual, and I'm not sure if I'm attracted to guys.\n\n*sigh*\n\nTL;DR: I'm not sure if I'm a lesbian or bisexual, and

**Reward Score** | -5.47 |

---

**1B CARDS**

I don't know how to deal with my body image issues or body dysmorphic disorder. I'm not sure if I'm straight, and I don't want to be.\n\nTL;DR: I don't know how to deal with my body image issues or body dysmorphic disorder. I'm not sure if I'm straight, and I don't want to be.

**Reward Score** | -2.23 |

---

**1B Guiding 1B Iter1**

Was raised Catholic, have body dysmorphic disorder, am confused as to whether or not I am attracted to guys, and if so, how to deal with the issue.\n\nocalypse!\n\nTL;DR: I was raised Catholic,

**Reward Score** | -3.65 |

---

**1B Guiding 1B Iter2**

I find myself doubting whether or not I'm attracted to guys that I think are "pretty cute" or "interesting to talk to" whenever it gets to the point where I should kiss them or make some other kind of physical move.\n\n

**Reward Score** | 0.98 |

**1B Guiding 1B Iter3**

I find myself doubting whether or not I'm sexually attracted to guys who I think are cute and interesting. I don't know if I'm just being paranoid and insecure, or if my body dysmorphic disorder is holding me back or something.\n

**Reward Score** 2.83

**6.9B Base**

Am a very self conscious person. It's so bad that if men want to talk to me in a non-friendlier environment I shut down and do nothing... or feel like I might be attracted to them? ——————————————————————————————————————————
—————————————————— ——————————————————————————————————————
——————————————————————————————————

**Reward Score** -3.52

**6.9B BoN 16**

not sure if I'm bi or gay, always feel anxious about hooking up, I have no idea how/if I should bring it up to anyone I might be interested in.

**Reward Score** 1.36

**6.9B ARGS**

Guys I'm interested in tend to be "cute" and "cuddly" and I'm not sure if I'm actually attracted to them.

**Reward Score** -2.95

**6.9B CARDS**

Guys I think are cute and interesting to talk to, and I want to kiss them but then I can't do it.

**Reward Score** -1.66

**1B Guiding 6.9B Iter1**

Catholic guilt, body dysmorphic, and self-consciousness lead to me not even trying to initiate anything with a guy I find cute and interesting.

**Reward Score** 1.27

**1B Guiding 6.9B Iter2**

I have body dysmorphia and am pretty self-conscious about my body. I find myself doing certain actions in situations where I think I should be flirting but am not, and I'm wondering if this is a sign that I'm gay?

**Reward Score** 1.39

**1B Guiding 6.9B Iter3**

I feel like a lot of my interactions with guys are very friendly and not at all sexual. I have body dysmorphic disorder and I'm not sure if I'm actually attracted to guys, and am also not sure if I'm bisexual or something.

**Reward Score** 2.20

### Example 1 in the Instruct Following Task

**Question** My friend's dog just died and they're really sad. How do I comfort them?

---

**8B Base**

I'm so sorry to hear that your friend is going through a tough time. Losing a pet is never easy. Here are some ways you can comfort your friend:\n\n1. Listen: Allow your friend to express their emotions and thoughts without interrupting or judging. Sometimes, people just need someone to listen to them without offering solutions.\n2. Validate their feelings: Let your friend know that their feelings are normal and that it's okay to feel sad, angry, or overwhelmed. You can say something like, "I'm so sorry to hear about the loss of your furry friend. It's completely normal to feel sad and grieving."\n3. Offer physical comfort: A hug, a pat on the back, or a gentle touch on the arm can be comforting. However, make sure to respect your friend's boundaries if they don't want physical contact.\n4. Share a memory: If you have a fond memory of the dog, share it with your friend. This can help them feel less alone in their grief and bring a smile to their face.\n5. Avoid clichés: Phrases like "It was just a dog" or "You can always get another one" can come across as insensitive. Instead, focus on the emotional bond your friend had with their pet.\n6. Help with practical tasks: Offer to help your friend with tasks like cleaning, cooking, or running errands. This can take some burden off their plate and give them more time to focus on their grief.\n7. Respect their boundaries: Everyone grieves differently, and some people may need more time and space than others. Respect your friend's boundaries and don't push them to talk or do things they're not ready for.\n8. Encourage self-care: Suggest healthy coping mechanisms like exercise, meditation, or spending time in nature. Encourage your friend to take care of themselves during this difficult time.\n9. Offer to go with them: If your friend wants to visit the dog's grave, go with them. Having someone to accompany them can make the experience less overwhelming.\n10. Be patient: Grief is a process, and it's okay if your friend takes time to work through their emotions. Be patient and understanding, and let them know that you're there for them whenever they need you.\n\nSome additional tips:\n\n* Avoid saying "It's just a dog" or "You can always get another one." These phrases can come across as insensitive and dismissive of your friend's feelings.\n* Don't try to "fix" the situation or offer a solution. Grief is a process, and there's no quick fix.\n* Avoid comparing your friend's grief to that of someone who has lost a human loved one. Each person's grief is unique, and it's important to respect that.\n* Consider sending a card, flowers, or a memorial gift to show your support and condolences.\n\nRemember, the most important thing you can do is be present and supportive for your friend during this difficult time.

**Reward Score** 3.22

---

**8B BoN-E 16**

I'm so sorry to hear that your friend is going through a tough time. Losing a beloved pet is never easy. Here are some ways you can comfort your friend:\n\n1. Listen and validate their feelings: Allow your friend to express their emotions without judgment. Let them know that their feelings are normal and that it's okay to feel sad, angry, or overwhelmed.\n2. Offer physical comfort: A hug, a pat on the back, or a gentle touch on the arm can be comforting. Physical contact can help release oxytocin, the "cuddle hormone," which can reduce stress and anxiety.\n3. Share fond memories: Ask your friend to share stories about their dog, and listen attentively. Share your own fond memories of the dog, if you have any. This can help your friend feel less alone and more connected to their pet.\n4. Avoid clichés: Phrases like "It was just a dog" or "They're in a better place" can come across as insensitive. Instead, focus on the dog's importance in your friend's life and acknowledge the depth of their grief.\n5. Respect their boundaries: Everyone grieves differently, and some people may not want to talk about their feelings or share memories. Respect your friend's boundaries and give them space if that's what they need.\n6. Offer practical help: Help your friend with tasks such as cleaning, laundry, or errands. This can take some burden off their plate and give them more time to focus on their grief.\n7. Suggest a memorial or ritual: If your friend is comfortable with it, suggest doing something to honor their dog's memory, such as planting a tree, making a donation to an animal welfare organization, or creating a memory box.\n8. Be patient:

Grief is a process, and it may take time for your friend to work through their emotions. Be patient and understanding, and let them know that you're there for them whenever they need you.\n9. Encourage self-care: Losing a pet can be physically and emotionally exhausting. Encourage your friend to take care of themselves by getting enough sleep, eating well, and engaging in activities that bring them comfort and joy.\n10. Remind them of their strength: Losing a pet can be a significant blow, but it's also a testament to your friend's capacity for love and compassion. Let them know that they're strong and capable of getting through this difficult time.\n\nRemember, there's no one-size-fits-all approach to comforting someone who's lost a pet. Be genuine, empathetic, and patient, and let your friend know that you're there to support them.

**Reward Score** 4.25

---

**Weak to Strong Search (7B Guiding 8B)**
I'm so sorry to hear that your friend is going through this. Losing a beloved pet can be a devastating experience. Here are some ways you can comfort your friend:\n\n1. Listen and validate their feelings: Allow your friend to express their emotions without judgment. Let them know that their feelings are normal and that it's okay to grieve the loss of their pet.\n2. Be present: Sometimes, physical presence can be comforting. Sit with your friend, hold their hand, or give them a hug. Let them know that you're there for them.\n3. Share fond memories: Share stories and memories of the dog with your friend. This can help them feel better and remember the good times they had with their pet.\n4. Avoid clichés: Phrases like "It was just a dog" or "They're in a better place" can come across as insensitive. Instead, focus on acknowledging your friend's feelings and the importance of their pet in their life.\n5. Offer practical help: Help your friend with tasks such as cleaning up after the pet, taking care of other pets in the household, or running errands. This can take some of the burden off their plate and give them more time to grieve.\n6. Respect their boundaries: Everyone grieves differently, and some people may need more time and space than others. Respect your friend's boundaries and don't push them to talk or share more than they're comfortable with.\n7. Encourage self-care: Losing a pet can be physically and emotionally draining. Encourage your friend to take care of themselves by getting enough rest, eating well, and engaging in activities that bring them comfort and relaxation.\n8. Consider a memorial or ritual: Some people find it helpful to create a memorial or ritual to honor their pet. This could be something as simple as lighting a candle, planting a tree, or creating a memory book.\n9. Be patient: Grief is a process, and it can take time for your friend to come to terms with their loss. Be patient and understanding, and let them know that you're there for them in the long haul.\n10. Remind them that they're not alone: Losing a pet can be a isolating experience, but it's important for your friend to know that they're not alone. Remind them that you're there for them, and that many people have gone through a similar experience.\n\nRemember, everyone grieves differently, and there's no one-size-fits-all approach to comforting your friend. The most important thing you can do is be present, supportive, and understanding.

**Reward Score** 3.65

---

**7B Guiding 8B Iter1**
I'm so sorry to hear that your friend is going through this tough time. Losing a beloved pet is never easy. Here are some ways you can comfort your friend:\n\n1. Listen: Allow your friend to express their emotions and thoughts without interrupting or judging. Listen actively and show empathy through your body language and tone of voice.\n2. Validate their feelings: Acknowledge that their feelings are normal and valid. Let them know that it's okay to feel sad, angry, or overwhelmed.\n3. Share a memory: Share a fond memory of their dog, if you have one. This can help your friend feel comforted and reminded of the good times they shared with their pet.\n4. Offer physical comfort: Sometimes, physical touch can be comforting. Offer a hug, hold their hand, or give them a gentle pat on the back.\n5. Help with practical tasks: Offer to help with tasks such as cleaning, cooking, or running errands. This can help take some of the burden off your friend and give them more time to grieve.\n6. Respect their boundaries: Everyone grieves differently, and it's essential to respect your friend's boundaries. If they don't want to talk or don't feel like being around

people, give them space.\n7. Encourage self-care: Suggest healthy coping mechanisms like exercise, meditation, or spending time in nature. Encourage your friend to take care of themselves physically and emotionally.\n8. Be patient: Grieving is a process that takes time. Be patient and understanding, and avoid putting pressure on your friend to "get over it" or "move on."\n9. Offer a memorial idea: Suggest a memorial idea, such as planting a tree, creating a memory box, or making a donation to an animal welfare organization in their dog's name.\n10. Check in regularly: Check in with your friend regularly to show that you're thinking of them and care about their well-being.\n\nSome specific phrases you could use to comfort your friend include:\n\n* "I'm so sorry to hear about the loss of your dog. My heart goes out to you."\n* "You're not alone in your grief. I'm here for you, and I'll support you through this tough time."\n* "Remember all the good times you had with your dog? Those memories will always be with you."\n* "It's okay to feel sad, angry, or overwhelmed. Those are all normal emotions when we lose someone we love."\n\nRemember to be genuine and sincere in your condolences, and your friend will appreciate your support during this difficult time.

**Reward Score** 3.89

---

### 7B Guiding 8B Iter2

I'm so sorry to hear that your friend is going through a tough time. Losing a beloved pet can be incredibly painful. Here are some ways you can comfort your friend:\n\n1. Listen actively: Allow your friend to express their emotions and thoughts without interrupting or judging. Listen attentively to what they're saying, and show empathy through your words and body language.\n2. Validate their feelings: Acknowledge the pain and sadness your friend is feeling. Let them know that it's okay to feel this way and that their emotions are valid. You can say something like, "I'm so sorry to hear about the loss of your dog. It's normal to feel sad and grieving."\n3. Offer physical comfort: Sometimes, physical touch can be comforting. Hold your friend's hand, give them a hug, or offer a gentle touch on the arm or shoulder. However, respect their boundaries and only offer physical comfort if they seem open to it.\n4. Share memories: If your friend is comfortable with it, share some fond memories of their dog. This can help them feel less alone in their grief and provide a sense of comfort and nostalgia.\n5. Help with practical tasks: Losing a pet can be overwhelming, and your friend may not have the energy or motivation to take care of practical tasks. Offer to help with things like cleaning, errands, or meal preparation.\n6. Respect their boundaries: Everyone grieves differently, and it's essential to respect your friend's boundaries and pace. Don't push them to talk or share more than they're comfortable with, and avoid minimizing their feelings or telling them to "get over it."\n7. Suggest a memorial or ritual: Creating a memorial or ritual can be a meaningful way to honor your friend's pet and help them process their grief. This could be something simple like lighting a candle, planting a tree, or creating a memory box.\n8. Be patient: Grief is a process that takes time, and it's essential to be patient with your friend. Avoid putting pressure on them to "move on" or "get over" their loss.\n9. Encourage self-care: Losing a pet can be emotionally and physically draining. Encourage your friend to take care of themselves by getting enough rest, eating well, and engaging in activities that bring them comfort and joy.\n10. Follow up: Check in with your friend regularly to show that you're still thinking of them and care about their well-being. This can help them feel less isolated and more supported during this difficult time.\n\nRemember, there's no one-size-fits-all approach to comforting someone who's lost a pet. The most important thing is to be present, empathetic, and understanding.

**Reward Score** 3.91

---

### 7B Guiding 8B Iter3

I'm so sorry to hear that your friend is going through this difficult time. Losing a beloved pet can be a devastating experience. Here are some ways you can comfort your friend:\n\n1. Listen actively: Allow your friend to express their emotions and thoughts without interrupting or judging. Listen attentively to their stories, memories, and feelings about their dog.\n2. Validate their emotions: Acknowledge your friend's pain and let them know that their feelings are normal. You can say something like, "I'm so sorry you're going through this. It's okay to feel sad, angry, or overwhelmed."\n3. Offer physical comfort: Sometimes, physical

touch can be comforting. Offer a hug, hold their hand, or give them a gentle pat on the back. However, respect their boundaries if they don't feel like physical contact.\n4. Share memories: Share fond memories of the dog or stories about your own pets. This can help your friend feel less alone and more connected to others who have experienced a similar loss.\n5. Avoid clichŎ0e9s: Phrases like "It was just a dog" or "They're in a better place" can come across as insensitive. Instead, focus on acknowledging your friend's grief and offering support.\n6. Help with practical tasks: Offer to help with tasks such as cleaning up the dog's area, taking care of other pets, or running errands. This can help take some burden off your friend's plate and give them more time to grieve.\n7. Respect their grief timeline: Everyone grieves at their own pace, and there's no set timeline for mourning. Respect your friend's emotional state and don't pressure them to "get over it" or "move on."\n8. Encourage self-care: Suggest healthy coping mechanisms like exercise, meditation, or spending time in nature. Encourage your friend to take care of themselves physically and emotionally.\n9. Be patient: Losing a pet can be a long and difficult process. Be patient with your friend and remember that it may take time for them to heal.\n10. Consider a memorial or ritual: Suggest doing something to honor the dog's memory, such as planting a tree, making a donation to an animal welfare organization, or creating a memory book. This can help your friend feel more connected to their pet and find closure.\n\nRemember, everyone grieves differently, and there's no one-size-fits-all approach to comforting your friend. Be present, empathetic, and understanding, and let your friend know that you're there for them during this difficult time.

**Reward Score** $\boxed{4.41}$

---

### 70B Base

I'm so sorry to hear that your friend is going through a tough time. Losing a pet is never easy, and it's wonderful that you want to be there for them. Here are some tips on how to comfort your friend:\n\n1. **Listen and validate their feelings**: Allow your friend to express their emotions without judgment. Let them know that their feelings are normal and that it's okay to feel sad, angry, or guilty. Validate their loss by acknowledging the significance of their pet in their life.\n2. **Offer physical comfort**: A gentle hug, a comforting touch, or a sympathetic ear can go a long way in providing comfort. Sometimes, physical touch can be more comforting than words.\n3. **Let them talk**: Encourage your friend to share stories and memories of their pet. This can help them process their grief and celebrate the good times they had with their furry friend.\n4. **Avoid clichés**: Phrases like "It was just a pet" or "You can always get another one" can come across as insensitive. Instead, focus on the unique bond your friend shared with their pet.\n5. **Respect their boundaries**: Everyone grieves differently, and it's essential to respect your friend's boundaries. If they don't feel like talking, that's okay. Let them know you're there for them when they're ready to talk or need company.\n6. **Offer practical help**: Help your friend with tasks like making arrangements for the pet's remains, taking care of other pets in the household, or running errands. This can help take some burden off their plate.\n7. **Share a fond memory**: If you have a fond memory of their pet, share it with your friend. This can help them feel comforted knowing that others also cared about their pet.\n8. **Encourage self-care**: Grief can be exhausting. Encourage your friend to take care of themselves, whether it's getting enough rest, eating well, or engaging in activities they enjoy.\n9. **Be patient**: Grieving is a process that takes time. Be patient and understanding, and let your friend know that you're there for them in the long haul.\n10. **Remember important dates**: Mark important dates like the anniversary of their pet's passing or their pet's birthday. Reach out to your friend on these days to show you're still thinking of them and their beloved pet.\n\nSome specific things you could say to comfort your friend include:\n\n* "I'm so sorry for your loss. My heart goes out to you."\n* "Your pet was such an important part of your life, and it's okay to feel sad."\n* "I'm here for you, and I'll support you through this tough time."\n* "Do you want to talk about your pet? I'm all ears."\n* "Remember all the good times you had with your pet? Those memories will always be with you."\n\nRemember, there's no one-size-fits-all approach to comforting a grieving friend. Be genuine, empathetic, and patient, and you'll be helping your friend navigate this difficult

time.
**Reward Score** 5.25

---

**70B BoN-E 16**

I'm so sorry to hear that your friend is going through a tough time. Losing a pet is never easy, and it's wonderful that you want to be there for them. Here are some tips on how to comfort your friend:\n\n1. **Listen and validate their feelings**: Allow your friend to express their emotions without judgment. Let them know that their feelings are normal and that it's okay to feel sad, angry, or guilty. Acknowledge the significance of their loss and the bond they shared with their dog.\n2. **Offer physical comfort**: A hug, a pat on the back, or a gentle touch on the arm can go a long way in providing comfort. Sometimes, physical touch can be more comforting than words.\n3. **Let them talk about their pet**: Encourage your friend to share stories, memories, and anecdotes about their dog. This can help them process their grief and celebrate their pet's life.\n4. **Avoid clichés**: Phrases like "It was just a dog" or "You can always get another pet" can come across as insensitive. Instead, focus on the unique bond your friend shared with their dog and the impact it had on their life.\n5. **Be present**: Spend time with your friend, even if it's just sitting together in silence. Your presence can be comforting and help them feel less alone.\n6. **Respect their boundaries**: Everyone grieves differently, and it's essential to respect your friend's boundaries and needs. If they don't feel like talking, that's okay. Let them know you're there for them when they're ready to talk or need support.\n7. **Offer practical help**: Depending on your friend's needs, you could offer to help with tasks like cleaning, cooking, or running errands. This can help take some burden off their plate and give them more time to focus on their grief.\n8. **Encourage self-care**: Suggest activities that can help your friend relax and cope with their emotions, such as going for a walk, practicing yoga, or engaging in a hobby they enjoy.\n9. **Remember important dates**: Mark the anniversary of their dog's passing or their birthday, and reach out to your friend on those days to show you're still thinking of them and their pet.\n10. **Be patient**: Grieving is a process that takes time, and it's essential to be patient and understanding. Avoid putting pressure on your friend to "get over it" or "move on" too quickly.\n\nRemember, there's no one-size-fits-all approach to comforting a friend who's lost a pet. Be genuine, empathetic, and supportive, and you'll be helping your friend navigate this difficult time.

**Reward Score** 5.5

---

**Weak to Strong Search (7B Guiding 70B)**

I'm so sorry to hear that your friend is going through a tough time. Losing a pet is never easy, and it's essential to be supportive and understanding during this difficult period. Here are some tips to help you comfort your friend:\n\n1. **Listen and validate their feelings**: Allow your friend to express their emotions without judgment. Let them know that their feelings are normal and that it's okay to feel sad, angry, or guilty. Validate their grief by acknowledging the significance of their pet's role in their life.\n2. **Offer physical comfort**: A gentle hug, a pat on the back, or a comforting touch on the arm can go a long way in providing comfort. Physical touch can help release oxytocin, a hormone that promotes feelings of calmness and bonding.\n3. **Let them talk about their pet**: Encourage your friend to share stories, memories, and anecdotes about their dog. This can help them process their grief and celebrate the life of their beloved pet.\n4. **Avoid clichés and platitudes**: Phrases like "It was just a dog," "You can always get another pet," or "Time heals all wounds" can come across as insensitive and dismissive. Instead, focus on expressing empathy and understanding.\n5. **Respect their boundaries**: Everyone grieves differently, and it's essential to respect your friend's boundaries and needs. If they don't feel like talking, that's okay. Let them know you're there for them when they're ready to open up.\n6. **Offer practical help**: Help your friend with tasks such as making arrangements for the pet's remains, taking care of other pets in the household, or running errands. This can help take some burden off their plate and give them more time to focus on their emotional well-being.\n7. **Encourage self-care**: Suggest activities that can help your friend relax

and cope with their grief, such as going for a walk, practicing yoga, or engaging in a hobby they enjoy.\n8. **Remind them of the good times**: Share fond memories of their dog and highlight the happy times they shared. This can help your friend focus on the positive aspects of their pet's life and find comfort in those memories.\n9. **Be patient**: Grief is a process that takes time, and it's essential to be patient and understanding. Let your friend know that you're there for them in the long haul, not just in the immediate aftermath of their pet's passing.\n10. **Check in regularly**: After the initial shock has passed, it's essential to continue checking in with your friend regularly. This can help them feel supported and remembered during a difficult time.\n\nRemember, there's no one-size-fits-all approach to comforting a friend who's lost a pet. Be genuine, empathetic, and understanding, and you'll be well on your way to providing the support they need.

**Reward Score** | 5.68 |

---

**7B Guiding 70B Iter1**

I'm so sorry to hear that your friend is going through a tough time. Losing a pet is never easy, and it's wonderful that you want to be there for them. Here are some tips on how to comfort your friend:\n\n1. **Listen and validate their feelings**: Allow your friend to express their emotions without judgment. Let them know that it's okay to feel sad, angry, or guilty, and that their feelings are valid. Listen attentively to what they say, and show empathy through your body language and tone of voice.\n2. **Acknowledge the loss**: Recognize the significance of the loss and acknowledge the bond between your friend and their pet. You can say something like, "I'm so sorry about the loss of your furry friend. I know how much they meant to you."\n3. **Share a fond memory (if you have one)**: If you have a fond memory of their pet, share it with your friend. This can help them remember happy times and feel comforted. For example, "I'll never forget when [pet's name] did [funny/cute thing]. They were such a character!"\n4. **Offer physical comfort**: Sometimes, physical touch can be comforting. Offer a hug, hold their hand, or give them a gentle pat on the back. Be mindful of personal boundaries, though, and respect their space if they don't want physical contact.\n5. **Avoid clichŎ0e9s**: Phrases like "It was just a pet" or "You can always get another one" can come across as insensitive. Avoid minimizing their loss or implying that their pet was replaceable.\n6. **Be present**: Spend time with your friend, even if it's just sitting together in silence. Your presence can provide comfort and let them know they're not alone.\n7. **Help with practical tasks**: Offer to help with tasks like making funeral arrangements, taking care of other pets, or doing errands. This can help take some burden off their plate and give them more time to grieve.\n8. **Respect their boundaries**: Everyone grieves differently, and it's essential to respect your friend's boundaries and needs. If they don't want to talk or need some alone time, give it to them.\n9. **Check in regularly**: Losing a pet can be a long and difficult process. Check in with your friend regularly to show you care and are thinking of them.\n10. **Encourage self-care**: Grief can be exhausting. Encourage your friend to take care of themselves, whether it's getting enough rest, eating well, or engaging in activities they enjoy.\n\nRemember, there's no one-size-fits-all approach to comforting a friend who's lost a pet. Be patient, understanding, and flexible, and let your friend know that you're there for them.

**Reward Score** | 5.19 |

---

**7B Guiding 70B Iter2**

I'm so sorry to hear that your friend is going through a tough time. Losing a pet is never easy, and it's wonderful that you want to be there for them. Here are some tips on how to comfort your friend:\n\n1. **Listen and validate their feelings**: Allow your friend to express their emotions without judgment. Let them know that it's okay to feel sad, angry, or guilty, and that their feelings are normal. Validate their grief by acknowledging the significance of their loss.\n2. **Offer physical comfort**: Sometimes, a gentle hug, a pat on the back, or a comforting touch on the arm can go a long way in providing comfort. Be mindful of your friend's personal space, but a gentle physical gesture can help them feel less alone.\n3. **Let them talk about their pet**: Encourage your friend to share stories, memories, and

anecdotes about their dog. This can help them process their grief and celebrate the life of their beloved pet.\n4. **Avoid clichŎ0e9s and platitudes**: Phrases like "It was just a dog" or "You can always get another pet" can come across as insensitive. Instead, focus on expressing empathy and understanding.\n5. **Respect their boundaries**: Everyone grieves differently, and it's essential to respect your friend's boundaries and needs. If they don't feel like talking, don't push them. Let them know you're there for them when they're ready to open up.\n6. **Offer practical support**: Help your friend with practical tasks, such as making arrangements for the pet's remains, cooking a meal, or running errands. This can help take some burden off their plate and give them time to focus on their grief.\n7. **Share a fond memory (if you have one)**: If you have a fond memory of their dog, share it with your friend. This can help them feel comforted knowing that others also cared about their pet.\n8. **Encourage self-care**: Grief can be exhausting. Encourage your friend to take care of themselves, whether it's getting enough rest, eating well, or engaging in activities that bring them comfort and relaxation.\n9. **Be patient**: Grief is a process that takes time, and it's essential to be patient and understanding. Let your friend know that you're there for the long haul and that you'll support them through this difficult time.\n10. **Remind them of their strength**: Losing a pet can make your friend feel vulnerable and weak. Gently remind them of their strength and resilience, and let them know that they can get through this tough time.\n\nRemember, there's no one-size-fits-all approach to comforting a friend who's lost a pet. Be genuine, empathetic, and patient, and let your friend know that you're there to support them every step of the way.

**Reward Score** | 5.53

---

### 7B Guiding 70B Iter3

I'm so sorry to hear that your friend is going through a tough time. Losing a pet can be incredibly painful, and it's wonderful that you want to be there for them. Here are some tips to help you comfort your friend:\n\n1. **Listen and validate their feelings**: Allow your friend to express their emotions without judgment. Let them know that their feelings are normal and that it's okay to feel sad, angry, or guilty. Validate their pain by acknowledging the significance of their loss.\n2. **Offer physical comfort**: Sometimes, a gentle hug, a hand on the shoulder, or a comforting touch can go a long way in providing comfort. Be mindful of your friend's boundaries, but a physical gesture can help them feel supported.\n3. **Let them talk about their pet**: Encourage your friend to share stories, memories, and anecdotes about their dog. This can help them process their grief and celebrate their pet's life. Listen attentively and show interest in what they're sharing.\n4. **Avoid clichŎ0e9s**: Phrases like "It was just a pet" or "You can always get another dog" can come across as insensitive. Avoid minimizing their loss or implying that their feelings are not justified. Instead, focus on the unique bond they shared with their pet.\n5. **Respect their grieving process**: Everyone grieves differently, and there's no set timeline for healing. Be patient and understanding, and let your friend know that you're there for them in the days, weeks, or months ahead.\n6. **Offer practical help**: Depending on your friend's needs, you could offer to help with errands, cook meals, or assist with tasks that might be overwhelming them. This can help take some burden off their plate and give them space to focus on their grief.\n7. **Create a memorial or ritual**: Suggest doing something to honor their pet's memory, like lighting a candle, planting a tree, or creating a memorial photo album. This can help your friend feel more connected to their pet and find closure.\n8. **Remind them of happy times**: Share fond memories of their dog or encourage your friend to do so. Focusing on the happy times can help balance out the sadness and provide some comfort.\n9. **Be present**: Sometimes, simply being present with your friend can be incredibly comforting. Sit with them, go for a walk, or engage in a quiet activity together. Your presence can help them feel less alone in their grief.\n10. **Check in regularly**: After the initial shock of the loss has passed, it's essential to continue checking in with your friend regularly. Grief can be a long and winding road, and your ongoing support can make a significant difference in their healing process.\n\nRemember, there's no one-size-fits-all approach to comforting a friend who's lost a pet. Be patient, understanding, and flexible, and let your friend guide you on how best to support them during this difficult time.

**Reward Score** $\boxed{6.0}$

## J  THE USE OF LARGE LANGUAGE MODELS

In this work, large language models (LLMs), such as GPT-5, were only used as a writing assistant. It includes using GPT-5 to help with writing, such as grammar checking, word refinement, and improving readability.

