# OpenReview forum: "Aligning Frozen LLMs by Reinforcement Learning: An Iterative Reweight-then-Optimize Approach"
_ICLR.cc/2026/Conference — Submitted to ICLR 2026_

### Official Review · Reviewer_57N6 · 2025-10-26

**Soundness:** 2
**Presentation:** 2
**Contribution:** 2
**Rating:** 4
**Confidence:** 5

**Summary:**

This paper tackles the critical challenge of aligning Large Language Models (LLMs) when their parameters are frozen or inaccessible (e.g., API deployment). Standard alignment methods (RLHF, DPO) require weight updates, while existing inference-time methods (like Best-of-N) are often computationally expensive or provide only suboptimal, one-shot guidance.

**Strengths:**

The paper addresses a crucial problem with high practical relevance: aligning and customizing frozen LLMs. The core idea of translating iterative RL updates into a sequence of external value functions, rather than internal weight changes, is novel and enables successive policy improvement where existing methods offer only one-shot correction.

The framework is well-motivated by established RL theory. Framing the iterative updates (Eq. 6) as successive solutions to KL-constrained optimization problems (TRPO-style) provides an elegant theoretical foundation (Sec 2.2), supported by convergence analysis (Theorem 1).

**Weaknesses:**

The paper presents IRO as an inference-time alignment method, but this framing obscures significant practical overheads.Training Overhead: IRO requires a substantial iterative training phase (Algorithm 3) to learn the sequence of value functions. This involves repeated cycles of generation from the "frozen" LLM, scoring, and VF training. The paper completely omits an analysis of the computational cost or time required for this pre-computation phase.Inference Overhead: At inference time, IRO requires loading and querying the entire sequence of learned value functions ($V_0, ..., V_{t-1}$) for every candidate in the beam search at every decoding step (Eq. 8). This significantly increases the memory footprint and latency. For example, using T=3 7B VFs to guide a 70B model adds 21B parameters of overhead. The paper lacks concrete measurements of inference latency and memory usage compared to baselines.

The value function training (Eq. 7) relies entirely on Monte Carlo (MC) regression, using the sparse, end-of-trajectory reward as the target for every prefix. MC methods are known to suffer from high variance in long-horizon tasks, leading to noisy value estimates and potential instability in the iterative training loop.4 The empirical observation that the $\beta_t$ schedule needs careful adjustment for stability (L427) suggests that the optimization process is indeed sensitive. The paper does not explore lower-variance methods (e.g., GAE or TD-learning).

The theoretical framework and convergence guarantees (Sec 2.2, Sec 4.1) assume exact policy updates over the entire action space (Eq. 5). The implementation, however, relies on a localized beam search approximation (Sec 3). This introduces a significant gap; the beam search may greedily exploit inaccuracies in the value functions and fail to explore promising paths, potentially undermining the theoretical guarantees of monotonic improvement.

**Questions:**

In addition to the weakness section, I have the below questions:

How does IRO handle the high variance inherent in using Monte Carlo returns for value estimation? Why were lower-variance methods like GAE or TD-learning not utilized or compared?

The paper mentions IRO is similar to Reinforcement Fine-Tuning (RFT). How does the performance and total cost of IRO compare to standard RFT applied to the same base model (assuming weights were accessible)?

---

> ### Author Response · Authors · 2025-11-21
> **Response 1 for Reviewer 57N6**
>
> We thank the reviewer for the detailed review of the paper and the valuable feedback. Below, we address the reviewer's questions in a point-by-point manner.
>
> **Weakness**
>
> >W1. Training Overhead: IRO requires a substantial iterative training phase (Algorithm 3) to learn the sequence of value functions. This involves repeated cycles of generation from the "frozen" LLM, scoring, and VF training. The paper completely omits an analysis of the computational cost or time required for this pre-computation phase.
>
> **Response**
>
> Thanks for your question about training overhead. Please see Reviewer zS4m's Weakness 1 about the detailed analysis of pre-computation phase.
>
> >W2. Inference Overhead: At inference time, IRO requires loading and querying the entire sequence of learned value functions () for every candidate in the beam search at every decoding step (Eq. 8). This significantly increases the memory footprint and latency. For example, using T=3 7B VFs to guide a 70B model adds 21B parameters of overhead. The paper lacks concrete measurements of inference latency and memory usage compared to baselines.
>
> **Response**
>
> Thanks for your question about inference overhead.  Please see Reviewer uYrT Weakness 1 about this problem.
>
> To address your concern, we conduct different experiments on both TL;DR task and Instruct following task to measure the inference latency and memory usage in the overall comment. Besides this, we also provide a solution to compress multiple value functions into one, which largely reduces the memory cost and latency, while maintaining a good performance.
>
> >W3. The value function training (Eq. 7) relies entirely on Monte Carlo (MC) regression, using the sparse, end-of-trajectory reward as the target for every prefix. MC methods are known to suffer from high variance in long-horizon tasks, leading to noisy value estimates and potential instability in the iterative training loop.4 The empirical observation that the schedule needs careful adjustment for stability (L427) suggests that the optimization process is indeed sensitive. The paper does not explore lower-variance methods (e.g., GAE or TD-learning).
>
> **Response**
>
> Thank you for your concern. While MC regression suffers from high variance problem in long-horizon tasks, in our implementation, it remains stable in pracitice, and we additionally use some method to mitigate variance as outlined below.
>
>
> To ensure stability, we adopt several strategies:
> * Large rollout dataset (8k) is used to decrease the variance.
> * We explicitly separate the rollout data into the training and evaluation set when training the value function.
> * For every iteration's value function, we select the checkpoint that achieves the lowest evaluation loss to avoid overfitting.
>
> In addition, we also discuss the comparison with lower-variance method GAE and TD-learning for value function training in Q5.
>
> We have added the discussion about value function training into the paper.
>
> >W4. The theoretical framework and convergence guarantees (Sec 2.2, Sec 4.1) assume exact policy updates over the entire action space (Eq. 5). The implementation, however, relies on a localized beam search approximation (Sec 3). This introduces a significant gap; the beam search may greedily exploit inaccuracies in the value functions and fail to explore promising paths, potentially undermining the theoretical guarantees of monotonic improvement.
>
>
> **Response**
> Thank you for highlighting this important point. We agree that there is a gap between practical implementation and theoretical analysis about convergence. This gap arises because exhaustively traversing the entire action space is impossible, as it would require scoring all chunk-level actions, for which the cost will be unacceptable.
>
> In the practical implementation, we therefore adopt a Beam Search style algorithm with a diversity-first principle, which prevents the search from greedy exploitation and encourages exploration of multiple promising candidates.
>
> Importantly, IRO doesn't require the trained value function to be perfectly accurate. Each value function $\hat{V}^{\hat{\pi}_k}$ is trained to regress the returns generated by its corresponding policy  $\hat{\pi}_k$, which directly relates to the idea of policy iteration. Thus, even when the action update is approximate, the value function remains aligned with the distribution of the current policy.
>
> Therefore, as discussed in Section 5.3 (test time scaling with value functions), subsequent value functions help correct suboptimal trajectories introduced by earlier guidance, and increasing the compute budget (expanding action space) consistently leads to better performance.

---

> > ### Author Response · Authors · 2025-11-21
> > **Response 2 for Reviewer 57N6**
> >
> > **Questions**
> >
> > >Q5. How does IRO handle the high variance inherent in using Monte Carlo returns for value estimation? Why were lower-variance methods like GAE or TD-learning not utilized or compared?
> >
> > **Response**
> >
> > Thank you for question about value function training. Although IRO uses MC returns, the variance issue is mitigated by sampling a larger dataset. Besides, we demonstrate that regression using FUDGE loss in our paper is more effective than TD-learning and GAE below.
> >
> > In IRO, since the reward is sparse and it is only assigned at the final token, we used FUDGE loss in IRO , where each token's prediction is regressed on the final return. This method is mathematically equivalent to TD error when discount factor $\gamma=1$. We adopt this loss function because it is simple, stable, and widely used in prior value function training work [1,2,3,4].
> >
> > We conduct an experiment to train the value function with TD-learning and GAE, respectively, below. We observe that FUDGE loss is more effective than TD learning and GAE in our case.
> >
> > **TD-Learning**
> > The loss function for TD-learning is:
> >
> > $$
> > \hat{V}^{\hat\pi_{t-1}} := \arg \min_{V} ~ \mathbb{E}_{\tau \sim \mathcal{D}_t}  [ \sum _{h=1}^H  [ r(s_h, a_h)+ \gamma V(s _{h+1}) - V(s_h) ]^2  ],
> > $$
> >
> > where parameter $\gamma$ is the discounted factor.
> >
> > #### Table 14: Comparison between using FUDGE and TD error for value function training on 1st iteration of IRO. Reward score evaluated by a 6.9B reward model and win rates against reference summary evaluated by GPT4o-mini on 300 samples from test dataset. 1B->1B means that using 1B value functions to guide 1B policy, and 1B->6.9B means that using 1B value functions to guide 6.9B policy.
> > |Method|Reward (1B→1B)| Win Rate (1B→1B %) | Reward (1B→6.9B) | Win Rate (1B→6.9B %) |
> > |-|-|-|-|-|
> > |SFT|-4.52|6.00|-1.33|16.33|
> > |TD $\lambda=0.99$|-2.01|17.67|1.73|23.67|
> > |IRO Iter1|-0.91|26.05|1.98|37.67|
> >
> >
> >
> > **GAE**
> >
> > For completeness, we also consider using GAE to train the token-level value function. Firstly, let's briefly formalize value function training using GAE.
> >
> > Let $V_{\phi}$ denote the value function parameterized by $\phi$, and $\tau$ be trajectory. For each timestep $t$, the temporal-difference error is:
> > $$
> > \delta_t = r_t + \gamma V_\phi(s_{t+1}) - V_\phi(s_t).
> > $$
> >
> > The GAE advantages are computed as:
> > $$
> > \hat{A}_t = \delta_t + \gamma \lambda \hat{A} _{t+1}, \qquad t = T-1,\dots,0.
> > $$
> >
> > The target for each prefix state is given by:
> > $$
> > \hat{V}^{\text{target}}_t = \hat{A} _t + V _\phi(s_t).
> > $$
> >
> > Finally, the value function parameters are updated by minimizing the MSE between predicted values and λ-return targets:
> > $$
> > \mathcal{L} _V(\phi)
> > = \mathbb{E} _{\tau,t}\left[ \big( V _\phi(s_t) - \hat{V}^{\text{target}} _t \big)^2 \right].
> > $$
> >
> > #### Table 15: Comparison between using FUDGE and TD error for value function training on 1st iteration of IRO. Reward score evaluated by a 6.9B reward model and win rates against reference summary evaluated by GPT4o-mini on 300 samples from test dataset. 1B->1B means that using 1B value functions to guide 1B policy, and 1B->6.9B means that using 1B value functions to guide 6.9B policy. Here $\lambda=1$.
> > |Method|Reward (1B→1B)| Win Rate (1B→1B %) | Reward (1B→6.9B) | Win Rate (1B→6.9B %) |
> > |-|-|-|-|-|
> > |GAE $\gamma=1$|-1.05|23.33|1.93|36.67|
> > |GAE $\gamma=0.99$|-1.01|21|1.95|37.67|
> > |GAE $\gamma=0.98$|-1.00|20.33|1.92|37.00|
> > |GAE $\gamma=0.97$|-1.01|21.33|1.93| 33.67|
> > |GAE $\gamma=0.95$| -0.76| 16.67|1.93|33.67|
> > |IRO Iter1|-0.91|26.05|1.98|37.67|
> >
> > #### Table 16: AlpacaEval 2 length-controlled win rate and raw win rate compared against GPT-4 for 8B model on 200 subsamples.
> > | Method                     | Compared Against GPT-4 LC Win Rate | Compared Against GPT-4 Win Rate | Compared Against BoN-E LC Win Rate | Compared Against BoN-E Win Rate |
> > |----------------------------|-------------------------------------|----------------------------------|-------------------------------------|----------------------------------|
> > | IRO Iter1                  |43.71|40.75|55.63|53.92|
> > | GAE  $\gamma=0.99$                 |38.92|37.46|48.92|43.96|
> >
> >
> >
> >
> >
> > In conclusion, we compare FUDGE loss with low-variance method for training token-level value function, FUDGE loss exhbits better performance. We will add this part into paper to make it more complete.
> >
> > **Reference**
> >
> > [1] Yang, Kevin, and Dan Klein. "FUDGE: Controlled text generation with future discriminators." arXiv preprint arXiv:2104.05218 (2021).
> >
> > [2] Mudgal, Sidharth, et al. "Controlled decoding from language models." arXiv preprint arXiv:2310.17022 (2023).
> >
> > [3] Liu, Zhixuan, et al. "Inference-time language model alignment via integrated value guidance." arXiv preprint arXiv:2409.17819 (2024).
> >
> > [4] Han, Seungwook, et al. "Value augmented sampling for language model alignment and personalization." arXiv preprint arXiv:2405.06639 (2024).

---

> ### Author Response · Authors · 2025-11-21
> **Response 3 for Reviewer 57N6**
>
> >Q6. The paper mentions IRO is similar to Reinforcement Fine-Tuning (RFT). How does the performance and total cost of IRO compare to standard RFT applied to the same base model (assuming weights were accessible)?
>
> **Response**
>
> Thanks for your question about the comparison between RFT [1] and IRO. We clarify that while IRO and RFT shares a high-level similar workflow, both uses a dataset with reward signal/feedback to finetune a model/API, but two methods are fundamentally different.
>
> * In RFT, user need to design a grader, then the model is trained to learn through iterative feedback by the grader, receiving rewards based on how closely its outputs align with desired behaviors or goals.
> * IRO sidestep weight updates, and the reward model is served as a grader, and IRO trains only a sequence of lightweight value functions to guide the output to align with the desired behaviors.
>
> Since the OpenAI RFT's implementation details are not publicly release, a controled comparison is not feasible. Here we provide a comparison analysis for these two methods.
>
> Assume that the base model is a 70B model and the weight is accessible.
> * RFT requires full 70B model fine-tuning, with the cost similar to or exceeding that of DPO training, which may needs at least multi-node H100 resources.
> * IRO requires only forward passes through the frozen LLM plus standard supervised training on small value functions (7B), which can be implement in 4 H100 GPUs.
>
>
>
> **Reference**
>
> [1]. https://platform.openai.com/docs/guides/reinforcement-fine-tuning

---

> ### Author Response · Authors · 2025-11-27
>
> Dear Reviewer 57N6,
>
> Thank you sincerely for taking the time to review our work. We sincerely appreciate your thoughtful comments and evaluations. In our response, we have addressed your concerns point by point. If you have any further questions or feedback, we would be happy to continue the discussion.
>
> Best regards,
>
> Authors

---

### Official Review · Reviewer_zS4m · 2025-10-27

**Soundness:** 3
**Presentation:** 4
**Contribution:** 3
**Rating:** 6
**Confidence:** 4

**Summary:**

This paper introduces Iterative Reweight-then-Optimize (IRO), a novel framework for aligning large language models (LLMs) that are "frozen," meaning their parameters cannot be directly updated. This work tackles the significant practical problem of how to improve or customize a deployed model (e.g., one accessed via API) using only an external outcome reward model (ORM).

The authors situate their work between two existing paradigms, highlighting the limitations of both:
 1. training-time alignment (e.g., RLHF, DPO) is effective but static; models can't be adapted post-deployment without costly retraining
 2. Inference-time alignment (e.g., Best-of-N, ARGS) is flexible but either computationally exorbitant (BoN requires a very large $N$) or suboptimal (relying on one-shot guidance or inaccurate token-level scores).

IRO bridges this gap by proposing an RL-style iterative refinement process that does not touch the base model's weights. The core idea is to iteratively train a sequence of lightweight value functions. In the training phase, each new value function is trained on data generated by the base model guided by the previous value functions. At inference time, the base model's generation is guided by a search algorithm that reweights candidate tokens using a weighted sum of all the learned value functions.

The paper claims three primary contributions:
1. A Novel Framework: A practical, multi-step alignment method for frozen LLMs.
2. Theoretical Grounding: The authors prove that IRO is a form of policy iteration that converges to the optimal policy and is more token- and query-efficient than the standard Best-of-N (BoN) baseline for achieving the same performance.
3. Strong Empirical Results: The paper demonstrates that IRO significantly outperforms strong baselines (including BoN and weak-to-strong search) on instruction-following (AlpacaEval 2.0) and summarization (TL;DR) benchmarks. Notably, it shows strong "weak-to-strong" capabilities, where small value functions (e.g., 1B or 7B) effectively guide much larger base models (e.g., 6.9B or 70B).

**Strengths:**

1. High Significance and Novelty: The problem of aligning frozen or "black-box" models is extremely relevant. The core idea of using an iterative RL process to learn a sequence of value functions—effectively performing policy iteration without requiring weight updates—is a novel and powerful concept that significantly extends beyond existing one-shot inference methods.
2. Strong Theoretical Foundation: The method is not just a heuristic. It is well-grounded in RL theory, with clear connections drawn to TRPO. The inclusion of a convergence proof (Theorem 1) and a formal cost-efficiency analysis against BoN (Proposition 1) adds significant rigor.
3. Comprehensive Empirical Evaluation: The experimental validation is a key strength. The authors test IRO across multiple model families and scales (Pythia 1B/6.9B, Llama-3 8B/70B) on standard, challenging benchmarks (AlpacaEval, TL;DR). The inclusion of an extension to math reasoning (GSM8K) also successfully demonstrates the method's generality beyond continuous rewards.
4. Strong Baselines and "Weak-to-Strong" Results: The paper compares IRO against relevant and strong baselines, including DPO, BoN-E, and Weak-to-Strong Search. The fact that IRO consistently outperforms them is impressive. The "weak-to-strong" results (e.g., a 7B value model guiding a 70B policy) are particularly impactful, suggesting a practical path for steering large models efficiently.
5. Thorough Ablation Studies: The appendix provides excellent ablation studies that validate key design choices. These include the importance of the diversity-first search strategy, the method's robustness to imperfect reward models, and the impact of hyperparameters like chunk length $L$ and value function size.

**Weaknesses:**

1. Training Cost Analysis: The paper heavily focuses on the inference-time efficiency gains over BoN. However, the IRO framework requires an iterative training process: $T$ iterations of generating a full dataset and training a (lightweight) value function. This "offline" training cost is not trivial and is not thoroughly compared against the one-time training cost of methods like DPO or the pure (but massive) inference cost of BoN with a very large $N$. A "total compute" comparison would make the efficiency claims more complete.
2. Hyperparameter Sensitivity: The method introduces several new hyperparameters, including the number of iterations $T$, the $\beta_t$ schedule, the search parameters $K$ and $B$, and the chunk length $L$. The ablations suggest that performance is quite sensitive to some of these (e.g., $\beta_t$ in Fig. 8 and $L$ in Fig. 11). This may pose a challenge for practitioners seeking to apply IRO to new tasks, as it could require significant tuning.
3. Inference Latency with $T$ Iterations: The inference process (Algorithm 4) requires running all $T$ (or $I$) learned value functions at each step of the beam search to compute the score (Eq. 8). While each VF is "lightweight," this still means that inference latency scales linearly with the number of training iterations. This is noted in the complexity analysis (Table 1), but it is a practical limitation that could be discussed more.

**Questions:**

Questions:

1. Training Cost Comparison: Could you provide a more detailed analysis of the training cost of IRO? How does the total compute for $T=3$ iterations of value function training (including data generation) compare to the one-time training cost of DPO on the same dataset, or the total inference cost of BoN at a level where it achieves comparable performance (e.g., $N=64$ or $N=128$)?
2. Choice of $\beta_t$ Schedule: The paper uses a constant $\beta_t=1$ for the TL;DR task but an increasing schedule ($\beta_t = \{1, 2, 2.5\}$) for the UltraFeedback task. Theorem 1 also relies on an increasing $\beta_t = \mathcal{O}(\sqrt{t})$. Could you elaborate on the empirical or theoretical intuition for this choice? Is an increasing schedule a form of regularization to ensure stability in later iterations, as suggested in the text?
3. Performance Saturation: The experiments consistently show improvement from Iter 1 to Iter 3. Have you experimented with more iterations (e.g., $T > 3$)? Does performance continue to improve, or does it saturate or become unstable?
4. Inference Latency: The inference cost scales with the number of value functions $I$ (Table 1), as all $I$ VFs must be queried. At $T=3$, this is likely manageable. But if $T$ were, for example, 10, would this become a significant latency bottleneck in practice compared to the base model's forward pass?

5. Clarity on "Diversity-First Principle": This principle is noted as being critical for performance (Fig. 10), but it is only briefly described in the main text. Expanding this with a slightly more formal definition or a small algorithm block in the appendix would be very helpful for reproducibility.
6. Figure 1 Clarity: Figure 1 is a bit dense. The "Guided generation step" and "Value training step" are clear, but the "Generation in the IRO" diagram at the bottom is complex and somewhat redundant with the top-left diagram. This could potentially be simplified to improve its role as an introductory illustration.
7. Reward Model Quality: The paper rightly includes an ablation on imperfect reward models. It might be worth adding a concluding sentence to the main discussion (Section 5) to explicitly state that the final performance of IRO is, naturally, bounded by the quality of the ORM used for supervision, even as the method proves robust to weaker RMs.

---

> ### Author Response · Authors · 2025-11-21
> **Response 1 for Reviewer zS4m**
>
> We thank the reviewer for the detailed review of the paper and the valuable feedback. Below, we address the reviewer's questions in a point-by-point manner.
>
> **Weakness**
> >W1. Training Cost Analysis: The paper heavily focuses on the inference-time efficiency gains over BoN. However, the IRO framework requires an iterative training process: iterations of generating a full dataset and training a (lightweight) value function. This "offline" training cost is not trivial and is not thoroughly compared against the one-time training cost of methods like DPO or the pure (but massive) inference cost of BoN with a very large....
>
> Thanks for your insightful suggestion. For the offline training cost of IRO, it mainly comes from two parts: rollout cost and value function training cost.
>
> **Rollout cost**
> At the i-th iteration,  $(i-1)$ trained value functions are used to guide the frozen policy. Assume that in each iteration, we need a rollout dataset $|\mathcal{D}| = D$. Let the sequence length $H$, beam width $K$, and select $B$ successors in each selection. Since the policy model and value function may differ in size, we assume that the per-token cost of policy and value function is $C_p$ and $C_v$, respectively. Thus, the total rollout compute cost is $O(H^2KBDC_p)+O(DH^2KB(i-1)C_v)$ (calculation details can be found in Appendix E).
>
> **Value function training cost**
> In each iteration, IRO trains a lightweight value function, and its cost is comparable to the supervised fine-tuning.
>
> **Total cost for IRO**
> Therefore, summing across $I$ iterations, the total "offline" training cost is $O(H^2KBDIC_p)+O(DH^2KBI^2C_v)$ plus $I-1$ value fucntion training cost. Because the value function can be much smaller than the base policy, and the total iteration $I$ is small (usually 2-3 in practice), the cost can be controlled.
>
> **DPO**
> For a fair comparison, we consider the online version of the DPO [1]. In this algorithm, the policy generates multiple candidate continuations for each prompt, and the reward model is used to identify the best and worst among them. These selected samples are then used to perform DPO updates. We believe that comparing the cost of IRO and this online version of DPO is fair in the sense that both involve online generation.
>
> The total rollout cost in $I$ iterations for online DPO is $O(H^2KBDC_pI)$. Since DPO is full model finetuning on the base policy, and the training cost and memory cost is much larger than SFT, whereas IRO trains only lightweight value function (1B instead of 6.9B or 7B instead of 70B). Therefore, $O(DH^2KBI^2C_v)$ is relatively small, and the total training FLOPs of IRO are controllable.
>
> **Best-of-N**
> The inference cost of BoN is $O(H^2NC_p)+O(H^2NC_r)$, where $N$ is the generation number and $C_r$ is the per-token cost for reward model.
>
> We have made the computational complexity analysis more complete in the revised version.
>
> Reference
>
> [1]: Guo, Shangmin, et al. "Direct language model alignment from online ai feedback." arXiv preprint arXiv:2402.04792 (2024).
>
>
> >2. Hyperparameter Sensitivity: The method introduces several new hyperparameters, including the number of iterations , the  schedule, the search parameters and , and the chunk length. The ablations suggest that performance is quite sensitive to some of these (e.g., in Fig. 8 and in Fig. 11). This may pose a challenge for practitioners seeking to apply IRO to new tasks, as it could require significant tuning.
>
> **Response**
>
> Thanks for pointing out this concern. While IRO does introduce several hyperparameters, our experiments indicate that most of these hyperparameters can be fixed, and only a few require task-specific tuning.
>
> * Chunk length: It affects the granularity of value guidance. We find that 16-32 tokens basically achieve a good balance between performance and cost. Importantly, even when chunk length is increased, IRO still maintains a very stable advantage over BoN.
> * Dataset size: For training the value function, we observe that a relatively small dataset (e.g., 8k trajectories) is effective.
> * Search parameter: As shown in Section 5.3, test-time scaling with value function, larger search budgets yield better results. But it depends on the cost, as larger search budgets lead to higher costs.
> * Number of iterations and schedule: Across all datasets, 2–3 iterations work well, and a monotone or constant $\beta$ is sufficient. We recommend choosing the best on a small held-out validation set.
>
> Overall, although the ablations in Fig. 8 and Fig. 11 illustrate that hyperparameters affect per-iteration dynamics, the final performance is not highly sensitive. In the revised version, we have included practical recommendations to make IRO straightforward to apply to new tasks.
>
> In addition, hyperparameter tuning is unavoidable. Alternative approaches such as RLHF or other inference-time methods require tuning hyperparameters—including KL penalties, learning rates, and so on.

---

> > ### Author Response · Authors · 2025-11-21
> > **Response 2 for Reviewer zS4m**
> >
> > >W3. Inference Latency with Iterations: The inference process (Algorithm 4) requires running all (or) learned value functions at each step of the beam search to compute the score (Eq. 8)....
> >
> > **Response**
> > Thanks for raising this concern. In our experiment, we found that 2-3 iterations for IRO achieves a good trade-off between performance and cost. Thus, we only need 2-3 value functions at most in practice.
> >
> > In addition, it is possible to use some method to reduce the memory and latency, such as using parameter-efficient method LoRA to train value functions. In the overall comment, we provide a alternative solution that compressing multiple value functions into single one. This cuts both inference-time peak memory by 18–45%  and latency by 24–40% while only drop 0-1.7% win rate performance, providing a practical solution for deployment in more constrained environments.
> >
> >
> >
> > **Questions**
> >
> > >Q4. Training Cost Comparison: Could you provide a more detailed analysis of the training cost of IRO? ...?
> >
> > **Response**
> >
> > Thanks for your concern. We provide a detailed comparison of IRO's training cost compared with DPO and BoN in Weakness 1. Also, we empirically show that IRO can achieve better performance than BoN, even under the same total inference.
> >
> > To address the question regarding the cost of BoN, we compare BoN and IRO in Table 11 with an increasing N on the summarization task. Here IRO still uses $K=4,B=4$ beam search style algorithm with 1b value fucntion.
> >
> > From the Table 12, increasing $N$ improves performance initially (up to $N=256$), but further increases yield diminishing win rate as the reward mode is not accurate enough. In addition, IRO iter3 surpasses the BoN performance with large N under the same reward model. This means that IRO achieves more effective policy improvement than simply increasing inference cost.
> >
> > #### Table 12: Win rate (%) comparison between Best-of-N and IRO. Here, the win rate is calculated against the reference summary evaluated by GPT4o-mini on 300 samples from the test dataset. For a fair comparison, we use the same 1b reward model.
> > | Method     | Win Rate (1B→1B %) | Win Rate (1B→6.9B %) |
> > |------------|---------------------|------------------------|
> > | SFT        | 6.00 | 16.33 |
> > | BoN 16     | 12.67 | 29.33 |
> > | BoN 32     | 18.33| 38.56  |
> > | BoN 64     | 20.33 | 40.78  |
> > | BoN 128    | 20.22 | 43.22 |
> > | BoN 256    | 23.44 | 44.56 |
> > | BoN 512    | 21.44 | 47.22 |
> > | BoN 1024   | 21.89 | 44.11 |
> > | BoN 2048   | 22.67 | 44.33 |
> > | IRO Iter1  | 26.00 | 37.67 |
> > | IRO Iter2  | 31.33 | 48.00 |
> > | IRO Iter3  | 35.56  | 49.67 |
> >
> >
> >
> > >Q5. Choice of $\beta$ Schedule: The paper uses a constant for the TL;DR task but an increasing schedule (1;2;2.5) for the UltraFeedback task....
> >
> > **Response**
> >
> > Thank you for the question about the $\beta$ schedule. We find that both constant and increasing $\beta$ schedules work in practice. The main effect of $\beta$ is to control the trade-off between exploitation of previously learned value functions and exploration guided by newly trained value function.
> >
> > Empirically, we find that
> > * On simple tasks such as summarization, a constant $\beta$ already provides stable convergence.
> > * On more challenging instruct following, a increasing $\beta$ schedule stabilizes later iterations.
> >
> > >Q6. Performance Saturation: The experiments consistently show improvement from Iter 1 to Iter 3....
> >
> > **Response**
> >
> > To address your question, we ran an additional 4th iteration on the summarization task and summerized the result in Table 1. We observe that while the 4th iteration continues to yield improvements on reward score, the gains are marginal:
> > 1. 1B value functions guide 1B policy: Win rate improves by +2.11%, reward score by +0.17.
> > 2. 1B value functions guide 6.9B policy: Win rate slightly decreases by -1.34%, reward increases by +0.01.
> >
> > In contrast to the substantial improvements from SFT to 3 iteration (e.g., +29.56% in 1B policy and +32.00% in 6.9B policy), the improvements from Iter3 to Iter4 is minor. Meanwhile, inference cost increases linearly with the number of value functions. Thus, we believe that in this case, three iterations offers a good trade-off between performance and efficiency.
> >
> > #### Table 13: Reward score evaluated by a 6.9B reward model and win rates against reference summary evaluated by GPT4o-mini on 300 samples from test dataset. We put baseline from paper for comparison. 1B->1B means that using 1B value functions to guide 1B policy, and 1B->6.9B means that using 1B value functions to guide 6.9B policy.
> > |Method|Reward (1B→1B)| Win Rate (1B→1B %) | Reward (1B→6.9B) | Win Rate (1B→6.9B %) |
> > |-|-|-|-|-|
> > |SFT|-4.52|6.00|-1.33|16.33|
> > |IRO Iter1|-0.91|26.00|1.98|37.67|
> > |IRO Iter2|-0.27|31.33|1.97|48.00|
> > |IRO Iter3|-0.06|35.56|2.04|49.67|
> > | **IRO Iter4**|**0.11**|**37.67**|**2.05**| **48.33**|

---

> ### Author Response · Authors · 2025-11-21
> **Response 3 for Reviewer zS4m**
>
> >Q7. Inference Latency: The inference cost scales with the number of value functions (Table 1), as all VFs must be queried. At T=3, this is likely manageable. But if T were, for example, 10, would this become a significant latency bottleneck in practice compared to the base model's forward pass?
>
> **Response**
>
> In our experiment, we find that three iterations offer a good trade-off between performance and efficiency. In practice, we can determine the optimal number of iteration on a held-out validation set, and early stop it.
>
>
> >Q8. Clarity on "Diversity-First Principle": This principle is noted as being critical for performance (Fig. 10), but it is only briefly described in the main text. Expanding this with a slightly more formal definition or a small algorithm block in the appendix would be very helpful for reproducibility.
>
> **Response**
>
> Thanks for your valuable suggestion. In the revised version, we have added a more formal definition along with a short algorithm block in the Appendix C.
>
> Formally, given a candidate set $\mathcal{C}$, let $g:\mathcal{C} \to \mathcal{K}$ be a grouping function that maps identical candidates to the same group. For any selected subset $\mathcal{S} \in \mathcal{C}$, we define the diversity measure $\mathrm{Div}(\mathcal{S}) = |\{g(y): y \in \mathcal{S}\}|$, which is exactly the number of different candidates. At each selection step, we choose the $K$ beams by solving:
> \begin{align}
>     \mathcal{S}^* = \arg \max_{\mathcal{S} \in \mathcal{C},|\mathcal{S}| = K} [\lambda \cdot \mathrm{Div}(\mathcal{S}) + V(y) ],
> \end{align}
> where $\lambda$ is sufficiently large to ensure the diversity..
>
> After selecting $\mathcal{S}^*$, each beam expands $B$ successors, and the procedure repeats at the next chunk.
>
> >Q9. Figure 1 Clarity: Figure 1 is a bit dense. The "Guided generation step" and "Value training step" are clear, but the "Generation in the IRO" diagram at the bottom is complex and somewhat redundant with the top-left diagram. This could potentially be simplified to improve its role as an introductory illustration.
>
> **Response**
>
> Thanks for your valuable suggestion. We have removed the bottom part in Fig 1, and moved the decoding part to the Appendix to further illustrate.
>
> >Q10. Reward Model Quality: The paper rightly includes an ablation on imperfect reward models. It might be worth adding a concluding sentence to the main discussion (Section 5) to explicitly state that the final performance of IRO is, naturally, bounded by the quality of the ORM used for supervision, even as the method proves robust to weaker RMs.
>
> **Response**
>
> Thanks for your valuable suggestion. We have clarified this point in section 5.

---

> ### Author Response · Authors · 2025-11-27
>
> Dear Reviewer zS4m,
>
> Thank you sincerely for taking the time to review our work. We sincerely appreciate your thoughtful comments and evaluations. In our response, we have addressed your concerns point by point. If you have any further questions or feedback, we would be happy to continue the discussion.
>
> Best regards,
>
> Authors

---

### Official Review · Reviewer_uYrT · 2025-10-31

**Soundness:** 3
**Presentation:** 3
**Contribution:** 2
**Rating:** 4
**Confidence:** 4

**Summary:**

The paper proposes a novel alignment algorithm for steering large language models without modifying their weights. The proposed approach, Iterative Reweight-then-Optimize (IRO), trains value functions, which are smaller than the base language model, to steer generation toward the desired reward. The proposed method brings significant improvements on the AlpacaEval 2.0 benchmark.

**Strengths:**

- The exposition of the idea is generally clear.
- Steering a large model without modifying its weight is an important problem.
- Theoretical claims seem valid.

**Weaknesses:**

- The primary concern is the computational overhead of the proposed method.
    - IRO may require significant training compute, and, furthermore, non-trivial inference-time memory and latency overhead. Please correct me if I am wrong.
    - For example, if we use the 7B value model and apply IRO for 3 iterations, does this mean that we need to run the 7B model three times during the inference, and the number of parameters being trained is 21B?
- Discussion on the connection to Blockwise Best-of-N decoding (Mudgal et al., 2023), a closely related decoding method, is missing.

**Questions:**

- Can you clarify the different to Blockwise Best-of-N approach? Empirical comparison will also be appreciated.
- Can you elaborate on the computational overhead of IRO? Will there be any measure that can be taken to reduce this overhead? For example, by using LoRA?
- From Appendix H.2.4, I found that the value function are initialized from the reward model. Is this process necessary? What if the reward model is a black box or not a neural network such that we can not initialize the value networks form the reward model?

---

> ### Author Response · Authors · 2025-11-21
> **Response 1 for Reviewer uYrT**
>
> We thank the reviewer for the detailed review of the paper and the valuable feedback. Below, we address the reviewer's questions in a point-by-point manner.
>
> **Weakness**
> >W1. The primary concern is the computational overhead of the proposed method. IRO may require significant training compute, and, furthermore, non-trivial inference-time memory and latency overhead. Please correct me if I am wrong. For example, if we use the 7B value model and apply IRO for 3 iterations, does this mean that we need to run the 7B model three times during the inference, and the number of parameters being trained is 21B?
>
> **Response**
>
> Thank you for raising this concern. While IRO does introduce additional value-function models across iterations at inference time, this design is standard in inference-time alignment methods for frozen-policy settings, and the overhead can be reduced in practice.
>
> First, the overall structure of IRO, which uses multiple lightweight auxiliary models to guide a larger frozen base model, is consistent with recent inference-time alignment frameworks. For example, TreeBoN [1] and weak-to-strong [2,4] compute the implicit reward using the log difference between a base model and a DPO finetuned model; multi-objective controlled decoding [5] requires 2-3 auxiliary models on different objectives; and IVG [3] relies on 3 model on both implicit reward and value function guidance.
>
> Second, the value functions in IRO are lightweight  and small relative to the base model (e.g., 1B v.s. 6.9B and 7B v.s. 70B). Even if three value functions are used, the compute and memory overhead at inference time remains dominated by the frozen base model. Moreover, parameter-efficient techniques such as LoRA can be used to train value functions to further reduce the cost.
>
> Finally, in our overall comment, we present a compression strategy to compress multiple value functions into a single one. This cuts both inference-time peak memory by 18–45%  and latency by 24–40% while only drop 0-1.7% win rate performance, providing a practical solution for deployment in more constrained environments.
>
> **References**
>
> [1] Qiu, Jiahao, et al. "Treebon: Enhancing inference-time alignment with speculative tree-search and best-of-n sampling." arXiv preprint arXiv:2410.16033 (2024).
>
> [2] Zhou, Zhanhui, et al. "Weak-to-strong search: Align large language models via searching over small language models." Advances in Neural Information Processing Systems 37 (2024): 4819-4851.
>
> [3] Liu, Zhixuan, et al. "Inference-time language model alignment via integrated value guidance." arXiv preprint arXiv:2409.17819 (2024).
>
> [4] Zhu, Wenhong, et al. "Weak-to-strong preference optimization: Stealing reward from weak aligned model." arXiv preprint arXiv:2410.18640 (2024).
>
> [5] Son, Seongho, et al. "Robust Multi-Objective Controlled Decoding of Large Language Models." arXiv preprint arXiv:2503.08796 (2025).
>
>
>
> >W2. Discussion on the connection to Blockwise Best-of-N decoding (Mudgal et al., 2023), a closely related decoding method, is missing.
>
> **Response**
>
> Thank you for raising this point. We include a brief discussion about Blockwise Best-of-N decoding in both the introduction and the related work section. We apologize for not making the connection more explicit. We now clarify the relationship and key differences.
>
> The method "Blockwise Best-of-N decoding" (also referred to as controlled decoding, CD) is closely related to IRO: it also trains an external token-level value function on the rollout dataset, which is generated by reference policy and evaluated by a reward model, to provide test-time guidance using blockwise best-of-n.
>
> Here are the key differences between CD and IRO:
>
> * CD adopts a one-shot guidance, which means that they train a single value function and apply it once at test time, which limits the improvement. However, IRO enables multi-iteration policy improvement by employing lightweight, iteratively trained value functions, which is supported by our theoretical analysis.
> * At the decoding stage, controlled decoding applies a blockwise best-of-N decoding strategy at each block, which means that it generates N partial candidates and uses the trained value function to select the top one. While effective, this method can greedily exploit inaccuracies in the value functions and fail to explore promising paths when the trained value function is not accurate enough. Instead, IRO adopts a beam search style decoding strategy with a diversity-first principle, which ensures exploration during decoding. And IRO trains multiple value functions, which avoids the problem that the first iteration's value function is not accurate enough.
>
> We will make this point clearer in the paper.

---

> ### Author Response · Authors · 2025-11-21
> **Response 2 for Reviewer uYrT**
>
> **Questions**
>
> >Q3. Can you clarify the different to Blockwise Best-of-N approach? Empirical comparison will also be appreciated.
>
> **Response**
>
> To address your concern, we provide a comparison between IRO and Blockwise Best-of-N decoding (CD) on both the summarization task and the following task.
>
> Since CD uses the same loss function to train the value function, and optimizes the same objective, where training data are generated from the current policy, we implement CD by using the value function from the first iteration to perform blockwise BoN.
>
> #### Table 9: Reward score evaluated by a 6.9B reward model and win rates against reference summary evaluated by GPT4o-mini on 300 samples from the test dataset. 1B->1B means that using 1B value functions to guide 1B policy, and 1B->6.9B means that using 1B value functions to guide 6.9B policy.
> |Method|Reward (1B→1B)| Win Rate (1B→1B %) | Reward (1B→6.9B) | Win Rate (1B→6.9B %) |
> |-|-|-|-|-|
> |SFT|-4.52|6.00|-1.33|16.33|
> |IRO Iter1|-0.91|26.00|1.98|37.67|
> |IRO Iter2|-0.27|31.33|1.97|48.00|
> |IRO Iter3|-0.06|35.56|2.04|49.67|
> |**CD**| **-2.35** | **18.33** |**0.58**|**25.33**|
>
> #### Table 10: AlpacaEval 2 length-controlled win rate and raw win rate compared against GPT-4 for 8B model. The decoding process maintains a beam width of 4 with 4 candidates preserved per state, and $l=16$ tokens as a state. For fairness, we use  $N=4*4$ for BoN.
> | Method                     | Compared Against GPT-4 LC Win Rate | Compared Against GPT-4 Win Rate | Compared Against BoN-E LC Win Rate | Compared Against BoN-E Win Rate |
> |--|-|-|-|-|
> | Meta-Llama-3-8B-Instruct   | 30.71 | 29.63  | 38.77| 38.32  |
> | IRO Iter1 | 41.06 | 37.32   | 55.42  | 53.60  |
> | IRO Iter2 | 42.20 | 40.00 | 55.75 | 55.65  |
> | IRO Iter3  | 43.11 | 41.00  | 57.00| 56.46 |
> |**CD**| **38.24** | **37.56** | **49.05** | **47.82**|
>
>
>
> #### Table 11: AlpacaEval 2 length-controlled win rate and raw win rate compared against GPT-4 for 70B model. The decoding process maintains a beam width of 4 with 4 candidates preserved per state, and $l=16$ tokens as a state. For fairness, we use  $N=4*4$ for BoN.
> | Method| Compared Against GPT-4 LC Win Rate | Compared Against GPT-4 Win Rate | Compared Against BoN-E LC Win Rate | Compared Against BoN-E Win Rate |
> |-|-|-|-|-|
> | Meta-Llama-3-70B-Instruct  | 43.11 | 43.11  | 42.39  | 41.80|
> | IRO Iter1| 49.00| 48.32| 53.39| 52.86|
> | IRO Iter2 | 49.75 | 49.19| 55.69| 53.91 |
> | IRO Iter3 | 49.77 | 49.07| 55.62| 55.45 |
> |CD| **46.25**|**45.33**|**49.90**|**48.60**
>
>
> From Table 9-11, we observe that
> * When using the same value function (IRO Iter1 v.s. CD), beam search with diversity-first principle consistently outperforms the block-wise BoN strategy. It suggests that under a fixed value function, the search strategy itself plays a crucial role: diversity-first beam search is more effective than blockwise BoN.
> * In addition, IRO enables multi-iteration improvements.
>
> To better complete the paper, we put the CD as a baseline comparison in our main paper.
>
> >Q4. Can you elaborate on the computational overhead of IRO? Will there be any measure that can be taken to reduce this overhead? For example, by using LoRA?
>
> **Response**
>
> Thanks for your question. We provide a detailed computational complexity analysis in Appendix E and practical measurements of latency in the overall comment. The computational overhead of IRO mainly comes from three parts: (1). training lightweight value function; (2) generating rollout data under the guidance of the current value functions for the next iteration ; and (3) using the trained value fucntions to guide the frozen LLM during inference. For the training stage computational analysis, please refer to Weakness 1 of zS4m 6.
>
> It is worth noting that compared to the fintuning base model, IRO is very efficient. For example, fine-tuning a 70B model using DPO or RL algorithm typically requires multi-node training with lots of computational resources, whereas IRO trains only lightweight value functions (7B in our experiment), making it feasible in limited resources.
>
> For the overhead reduction, IRO is compatible with parameter-efficient training techniques. For example, LoRA can be used during the value function training part, which further reduces the training cost and inference memory cost.
>
> Finally, as noted in our overall comment, we also provide a compression strategy to compress multiple value functions into a single one. This cuts both inference-time peak memory by 18–45%  and latency by 24–40% while only dropping 0-1.7% win rate performance, providing a practical solution for deployment in more constrained environments.

---

> > ### Author Response · Authors · 2025-11-21
> > **Response 3 for Reviewer uYrT**
> >
> > >Q5. From Appendix H.2.4, I found that the value functions are initialized from the reward model. Is this process necessary? What if the reward model is a black box or not a neural network such that we can not initialize the value networks form the reward model?
> >
> > **Response**
> >
> > Thank you for your concern. Initializing the value function from the reward model is not necessary. IRO only needs the reward model to score trajectories, and the value function can be initialized from any LLM backbone with a randomly initialized scalar head. Therefore, we can initialize any value function from any LLM.
> >
> > The value function is directly trained on the generated dataset through regression on the returns scored by the reward model. In Appendix H.5, we show that value functions of different sizes are effective, which are initialized from the policy backbone with a newly added scalar head rather than a reward model.
> >
> > In addition, IRO works even when the reward model is a black box or not a neural network, as IRO only needs a grader or rule to score each trajectory. For example, in the mathematical reasoning task (Appendix H.4), we use a verifiable reward, where each trajectory receives a binary score (1 if the final answer is correct, 0 otherwise), and the value function is simply initialized from the policy backbone and a randomly initialized scalar head, IRO still works.

---

> ### Author Response · Authors · 2025-11-27
>
> Dear Reviewer uYrT,
>
> Thank you sincerely for taking the time to review our work. We sincerely appreciate your thoughtful comments and evaluations. In our response, we have addressed your concerns point by point. If you have any further questions or feedback, we would be happy to continue the discussion.
>
> Best regards,
>
> Authors

---

### Official Review · Reviewer_wRVA · 2025-11-01

**Soundness:** 2
**Presentation:** 3
**Contribution:** 2
**Rating:** 4
**Confidence:** 3

**Summary:**

This paper proposes a training framework named IRO, aiming to align large language models during inference time to better adapt to human preferences while avoiding direct modification of model weights. IRO achieves multi-round policy improvements through a sequence of lightweight value models.
The authors provide theoretical proof at the level of convergence and complexity analysis. Experimental results suggest that this method is superior to standard inference-time alignment methods.

**Strengths:**

1.	Introduces a novel approach for reweighting self-generated data to facilitate successive policy improvement.

2.	Provides thorough theoretical analysis, considering convergence and efficiency

3.	Provides insightful ablation studies, including β selection, chunk length, data volume, and reward model quality.

**Weaknesses:**

1.	The article devotes a considerable amount of space to demonstrating the rationality of the method, but provides limited explanation of its operational process.

2.	There may be overfitting or cumulative bias in multi-round iterative training of the value function, and the paper lacks quantitative analysis for this.

3.	While token efficiency is theoretically improved, runtime latency from multiple value evaluations and beam search is not quantified. For real-time systems, this trade-off is essential.

4.	The method requires running multiple models simultaneously, which may not be practical in many real-world scenarios, especially for commercial models, which limits the practical application of this method.

**Questions:**

1.	In Figure 3 and Figure 4, the experiments employ different β values across iterations. Do different β schedules lead to consistent convergence behavior across datasets, or is the performance highly sensitive to β tuning?

2.	It is worrying that in Tables 8 and 9, the higher-quality reward models bring greater benefits in the first round of iterations, but as the number of iterations of the value model increases, the winning rate actually decreases. Could the author clarify the cause of this performance decline and whether you have considered any mechanisms to regulate subsequent iterations?

3.	In practical deployment, how many iterations are typically required for the method to converge to optimal performance?

4.	Have the authors considered comparing their approach with other PRM-based methods to better demonstrate its effectiveness?

---

> ### Author Response · Authors · 2025-11-21
> **Responses 1 For Reviewer wRVA**
>
> We thank the reviewer for the detailed review of the paper and the valuable feedback. Below, we address the reviewer's questions in a point-by-point manner.
>
> **Weakness**
> >W1. The article devotes a considerable amount of space...
>
> **Response**
>
> Thanks for your advice about writing. We have revised the paper and put more explanation on the operational process.
>
> >W2. There may be overfitting or cumulative bias...
>
> **Response**
>
> Thanks for your concern about the overfitting and bias. We address overfitting and analyze the bias below.
>
> **Overfitting**
> To mitigate overfitting, we explicitly separate the rollout data into the training and evaluation set when training value function. For every iteration, we select the checkpoint that achieves the lowest evaluation loss to avoid overfitting.
>
> **Bias**
> The bias in our setting is mainly from the reward model, rather than the training process of value function. Because the value function is trained on returns scored by the reward model and the reward model is not perfect, so in each iteration, the trained value function has the bias induced by the reward model. As IRO iterates, it leads to the reward over-optimization problem: reward model’s score continues to improve, while the preference by human may decrease.
>
> In Appendix H.5, we report the comparison of IRO under different imperfect reward models, which measures the bias from the reward model. To further quantify this bias, we also provide the reward score evaluated by the 6.9B reward model in Table 5 and Table 6.
>
> #### Table 6: IRO with 1B value functions guide 1B policy on TL;DR dataset when involving different reward models. The reward score is evaluated by a 6.9B reward model, and the win rate over reference summary judged by GPT-4o.
> | IRO Iteration | RM1 Reward Score | RM1 Win Rate | RM2 Reward Score | RM2 Win Rate | RM3 Reward Score | RM3 Win Rate |
> |-----|--------|-----|----|----|-----|---|
> | Iter1 | -0.91 | 26.00% | -0.43 | 28.11% | -0.38 | 28.11% |
> | Iter2 | -0.27 | 31.33% | -0.22 | 37.55% | 0.39 | 36.00% |
> | Iter3 | -0.06 | 35.33% | 0.39 | 30.33% | 0.54 | 37.22% |
>
>
> #### Table 7: IRO with 1B value functions guide 6.9B policy on TL;DR dataset when involving different reward model. The reward score is evaluated by a 6.9B reward model, and the win rate over reference summary judged by GPT-4o
>
> | IRO Iteration | RM1 Reward Score | RM1 Win Rate | RM2 Reward Score | RM2 Win Rate | RM3 Reward Score | RM3 Win Rate |
> |-|--|------|-----|------|-|-|
> | Iter1 | 1.97 | 37.66% | 1.96 | 44.55% | 2.00 | 40.56% |
> | Iter2 | 1.98 | 48.00% | 1.92 | 56.33% | 2.10 | 55.00% |
> | Iter3 | 2.04 | 49.67% | 1.92 | 50.44% | 2.16 | 51.11% |
>
> From Tables 6 and 7, we can see that when using RM2 and RM3 to give the score, the reward score evaluated by the 6.9B reward model continues to increase, but the winrate decreases, which stems from the reward model over-optimization phenomenon.
>
>
> >W3. While token efficiency is theoretically improved...
>
> **Response**
>
> Thank you for your concern. In Appendix E, we provide a theoretical latency analysis that explicitly accounts for the cost of using multiple value functions and performing beam search. Beyond the theoretical complexity, we have included empirical measurements of both latency and peak memory usage in our overall comment, which directly quantify the overhead.
>
> >W4. For real-time systems, this trade-off is essential...
>
> **Response**
>
> Thank you for raising your concern. However, we would like to clarify that IRO specifically targets frozen LLM settings, including commercial APIs, where the base model's weight is not accessible. In these scenarios,  DPO, RLHF, or any train-based alignment method is not feasible. Thus, lightweight models (value functions) are an alternative solution for alignment.
>
> Importantly, IRO only needs lightweight value functions. Running a sequence of 7B value functions alongside a 70B base model is far more practical than fine-tuning a 70B model. In API case, IRO's advantage becomes more obvious.
>
> To quantify the actual overhead, we report measured decoding latency and peak memory usage (Tables 3–4 in the overall comment). The results show:
> * When using small value functions, the overhead remains modest and practical compared against the BoN method. For example, when using 1b value functions to guide 6.9B policy, the extra average decoding time of IRO-Iter3 is 1s (67.32%), and 18.51s (42.9%) when using 7B value function guide 70B.
> * Compressed-IRO further reduces cost and latency by merging multiple value functions into a single one while maintaining performance. In this case, the extra latency is only 0.45s (≈29%) for 1B guiding 6.9B and 2.12s (≈4.9%) for 7B guiding 70B, nearly identical to BoN.
>
>
> Therefore, IRO provides an alternative solution for fine-tuning LLM or API in real-world applications.

---

> ### Author Response · Authors · 2025-11-21
> **Response 2 for Reviewer wRVA**
>
> **Questions**
> >Q5. In Figure 3 and Figure 4, the experiments employ different β values across iterations. Do different β schedules lead to consistent convergence behavior across datasets, or is the performance highly sensitive to β tuning?
>
> **Response**
>
> Thank you for the question. In IRO, the $\beta$ schedule controls the exploration–exploitation trade-off across iterations. Different $\beta$ leads to different convergence trajectory (how aggressively we exploit previous value functions vs. explore new value function), but we observe that IRO is not highly sensitive to the choice of it.
>
> Specifically, $\beta$ controls how strongly the current policy relies from the previous value function. A larger $\beta$ makes the update more conservative, while a smaller $\beta$ explore more based on the newly trained value function.
>
> We report the ablation study on the choice of $\beta_2$ in the Appendix Section H.1.5 and H.2.6. Across a wide range of $\beta_2$ (0.6-2.5), all settings on instruct following outperform a strong BoN baseline, and the variance remains small, 1.27% on the summarization taks and 4.05% on a 200-subset of Alpaca. This suggest that IRO is not highly sensitive to $\beta$.
>
>
>
> >Q6. It is worrying that in Tables 8 and 9, the higher-quality reward models bring greater benefits in the first round of iterations, but as the number of iterations of the value model increases, the winning rate actually decreases. Could the author clarify the cause of this performance decline and whether you have considered any mechanisms to regulate subsequent iterations?
>
> **Response**
>
> Thank you for pointing out this observation. As shown in Tables 5 and 6 in Weakness 2, when using stronger reward models (RM2 and RM3), IRO achieves larger improvements in the first and second iterations. However, in the third iteration, the GPT-4 win rate decrease even though the reward score (evaluated by the 6.9B RM) continues to increase.
>
> This behavior arises from reward hacking: as the value functions are iteratively trained on the reward model’s signal, the policy is pushed toward regions where the reward model assigns high scores.
>
> Stronger reward models provide a more accurate signal, enabling IRO to improve rapidly under RM2 and RM3. Once the policy reaches the regions far from region where reward is well-calibrated, the third iteration may cause the RM score to increase while the GPT-4 win rate decreases.
>
> In contrast, the weaker RM1 provides a less accurate signal, resulting in slow improvements. Notably, as shown in  Table 11, RM1 also exhibits a mild decline (49.67% to 48%) when running the 4 iterations, consistent with the reward hacking problem.
>
> In practice, we can evaluate IRO's performance on a small held-out validation set, then select the iteration with the best performance to avoid reward hacking.
>
> >Q7. In practical deployment, how many iterations are typically required for the method to converge to optimal performance?
>
> **Response**
>
> In our experiment, we find that three iterations offers a good trade-off between performance and efficiency.
>
>
> >Q8. Have the authors considered comparing their approach with other PRM-based methods to better demonstrate its effectiveness?
>
> **Response**
>
> Thanks for your suggestion. Considering that PRM is mainly used in math reasoning task, we evaluated IRO against PRM-based method using two widely PRMs:
>
> 1. PRM1: RLHFlow/Llama3.1-8B-PRM-Deepseek-Data
> 2. PRM2: MathShepherd-PRM-7B.
>
> To ensure fairness, both PRM baselines are applied using a beam-search decoding strategy with the same compute budget as IRO. Concretely, during decoding, we expand candidates at each delimiter “\n\n” (following standard PRM usage), query the PRM for a scalar score, and select the top-scoring candidates to continue the beam.
>
>
> #### Table 8: Comparison between PRM and IRO on math reasoning task. Here, the base model is Qwen-3B. IRO operates with $K=4, B=4, L=8$.
> | Method              | GSM8K (%) | Math500 (%) |
> |---------------------|-----------|--------------|
> | Qwen-3B             | 44.80     | 21.00        |
> | IRO Iter1           | 79.60     | 31.50        |
> | IRO Iter2           | 81.20     | 33.40        |
> | PRM1  Beam Search | 76.20| 31.80 |
> |    PRM 2 Beam Search |70.00 |31.00|
>
> It is observed that both PRMs provide limit improvements, demonstrating that the trained token-level value function in IRO is effective and tailored to the rollout distribution.

---

> ### Author Response · Authors · 2025-11-27
>
> Dear Reviewer wRVA,
>
> Thank you sincerely for taking the time to review our work. We sincerely appreciate your thoughtful comments and evaluations. In our response, we have addressed your concerns point by point. If you have any further questions or feedback, we would be happy to continue the discussion.
>
> Best regards,
>
> Authors

---

### Author Response · Authors · 2025-11-21
**Overall Comment**

**Memory and Latency**

Considering several reviewers raised concerns about the inference time memory and latency of our method IRO. Here we provide (1) a detailed analysis of latency and memory cost and (2) an alternative solution to reduce the computational overhead: compressing multiple value functions into a single value function. With this compression, IRO requires loading only one lightweight value function at inference time, which substantially lowers both latency and memory cost. This makes IRO significantly more practical and suitable for real-world deployment scenarios.


Based on the collected history data across all iterations $\mathcal{D}_{0:T-1}$, we use trained value functions to provide a token-level label for every token in every trajectory. Then we train a single compressed value function using these annotated token level label. In our experiment with three iterations, the objective can be written as

$$
\hat{V}^{m} := \arg \min _{V} ~ \mathbb{E} _{\tau \sim \mathcal{D} _{0:2}} \bigg [ \sum _{h=1}^{H} \big[ \sum _{t=0}^2\frac{1}{\beta _i}\hat{V}^{\hat{\pi} _i}(s _h)- V(s _h) \big]^2 \bigg].
$$

The resulting value function $\hat{V}^{m}$ effectively distills the guidance signals from all iterations into a single lightweight value function. At inference time, we therefore replace the full set of $T$ value functions with just one compressed value function, which greatly reduces both memory cost and latency.

We evaluate this compression approach on the summarization task and instruct following task. We also report the latency and peak memory usage measurements to demonstrate the practical benefits of this strategy.


#### Table 1: Reward score evaluated by a 6.9B reward model and win rates against reference summary evaluated by GPT4o-mini on 300 samples from test dataset. We put a baseline from the paper for comparison. 1B->1B means that using 1B value functions to guide 1B policy, and 1B->6.9B means that using 1B value functions to guide 6.9B policy. (New experiment are displayed in bold.)
|Method|Reward (1B→1B)| Win Rate (1B→1B %) | Reward (1B→6.9B) | Win Rate (1B→6.9B %) |
|-|-|-|-|-|
|SFT|-4.52|6.00|-1.33|16.33|
|ARGS|-3.98|9.67| -0.18|22.00|
|CARDS|-2.58|12.33|-0.36|23.00|
|IRO Iter1|-0.91|26.00|1.98|37.67|
|IRO Iter2|-0.27|31.33|1.97|48.00|
|IRO Iter3|-0.06|35.56|2.04|49.67|
|DPO|-1.06|20.11|1.86|48.00|
|BoN 16|-1.13|12.67|1.39|29.33|
|**IRO Compressed**| **-0.11** | **33.89** |**2.06**|**48.78**|

#### Table 2: AlpacaEval 2 length-controlled win rate and raw win rate compared against GPT-4 for Llama-3-8B-Instruct model (New experiment are displayed in bold.)
| Method                     | Compared Against GPT-4 LC Win Rate | Compared Against GPT-4 Win Rate | Compared Against BoN-E LC Win Rate | Compared Against BoN-E Win Rate |
|-|-|-|-|-|
| Meta-Llama-3-8B-Instruct   | 30.71  | 29.63  | 38.77 | 38.32|
| ARGS| 30.62  | 30.35| 36.51 | 37.38 |
| BoN-I 16| 35.25 | 32.67| 43.52| 40.68 |
| BoN-E 16 | 39.61 | 38.14  | 50.00 | 50.00 |
| weak to strong search      | 36.20 | 35.90 | 44.13| 42.92|
| IRO Iter1 | 41.06 | 37.32 | 55.42 | 53.60 |
| IRO Iter2 | 42.20| 40.00 | 55.75 | 55.65 |
| IRO Iter3  | 43.11| 41.00 | 57.00 | 56.46 |
|**IRO Compressed**| **42.59** | **39.63** | **58.92** | **56.92** |

#### Table 3: AlpacaEval 2 length-controlled win rate and raw win rate compared against GPT-4 for Llama-3-70B-Instruct model (New experiment are displayed in bold.)
| Method                     | Compared Against GPT-4 LC Win Rate | Compared Against GPT-4 Win Rate | Compared Against BoN-E LC Win Rate | Compared Against BoN-E Win Rate |
|-|-|-|-|-|
| Meta-Llama-3-70B-Instruct | 43.11                  | 43.11     | 42.39                  | 41.80     |
| ARGS                      | 43.02                  | 43.21     | 42.01                  | 41.37     |
| BoN-I 16                  | 44.45                  | 44.67     | 44.68                  | 43.68     |
| BoN-E 16                  | 47.45                  | 47.58     | 50.00                  | 50.00     |
| weak to strong search     | 45.37                  | 46.21     | 50.00                  | 47.45     |
| IRO Iter1                 | 49.00                  | 48.32                  | 48.32     | 52.86     |
| IRO Iter2                 | 49.75                  | 55.69                  | 49.19     | 53.91     |
| IRO Iter3                 | 49.77                  | 55.62                  | 49.07     | 55.45     |
| **IRO Compressed**        | **49.24**              | **55.56**              | **48.83** | **54.29** |

---

> ### Author Response · Authors · 2025-11-21
>
> #### Table 4: Average decoding time (s), evaluated over 30 responses.  Here 1b->1b means that using 1b value function to guide 1b policy. All first three are conducted under one H100 GPU, 7b->70b is conducted on 4 H100 GPUs.
> | Method      | 1b->1b | 1b->6.9b|7b->8b|7b->70b|
> |-|-|-|-|-|
> | Policy Only | 0.40                      | 0.82                     |7.37 | 34.20
> |BoN| 0.50| 1.53|10.12|43.14
> | IRO Iter 1  | 0.94                     |1.98 | 12.38|45.27|
> | IRO Iter 2  | 1.22                     |2.27 | 17.42| 56.30
> | IRO Iter 3  | 1.51                     |2.56 |22.90| 61.65|
> |IRO Compressed| 0.91 | 1.95 | 12.40 | 45.26 |
>
>
>
> #### Table 5: Average peak memory (GB), evaluated over 30 response on one H100 GPU. Here 1b->1b means that using 1b value function to guide 1b policy. All first three are conducted under one H100 GPU, 7b->70b is conducted on 4 H100 GPUs.
> | Method      | 1b->1b | 1b->6.9b| 7b->8b|7b->70b|
> |-------------|---------------------------|---------------------------|---------------------------|---------------------------|
> | Policy Only |     2.21                 |   13.71                   |15.11| 32.35
> |BoN |2.45 | 14.55|17.27|33.62
> | IRO Iter 1  | 4.44                  |16.46 | 31.25| 38.68
> | IRO Iter 2  | 6.21                      |18.25 |45.00|42.05
> | IRO Iter 3  | 7.98                      |20.02 |58.82|45.17|
> |IRO Compressed| 4.47 | 16.39 | 31.23 | 38.64 |
>
>
> On the summarization task (Table 1,4 and 5), we observe the following:
> * Using compressed value function leads to only a 1.67% drop in win rate (35.56% → 33.89%) for the 1B→1B setting and 0.9% (49.67% → 48.78%) for the 1B→6.9B setting,while requiring only one value function instead of three.
> * With the compressed value function, IRO achieves a slightly higher win rate than IRO Iter2, demonstrating that most of the iterative guidance signal is preserved after compression.
> * Compression reduces peak memory by 44.0% (7.98GB → 4.47GB) in the 1B→1B case and by 18.1% (20.02GB → 16.39GB) in the 1B→6.9B case. It also reduces decoding latency by 39.7% (1.51s → 0.91s) for 1B→1B and 23.8% (2.56s → 1.95s) for 1B→6.9B.
>
> On the instruct following task (Table 2,3,4, and 5), we observe a similar effect:
> * The compressed value function achieves win rates that are within 0.2–1.0% of IRO Iter3 across both the 8B and 70B policy settings. For the 8B model, compression even outperforms Iter3 when compared against BoN-E (58.92% vs. 57.00% LC win rate).
> * Across all metrics (GPT-4 comparisons and BoN-E comparisons), the compressed model performs slightly better than IRO Iter2, despite using only a single value function.
> * Compression preserves almost the same latency and peak memory as Iter1. For example, in the 7B→8B setting, latency drops from 22.90s → 12.40s, and memory drops 45.21% (58.82GB → 31.23GB).
>
> Overall, compressing multiple value functions effectively lowers inference memory cost and latency without significantly compromising performance, exceeding the winrate performance of iteration 2 and only slightly below those of iteration 3.

---

### Author Response · Authors · 2025-12-02
**Summary to AC**

Dear AC,

We sincerely appreciate the time and effort you devoted to reviewing our work. Due to the unfortunate circumstances of this year's conference, that resulted into the early termination of the discussion phase, we didn't receive any response from the reviewers to our rebuttal. In hopes of easing your increased workload, we provide a summary of the reviews and our responses. We believe the extensive new experiments and clarifications address the reviewers' initial concenrs.

First, we would like to highlight that across all reviews, the paper was consistently recognized for:

**Novelty and Practical Importance**: Introducing a new iterative RL framework that aligns and steers frozen LLMs using a sequence of external value functions—enabling successive policy improvement where prior methods offer only one-shot guidance.

**Strong Theoretical Foundation**: Clear connections to TRPO-style updates, along with convergence guarantees and a formal cost-efficiency analysis, were praised as elegant and rigorous.

**Comprehensive Empirical Validation**: Extensive experiments across multiple model families and scales, strong results on standard benchmarks, generalization to math reasoning, and impactful weak-to-strong guidance (e.g., 7B guiding 70B).

**High-Quality Ablations**: Thorough studies on β schedules, chunk length, data size, reward quality, and value function size that justify key design choices.

**Clarity and Relevance**: Reviewers found the method clearly explained and highly relevant for the important problem of steering large models without weight updates.


Secondly, we are grateful for constructive comments provided by reviewers, and we believe that we have addressed all of them, via developing a simple compression methods that extended our approach, and by additional experiments that addresses questions such as memory cost, latency during inference, and ablation for value function training.


Below, we provide a brief rebuttal summary for each reviewer.


### Reviewer wRVA:
Soundness: 2: fair, Presentation: 3: good, Contribution: 2: fair, Rating: 4

Strengths: novel approach for reweighting self-generated data to facilitate successive policy improvement; thorough theoretical analysis.

Weaknesses: provides limited explanation; quantitative analysis for cumulative bias analysis; requires runtime latency; running multiple models simultaneously limits the practical usage; other PRM-based methods.

Rebuttal: We provide a method to compress multiple value functions, significantly reducing latency and memory while preserving performance in Sections 3 and 5. We provide detailed latency and memory metrics in Section 5.3 and add a comparison with PRM-based methods, showing the effectiveness of our approach. Also, we use different reward models under IRO to quantify that the cumulative bias is mainly from the reward model's bias.


### Reviewer wRVA:
Soundness: 3: good, Presentation: 3: good, Contribution: 2: fair, Rating: 4.

Strengths:  idea is clear; steering a frozen LLM is an important problem; Theoretical claims seem valid.

Weaknesses: loading multiple value functions adds memory cost and latency; lack of comparison between controlled decoding; more detail analysis computational overhead

Rebuttal: We provide a method for compressing multiple value functions, significantly reducing latency and memory usage while preserving performance in Sections 3 and 5. We provide detailed latency and memory analysis in Section 5.3. We additionally add a comparison to controlled decoding methods on both summarization and the Instruct following task, and provide a detailed computational-cost theoretical analysis in Appendix E.

### Reviewer zS4m:
Soundness: 3: good, Presentation: 4: excellent, Contribution: 3: good, Rating: 6.

Strengths:
High Significance and Novelty significantly extends beyond existing one-shot inference methods; Strong Theoretical Foundation, well grounded in RL theory; Comprehensive Empirical evaluation on TL;DR and Ultrafeedback dataset, and generality beyond continuous rewards; Strong Baselines and "Weak-to-Strong" Results like 7b guiding 70B; Thorough Ablation Studies on different parameters.

Weaknesses: lakc of training cost analysis for IRO; Hyperparameter Sensitivity such $\beta$ schedule; inference latency when increasing iterations; clarification on diversity-first principle, and reward quailty.

Rebuttal: We provide a method to compress multiple value functions, significantly reducing latency and memory while preserving performance in Sections 3 and 5. We provide detailed latency and memory analysis in Section 5.3. We provide training cost analysis for IRO in Appendix E and demonstrate in Appendix H that most hyperparameters, including the β schedule, can be fixed. We also demonstrate that 3 iterations provide a trade-off between effectiveness and efficiency. We also clarify the diversity-first principle in Appendix C and Section 3, and reward-quality considerations in Section 5.

---

> ### Author Response · Authors · 2025-12-02
>
> ### Reviewer 57N6:
> Soundness: 2: good, Presentation: 2: excellent, Contribution: 2: fair, Rating: 4.
>
> Strengths:
> Address crucial problem and idea is novel and enables multiple improvements rather than one-shot; the framework is well-motivated by established RL theory, supported by elegant theoratical analysis.
>
> Weaknesses: lack of comparison of value function training using low-variance methods such as GAE and TD; mismatch between theory and practical implementation; lacks concrete measurements of inference latency and memory.
>
> Rebuttal:We provide a solution to compress multiple value functions to further improve latency and memory while maintaining performance in Sections 3 and 5. In addition, we provide a detailed memory and latency in Section 5.3 to show the effectiveness of IRO. We provide a detailed computational-cost theoretical analysis in Appendix E. We include new results in Appendix H.5 comparing value-function training methods such as GAE, TD, and FUDGE, showing that FUDGE loss has a good performance in training the token-level value function. Section 5.3 now provides concrete latency and memory metrics, and we clarify the connection between theory and practice throughout the revised sections.
>
>
> We hope this summary helps contextualize our revisions. Across all reviews, we have added extensive new experiments, analyses, and clarifications that directly address the concerns.
>
> Thank you again for your time and consideration.

---

### Meta-Review · Area_Chair_cCki · 2026-01-09

**Summary:**

This paper introduces a new inference-time alignment approach by learning a sequence of value functions.

Strengths

- Framing inference-time alignment through the learning of a sequence of value functions.

- Thorough theoretical analysis of convergence and efficiency.

- Extensive experimental evaluation.

Weaknesses

- Lack of cost analysis. During the rebuttal, the authors provide cost comparisons to Best-of-N and Policy-Only methods. However, given that several inference-time alignment algorithms already exist (e.g., ARGS, CARDS, Controlled Decoding), it is necessary to compare against these baselines as well.

- Potentially high training cost for value functions. To address this issue, the authors propose a new algorithm during the rebuttal that compresses multiple value functions into a single one.

Remaining concerns after the rebuttal

- Since the main goal of the algorithm is to reduce inference-time cost, the paper does not provide sufficient evidence that this goal is achieved. It remains unclear how the proposed method compares, in terms of cost, to existing inference-time alignment approaches.

- The compressed value function is an important component to support the paper’s claims. However, because it is introduced during the rebuttal, its analysis and experimental evaluation are limited. In particular, the performance comparison between the original and compressed versions of IRO is not clearly explained.

Given these remaining concerns, I recommend that the authors carefully revise the paper and consider submission to a future venue.

**Reviewer Concerns:**

Addressed

- Number of required iterations
- Discussion of connection to clockwise BoN
- Hyperparameter sensitivity

Unaddressed

- The proposed algorithm induces a high cost

- Reward overoptimization occurs as the number of iterations of the value function increases

**Reviewer Scores:**

Likely remain similar.

---

### Decision · Program_Chairs · 2026-01-26

Reject